

# Exploring the Use of Multi-source High-Resolution Satellite Data for Snow Water Equivalent Reconstruction over Mountainous Catchments

Valentina Premier[1,3], Carlo Marin[1], Giacomo Bertoldi[2], Riccardo Barella[1], Claudia Notarnicola[1], and Lorenzo Bruzzone[3]

[1]Institute for Earth Observation, Eurac Research, Viale Druso, 1 - 39100 Bolzano, Italy
[2]Institute for Alpine Environment, Eurac Research, Viale Druso, 1 - 39100 Bolzano, Italy
[3]Department of Information Engineering and Computer Science, University of Trento, Via Sommarive, 9 I - 38123 Povo, Italy

**Correspondence:** Valentina Premier (valentina.premier@eurac.edu)

**Abstract.** Seasonal snow accumulation and release are so crucial for the hydrological cycle to the point that mountains have been claimed as the "water towers" of the world. A key variable in this sense is the snow water equivalent (SWE). However, the complex accumulation and snow redistribution processes render its quantification and prediction very challenging. In this work, we explore the use of multi-source data to reconstruct SWE at a high spatial resolution (HR) of 25 m by proposing a novel

approach designed for mountainous catchments. To this purpose, we exploit i) daily HR time-series of snow cover area (SCA) derived by high- and low-resolution optical images to define the days of snow presence, ii) a degree-day model driven by in-situ temperature to determine the potential melting, and iii) in-situ snow depth and Synthetic Aperture Radar (SAR) images to determine the state of the catchment (i.e., accumulation or ablation) that is needed to add or remove SWE to the reconstruction. Given the typical high spatial heterogeneity of snow in mountainous areas, HR data sample more adequately its distribution thus

resulting in a highly detailed spatialized information that represents an important novelty. The proposed SWE reconstruction approach also foresees a novel SCA time-series regularization from impossible transitions. Moreover it reconstructs SWE for all the hydrological season without the need of spatialized precipitation information as input, that is usually affected by uncertainty. Despite the simple approach based on a set of empirical assumptions, it shows good performances when tested in two different catchments: the South Fork catchment, California, and the Schnals catchment, Italy, showing a good agreement

with an average bias of -40 mm when evaluated against a HR spatialized reference product and of 38 mm when evaluated against manual measurements. The main sources of error introduced by each step of the method have been finally discussed to provide insights about the applicability and future improvements of the method that may be of great interest for several hydrological and ecological applications.

## 1 Introduction

Seasonal snow accumulation and melting are of crucial importance for the hydrological cycle and the total water supply. Especially in mountainous areas, the snow has such a large impact on the local hydrology that mountains have been claimed to be





"the water towers of the world" (Immerzeel et al., 2020). For example, in the Alps the snowmelt contribution to the streamflow ranges from at least 50% of the total flow to sometimes over 95% (Viviroli et al., 2003). Hence, it is essential to estimate the amount of water stored during the winter not only for river discharge forecasting but also for a correct planning of human
activities such as agriculture irrigation, drinking water supply and hydropower production (Beniston et al., 2018; DeWalle and Rango, 2008). However, especially in mountain regions, snow distribution is highly variable in space due to redistribution processes (Balk and Elder, 2000) and precipitation observations are often affected by large errors due to orographic effects (Prein and Gobiet, 2017). This also limits the spatial accuracy of snow accumulation and melt models (Engel et al., 2017; Günther et al., 2019). In this context, given the difficulty to dispose of spatialized and continuous observations especially in remote
areas (Rees, 2005), remote sensing (RS) has shown to be a valuable tool for snow hydrology, .

A spatial characterization of the snow properties requires both information about the extent of the snow cover, i.e., the snow cover area (SCA), and appropriate snowpack information. A key variable is the snow water equivalent (SWE), i.e., the total amount of water stored in the snowpack that would be released upon complete melting While a long list of methods for SCA
detection that exploit multispectral optical satellites is available in the literature (see Dietz et al. (2012) for a review), we do not have operational methods to directly map SWE with high spatial resolution (HR). Direct SWE observations are limited to point measurements trough manual sampling, snow scales or snow pillows (Archer and Stewart, 1995; Meløysund et al., 2007), or with a limited spatial footprint ($\sim 500$ m) as cosmic-ray neutron probes (Schattan et al., 2019). Spatialized snow depth (SD) information can be provided by differential lidar altimetry (Painter et al., 2016) or stereo photogrammetry (Deschamps-Berger
et al., 2020). Currently, these methods can be applied only to limited areas and with a low temporal sampling. Moreover, to derive SWE from SD additional a priori information is needed to infer the snow density (Helfricht et al., 2018). Physically-based snow models represent a valid alternative (e.g., Lehning et al., 2006; Vionnet et al., 2012; Endrizzi et al., 2014) that can provide HR SWE information on large areas. However, their accuracy is strongly limited by the availability of meteorological observations and by the gravitational and wind-induced snow redistribution processes (Jost et al., 2007; Mott et al., 2018).
Active and passive microwave sensors can potentially provide information about the snowpack. In particular, passive microwave sensors have been used to retrieve long time-series of SWE by exploiting the correlation between the brightness temperature and the SWE (Pulliainen et al., 2020). However, the observations are limited by a poor spatial resolution (i.e., 25 km) and mountain areas are excluded. The use of active microwave sensors such as Synthetic Aperture Radars (SAR) has also been investigated for the HR retrieval of SWE (Shi et al., 1994; Baghdadi et al., 1997; Ulaby et al., 1981; Rott et al., 2010)
and differential SWE (Guneriussen et al., 2001; Leinss et al., 2015). Despite the better spatial resolution also active microwave sensors suffer for the complexity of non-linear effects introduced on the total backscattering, such as snow layering, surface roughness, snow density, grain type and size which in turn are all affected by the complex snow metamorphism and change in time. Moreover, all these techniques work only in dry conditions while the scarce penetration of the electromagnetic signal in wet conditions is invalidating their applicability in monitoring the SWE evolution during the melting season. Several review
articles are available for more details about SWE retrieval using SAR acquisitions (e.g., Tsang et al., 2021).





Even though SAR is still far from providing unambiguous information on SWE for all situations, it represents a promising tool to monitor the melting phases of the snowpack, i.e., the moistening, ripening and runoff phases or in other words, the presence of liquid water inside the snowpack and its evolution (Marin et al., 2020). If combined with optical data, the runoff onset, i.e., the time when SWE reaches its maximum, does add value to the well known concept of the snow depletion curves (SDC). SDC are functions that describe the relationship between SCA and SD or SWE (Cline et al., 1998). Thus, time-series of SCA maps can be used to provide an indirect measurement of SWE (Yang et al., 2022). Indeed, SWE is a function of the duration of the snow cover, which intrinsically considers the energy exchanges responsible of the melting process (Durand et al., 2008). For example, a shallow snowpack and high melt rates are associated with a SDC with high derivative while a deep snowpack and low melt rates are characterized by a longer curve. Consequently, spatial accumulation and melt variability, which are linked to the geomorphology of the study area (Anderton et al., 2002), result in different snowpack persistence (Luce et al., 1998). Therefore, by knowing the SDC and the maximum of SWE at the end of the accumulation for an area or an entire catchment, it is possible to derive the evolution of SWE during the melting. This intuitive idea opens the possibility to assimilate SCA and SDC into physically based snow models to correct the SWE evolution and improve the simulations (Arsenault and Houser, 2018).

In fact, another way to exploit the SDC for SWE estimation is the combination with distributed snowmelt models to reconstruct SWE time-series in re-analysis (Martinec and Rango, 1981; Molotch and Margulis, 2008; Rittger et al., 2016). Differently from the methods that require to know the precipitations and the meteorological forces that redistribute the snowpack during the accumulation, SWE reconstruction builds the SWE timeseries backward from the last day of snow presence up to the peak of accumulation by exploiting the estimation of the potential melt energy and the knowledge about the presence of snow cover, simplifying in this way the problem. SWE reconstruction approaches show good performances over large basin and even mountain ranges, outperforming the accuracy provided by snow models or spatial interpolation approaches of in-situ SWE measurements (Bair et al., 2016). Nevertheless, the accuracy of the results depends on a robust estimation of both the SCA and the melt energy. For this purpose, several methods have been proposed for the computation of the potential melt energy that range from a simple yet robust degree day (DD) model (Martinec and Rango, 1981) to a complete radiation energy computation that takes into account also the snow albedo (Bair et al., 2016). These models generally consider a calibration factor that balances out the possible inaccuracies providing accurate results. Furthermore, the derivation of the HR SCA is hampered by the cloud presence (e.g., Premier et al., 2021). In this regards the works presented in the literature exploited only low-resolution (LR) images since the large swath allows a high repetition time, i.e., with daily or sub-daily acquisitions, mitigating in this way cloud obstruction. However, the LR images are not providing the spatial details, in the Shannon sense, that allow a proper sampling of the snow cover evolution in the mountains, which is in the order of few dozen of meters. Moreover, the use of LR sensors results in a non-linear combination of the different contributions of the elements within the pixel and this should be properly taken into account by the snow classification approaches to avoid large errors especially in complex terrains. On the other hand, the use of HR snow maps introduces important benefits both in SWE determination as well as in streamflow forecasting (Li et al., 2019). With the introduction of the Copernicus Sentinel-2 (S2) mission, the HR images are made available free of charge with a temporal resolution at the equator of 5 days. This opens new opportunities to monitor the heterogeneous





snow conditions in the mountains. However, due to the cloud coverage, the useful acquisitions are reduced by 50% in the Alps (Parajka and Blöschl, 2006). Thus, even if the Landsat images are exploited together with the S2 images, only few acquisitions are available per month. Recently we proposed an approach to the reconstruction of daily HR snow cover maps. The approach

performs a gap-filling and a downscaling of snow cover fraction (SCF) maps derived at LR based on the idea that melting and accumulation patterns are repeating inter-annually (Premier et al., 2021; Revuelto et al., 2021). Therefore, by observing partial HR or LR acquisitions it is possible to reconstruct a daily HR snow cover.

In this paper we explore multi-source satellite data to reconstruct HR SWE for a given catchment. The approach exploits: i)

daily HR snow cover time-series derived by fusion of high- and low-resolution optical sensors to determine the dates of snow appearance and disappearance, ii) potential melting derived by in-situ temperature observations with a degree day (DD) model, iii) in-situ SD/SWE observations to determine the *accumulation* state, and iv) SAR information to determine the *ablation* state. In detail, the method starts with the determination of the catchment state, i.e. *accumulation* (SWE increase) or *ablation* (SWE decrease). According to the state, that is assumed to be homogeneous for all the pixels of the catchment, we regularize the HR

SCA time-series from impossible transitions to correctly estimate the dates of snow appearance and disappearance for each pixel. This simplificative assumption is a consequence of technological limitations in spatializing this information. The state information is needed together with the potential melting to reconstruct the daily HR SWE maps with a resolution of 25 m by adding or removing SWE according to the catchment state. Note that the reconstruction also includes the accumulation phase without the need of precipitation data as input, which are often unreliable over complex terrains. The main novelties of

the proposed approach are: i) the generation of daily HR SWE maps, ii) the regularization of the daily HR SCA time-series from impossible transitions, and iii) the precipitation independent SWE reconstruction. The approach has been validated in two catchments: i) the South Fork catchment, located in the Sierra Nevada - California (USA), and ii) the Schnals catchment, located in the Alps - South Tyrol, Italy.

The paper is structured into five sections. Sec. 2 presents the different steps of the proposed approach to reconstruct daily HR SWE. The two test sites and the used dataset are presented in Sec. 3. The obtained results are illustrated in Sec. 4. A detailed discussion on the approach advantages and limitations and on the results follows in Sec. 5. Finally, Sec. 6 draws the conclusions of the work and gives indications for further exploitation of the proposed approach.

## 2 Proposed approach to HR SWE reconstruction

In this section, the proposed approach to HR SWE reconstruction will be presented. The approach is made up of three main parts: i) the identification of the catchment state, ii) the characterization of the snow season from the regularized SCA time-series, and iii) the SWE calculation. The details will be illustrated in the three next subsections. As depicted in Fig. 1, the method initially determines the state of the catchment (see Sec. 2.1). This allows to properly reconstruct SWE in case of solid precipitations after the peak of accumulation and also to redistribute the total amount of SWE calculated for the melting in





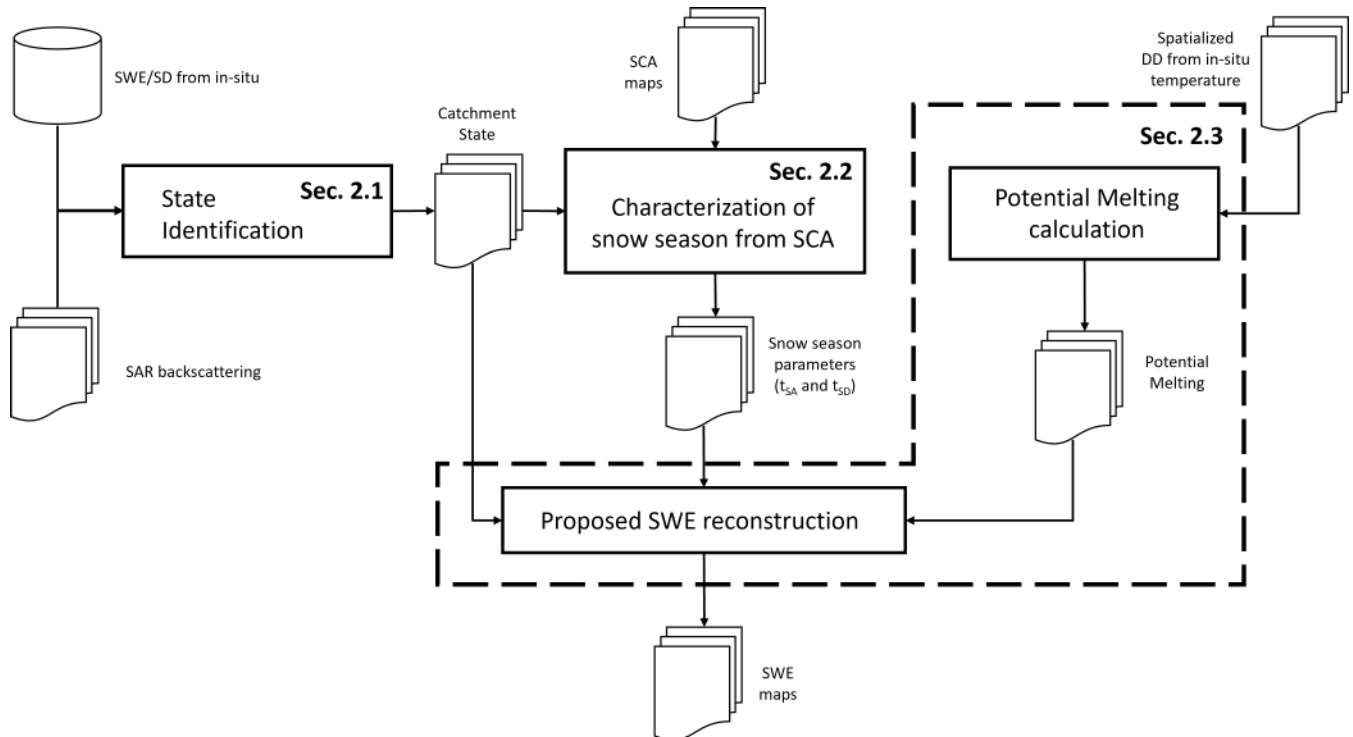

**Figure 1.** Workflow of the proposed approach showing the three main steps: i) state identification, ii) characterization of snow season from SCA, and iii) SWE reconstruction. The inputs are: i) a HR SCA time-series and daily spatialized DD maps derived from AWS, ii) SAR backscattering time-series, and iii) SWE or SD from automatic weather stations (AWS). As output, we obtain the daily HR SWE maps.

the accumulation period. Both cases are generally omitted in state-of-the-art SWE reconstruction methods (e.g., Molotch and Margulis, 2008; Martinec and Rango, 1981). Moreover, the catchment state information is used to regularize the time-series of SCA from impossible transitions and to correctly determine the beginning and end of the season (see Sec. 2.2). Finally, from the potential melting and the regularized time-series of SCA the proposed approach reconstructs the daily HR SWE maps (see Sec. 2.3).

## 2.1 Identification of the catchment state

The proposed reconstruction approach is designed for a hydrological catchment that is subjected to similar weather forcings and consequently to similar accumulation and ablation events i.e., the catchment is not too vast. If we take this assumption strictly, we can define three *states* that describe the three possible SWE changes, i.e., $\Delta$SWE between two times $t-1$ and $t$. These states are *accumulation* ($\Delta$SWE $> 0$), *ablation* ($\Delta$SWE $< 0$) and *equilibrium* ($\Delta$SWE $= 0$), as shown in Fig. 2. The identification
of the state is necessary in the proposed approach to decide whether SWE is added or removed for the reconstruction (see





Sec. 2.3). In detail, the catchment state does not only characterize the change in terms of SWE but also in terms of SCA, if observable, as described as follows:

- *Accumulation* when the total $\Delta\text{SWE} > 0$ due to a snowfall. We can either observe i) a positive $\Delta\text{SCA}$, if the snow covers the bare ground, or ii) $\Delta\text{SCA} = 0$, if the snowfall happens at higher elevation than the actual snowline. However, $\Delta\text{SCA}$ is never negative and consequently a single pixel can not turn from *snow* to *snow-free* from $t-1$ to $t$. So, depending on the extension of the snowfall, the spatial SWE increment can interest the total snow covered area or a part of the catchment. It is usually a rapid event - one to a couple of days (Thorp and Scott, 1982).

- *Ablation* when the total $\Delta\text{SWE} < 0$ due to any kind of energy exchanges (as increasing temperature, solar radiation, or rain-on-snow). We can either observe a negative $\Delta\text{SCA}$, if the snowpack is completely melted out and shows the bare ground, or $\Delta\text{SCA} = 0$, if the melting is only affecting the height of the snowpack. However, $\Delta\text{SCA}$ is never positive and a pixel can not turn from *snow-free* to *snow* from $t-1$ to $t$. So, depending on the amount of energy and on the depth of the snowpack, the spatial SWE decrement can interest the total SCA or a part of the catchment. It can be a long period, especially in the last phase of the season.

- *Equilibrium* when $\Delta\text{SWE} = 0$. This is the case of a steady state, i.e., no changes within the catchment, or redistribution due to wind or gravitational transport, e.g., avalanches. This last condition does not affect the overall SWE balance that remains constant, even though we can have local deposition and erosion looking at a pixel scale. In terms of $\Delta\text{SCA}$, these changes are mainly increments of *snow-free* pixels, hence a pixel can not turn from *snow-free* to *snow* from $t-1$ to $t$. In fact, we expect that snow moves from exposed areas - that can become *snow-free* - to sheltered areas - that were already covered by snow, due to the terrain properties that facilitate snow deposition.

To determine the catchment state according to the aforementioned definitions, it is clear that $\Delta\text{SCA}$ is ambiguous and consequently we need to estimate if SWE is increasing, decreasing or is constant. This information can be retrieved by a network of automatic weather station (AWS) that provide continuous information about the occurrence and elevation of snowfall events, e.g., direct SWE measurements or indirect precipitation/SD measurements. While continuous SWE measurements are hardly available, by mean of pluviometers and temperature observations it is possible to split precipitation between liquid and solid (Mair et al., 2013) and identify the catchment state accordingly, but with the limitation that they are rarely installed at high elevations. SD sensors are more suitable for our purpose but their observations are often affected by wind and gravitational transport leading to deposition/removal that may be falsely interpreted as *accumulation*/*ablation*. Hence, even if the AWS are generally situated in locations undisturbed from the wind action, it is more convenient to dispose of a large number of AWS that need to be screened to exclude possible sensor errors or wind/gravitational redistribution. For these reasons, the SWE/SD increment should be greater than a certain threshold that is fixed at 2 cm for SD according to values found in literature (Engel et al., 2017), resulting in a value of 2 mm for SWE when considering the typical density of fresh snow ($100\,\text{kg/m}^3$).

Nevertheless, the general scarce availability of distributed measurements inside a given catchment render the localization of the accumulation and ablation events very challenging and therefore the definition of the catchment state prone to error.





| Event description at catchment level | t-1 | t | ΔSWE | State |
|---|---|---|---|---|
| *Snow on bare ground* <br> Wide snowfall that involves lower elevation belts | | | >0 | Accumulation |
| | ΔSCA > 0 | | | |
| *Snow on snow* <br> Confined snowfall that involves high elevation belts | | | | |
| | ΔSCA = 0 | | | |
| *Snowpack melting* <br> Energy exchange that leads to the heating of the snowpack and to a reduction of the snow height | | | <0 | Ablation |
| | ΔSCA = 0 | | | |
| *Aereal ablation* <br> Snowpack melting that leads to the complete disappearance of snow | | | | |
| | ΔSCA < 0 | | | |
| *Steady state* <br> No changes within the catchment | | | =0 | Equilibrium |
| | ΔSCA = 0 | | | |
| *Redistribution* <br> Wind or gravitational transport | | | | |
| | ΔSCA < 0 | | | |

**Figure 2.** Definition of the three possible catchment states: *accumulation*, *ablation* and *equilibrium*. The description of the possible events that characterize the catchment between two consecutive dates is reported together with the difference in terms of SWE and SCA. As one can notice only ΔSWE can be used to unambiguously identify the three states. The illustrations represent the SCA in an idealized catchment where white and brown areas are the *snow* and *snow-free* areas, respectively and the dashed lines represent the contour lines.



Multi-temporal SAR observations have shown to be of great potentiality and to be able to detect the presence of a melted
snowpack as explained by Marin et al. (2020). In this work the relationship between the SAR backscattering and the three
melting phases have been investigated. It has been shown that when the backscattering is interested by a decrease of at least 2
dB, the snowpack is assumed to get moisted (Nagler and Rott, 2000). The minimum of the backscattering corresponds instead
with the maximum of SWE. After that moment, the snowpack starts to release water and enters in the so-called runoff phase.
This moment represents the most important contribution to the water release and can be provided in a HR spazialized manner,
i.e., marking the dates of *ablation* for each pixel.

On the other hand, at the best of our knowledge there are no remotely sensed data that can be exploited to identify the
snowfalls in a spatialized HR manner and that can be used to identify the *accumulation* at the level of each single pixel. In
the majority of the situations, only one AWS located at a given high point of the catchment (which is a common configuration
for snow monitoring) would be informative enough to identify all the accumulation events, but this may introduce errors in
particular cases of mixed conditions. For example, it is possible to observe snowfalls at high elevations, rain-on-snow at low
quotes that cause snowmelt and even steady state conditions for mid altitude belts. A correct characterization of such a situation
requires to consider different areas with different states separately, which for the moment is out of the scope of the paper. We
will discuss the limitations and possible future steps to improve this aspect in Sec. 5.1.

In summary we propose a hybrid approach to identify the state by satisfying the following necessary conditions in order
of priority: i) *accumulation* when the AWS show an increment greater than a defined threshold, ii) *ablation* when the SAR
backscattering presents a relevant drop (and not *accumulation*), and iii) *equilibrium* otherwise. In this way, even though the *accumulation* does not allow to correctly spatialize the snowfalls, a spatialized information on the *ablation* allows to distinguish
among pixels that are really subjected to melting and pixels that do not experience any change. In other words, a coexistent
*ablation* and *equilibrium* is possible. Finally, it is worth to mention that mixed situations within the same day and for the same
area are also possible given diurnal fluctuations in the meteorological forces. However in this work, we do not consider sub-
daily variations but only changes that are sampled in the temporal resolution of the exploited HR SCA time-series, i.e., one day.

## 2.2 Characterization of the snow season from regularized SCA time-series

A HR SCA is an input needed for the proposed SWE retrieval, as it is used to estimate the date of snow appearance $t_{SA}$ and
disappearance $t_{SD}$. As mentioned in the Introduction, such a product is not available directly from remotely sensed images
due to limitations in the revisit time and cloud contamination. Therefore, there is the need to reconstruct a daily HR SCA.
Among the several methods present in literature, we used the approach proposed by Premier et al. (2021), which merges the
information coming from a sparse long HR time-series and a continuous daily LR time-series acquired in the period of interest.
Gap-filling and downscaling steps are performed by applying a set of hierarchical rules based on historical analyses and geo-
morphometrical features. The main idea behind the approach is that snow patterns are persistent over time and follow a regular
distribution that is strongly dependent on the geomorphology and meteorology of the area of interest (Mendoza et al., 2020).





We refer the reader to Premier et al. (2021) for the details.

Despite the generally accurate results of the above-mentioned approach, the output HR SCA is still affected by possible
inconsistencies. Errors may arise either from the classification algorithm applied to the multi-spectral input images or from the
reconstruction approach. By applying the approach presented in Premier et al. (2021), we can highlight the presence of two
main sources of errors: i) an underestimation of snow presence in forested areas when the snow falls below the canopy and is
not visible anymore from the satellite point of view, i.e., snow on ground, and ii) the missed identification of snow-patches at
the end of the season. The first error source is due to the fact that the classification methods used for snow retrieval for both
HR and LR data relay only on the spectral information measured inside the resolution cell of the sensor without a dedicated
module for inferring the presence of snow if hidden by the canopy. This affects the detection of snow on ground particularly for
HR images since the small resolution cell is likely to contain a majority fraction of canopy especially over very dense forests.
This problem is instead mitigated for LR pixels that are likely to contain not only forested areas but also open fields where
the snow is visible, increasing in this way the possibility to detect the snow presence. Hence, in the daily time-series of SCA,
a discontinuity that happens mainly when HR images are acquired, can be identified as a local decrease of SCA. The second
error involves mostly LR images whose spatial detail is not enough to detect mixed pixels with low SCF. It is an error that
persists over time since LR acquisitions are more frequent than HR acquisitions. In other words, in both cases *snow-free* pixels
may be falsely detected, i.e., false negative (FN) errors. *Snow* pixels can also be falsely detected i.e., a false positive (FP) errors
due mainly to possible residual misclassified clouds that are identified as *snow* - this is usually an error isolated in time. All
these errors are detectable by looking at the class transitions in the time-series of each pixel: the snow presence is not smooth
in time. This results in an erroneous SWE determination that is strongly related to the persistence of the snow, requiring a
regularized time-series.

A helpful information to regularize the SCA is the state of the catchment that is daily identified by following the rules
described in the previous subsection. If the snow cover maps are coherent with the state, it follows that: i) if *accumulation*, all
the pixels can only turn from *snow-free* to *snow* or maintain their label, and ii) if *ablation* or *equilibrium*, all the pixels can
only turn from *snow* to *snow-free* or maintain their label. Since *ablation* and *equilibrium* imply the same rule as explained also
in Sec. 2.1, we will refer more generally to *ablation* only. In other words, $t_{SA}$ and $t_{SD}$ may vary for each pixel but for sake
of coherence they must coincide with an *accumulation* and an *ablation* date, respectively. Pixels that do not respect these rules
are potential mistakes that need to be corrected. When we face with an erroneous class transition, we do not know a priori if
the correct label is the one at $t-1$ or the one at $t$. To understand what is the correct solution, we consider an appropriate time
window and compute the most frequent label for it according to a majority rule. The time window is chosen in a different way
in the case that we are facing with a recent or an old date of snow appearance $t_{SA}$. In detail, for a given pixel, we consider:

– a *recent date of snow appearance* when $t - t_{SA} < 10$ days. We observe in this period FN mostly due to missed detection
of snow under canopy especially by HR sensors. In this condition, we do not expect fast changes since temperatures are





low and thus the potential melting is low (see Eq. 1). Accordingly, we propose to consider a daily time window of $\pm 5$ days from $t$ to check what is the most persistent label of the considered pixel (Parajka and Blöschl, 2006);

- an *old date of snow appearance* when $t - t_{SA} > 10$ days. If the final melting has already started, changes may be quick and the most common situation is the missed detection of mixed pixels as snow patches. We observe that in this
period HR sensors detect snow patches that are completely omitted by LR sensors. In this case: i) the dates after $t$ are not informative since the snow patches are disappearing quickly, and ii) daily SCA may not be informative when it is derived by LR. For this reason, we consider only the last up-to-5 dates when a HR was originally acquired in a time window between $t_{SA}$ and $t$.

Once we have determined the state and whether if we are handling with a recent or an old $t_{SA}$, we can compute the most
frequent label in the considered time window and apply the following correction (see Algorithm 1).

The correction is performed by advancing forward in time, i.e. we assume that previous labels are always coherent with previous states. The correction itself cannot introduce inconsistencies. In case we are in *accumulation*, a transition from *snow* to *snow-free* is not allowed. If the pixel is labelled as *snow* at $t-1$, it means that for sake of coherence it turned to *snow* at $t_{SA}$. We also know the date of the last ablation after $t_{SA}$, i.e., $t_{lastAB}$ with $t_{SA} \leq t_{lastAB} \leq t-1$. Hence, we compute the most
frequent label accordingly with the majority rules described in the previous paragraph which vary depending if the snowfall is old or recent. If it results that i) $t$ is a false negative (FN), we can simply set it as *snow*; ii) $t-1$ is a false positive (FP), we need to replace all times starting from the day of the last ablation state (or $t_{SA}$, in case the last ablation precedes $t_{SA}$) until $t-1$ with *snow-free*, since the transition *snow* to *snow-free* can happen during ablation only.

Analogously, the transition from *snow-free* to *snow* is not allowed in *ablation*. If the pixel is labelled as *snow-free* at $t-1$, it
means that for sake of coherence it turned to *snow-free* at $t_{SD}$. We also know the date of the last accumulation after $t_{SD}$, i.e., $t_{lastAC}$ with $t_{SD} \leq t_{lastAC} \leq t-1$. Hence, we compute the most frequent label accordingly with the majority rules and if it results that i) $t$ is a false positive (FP), we simply set it as *snow-free*; ii) $t-1$ is a false negative (FN), we need to replace all times starting from the day of the last accumulation state (or $t_{SD}$, in case the last accumulation precedes $t_{SD}$) until $t-1$ with *snow-free* since the transition *snow-free* tro *snow* can happen during *ablation* only.



---

**Algorithm 1** Regularization of the snow cover maps with the catchment state.

---

**if** *Accumulation* **then**

> **# Transition *snow* to *snow-free* is not allowed!** 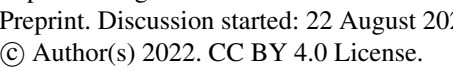

> # The pixel is *snow* from $t_{SA} \leq t - 1$. Between $t_{SA}$ and $t - 1$ all states are possible.

> # We indicate with $t_{lastAB}$ a day $t_{SA} \leq t_{lastAB} \leq t - 1$ representing the date of the last ablation after $t_{SA}$

> **if** $t - t_{SA} < 10$ *days* **then**
> > **Recent** $t_{SA}$**:** check $t \pm 5$ days and compute the most frequent label
>
> **else**
> > **Old** $t_{SA}$**:** check last up to 5 HR from $t_{SA}$ to $t$ and compute the most frequent label
>
> **end**

> **if** *most frequent label is snow* **then**
> > $t$ is a FN (e.g., missed snow under canopy): set $t$ as *snow*
>
> **else**
> > $t - 1$ is a FP (e.g., cloud detected as snow): set $[t_{lastAB}; t - 1]$ as *snow-free*
>
> **end**

**else**

> **# Transition *snow-free* to *snow* is not allowed!**

> # The pixel is *snow-free* from $t_{SD} \leq t - 1$. Between $t_{SD}$ and $t - 1$ all states are possible.

> # We indicate with $t_{lastAC}$ a day $t_{SD} \leq t_{lastAC} \leq t - 1$ representing the date of the last accumulation after $t_{SD}$

> **if** $t - t_{SA} < 10$ *days* **then**
> > **Recent** $t_{SA}$**:** check $t \pm 5$ days and compute the most frequent label
>
> **else**
> > **Old** $t_{SA}$**:** check last up to 5 HR from $t_{SA}$ to $t$ and compute the most frequent label
>
> **end**

> **if** *most frequent label is snow* **then**
> > $t - 1$ is a FN (e.g., missed snow patches): set $[t_{lastAC}; t - 1]$ as *snow*
>
> **else**
> > $t$ is a FP (e.g., cloud detected as snow): set $t$ as *snow-free*
>
> **end**

**end**

---



## 2.3 HR SWE reconstruction

Once the catchment state has been defined for each day as described in 2.1, and the daily HR SCA time-series has been regularized with the catchment state as described in section 2.2, the proposed approach to the HR SWE reconstruction can be initiated. This operation requires to calculate the total amount of melting and redistribute it during the snow season according to the preservation of the mass and the catchment state. For this purpose we estimate the daily potential melting with the degree day (DD) model. For a generic time interval $[t-1;t]$, the potential melting $M_{t-1,t}$ is estimated through the following equation:

$$M_{t-1,t}\,[mm] = a\,[mm^oC^{-1}d^{-1}] \cdot DD_{t-1,t}\,[^oCd] \tag{1}$$

where $a$ is the so called DD factor and varies depending on the considered area as well on the considered snow period. We used a value of $a = 4.5$ m$^o$ C$^{-1}$d$^{-1}$ for the South Fork catchment and $a = 5.2$ mm$^o$C$^{-1}$d$^{-1}$ for the Schnals catchment. The coefficient is calibrated by considering measured SWE and temperature at the AWSs (if available) and taking into account also the range of values derived in previous literature works (Hock, 2003). The limitations of this approach will be discussed in Section 5.3. $DD_{t-1,t}$ is the DD given by the cumulative sum of the hourly temperatures exceeding a certain threshold:

$$DD_{t-1,t} = \sum_{t-1}^{t} T_h \; if \; T_h > \hat{T} \tag{2}$$

The threshold temperature $\hat{T}$ is set to $0^oC$.

The DD is first calculated for each station and then spatially interpolated using a three-dimensional universal kriging routine with linear variogram and external drift (Murphy et al., 2020). The choice arises from the results of a leave one out (LOO) cross validation (see Appendix B). The variogram parameters are automatically calculated at each time step using a "soft" L1 norm minimization scheme. The number of averaging bins is set as 6 (default value). The kriging is performed on the daily DD values instead of on the raw hourly temperature values to reduce computational times.

We can determine the total amount of melting $M_{tot}$ by summing up all the daily $M_{t-1,t}$ for all those days in *ablation* within the time range $[t_{SA};t_{SD}]$. It is worth noting that a single pixel may have more than one single snow period, hence we can have more than a couple of $t_{SA}$-$t_{SD}$. $M_{tot}$, which has to be equal the total accumulation $A_{tot}$, is then calculated as follows:

$$A_{tot} = M_{tot} = \sum_{t=t_{SA}}^{t_{SD}} M_{t-1,t} \quad if \quad ablation \quad for \quad t \tag{3}$$

Consequently, it is possible to reallocate the total accumulation on those days which are in *accumulation*:

$$A_{t-1,t} = k_{t-1,t}A_{tot} \quad if \quad accumulation \quad for \quad t \tag{4}$$

where $k_{t-1,t}$ is a coefficient that represents the quantity of the snowfall. In case we have a network made by $S$ AWS with measured SWE (or similarly, SD), $k$ is set proportional to the observed snowfalls:

$$k_{t-1,t} = \frac{\sum_{s=1}^{S}(SWE_t^s - SWE_{t-1}^s)}{\sum_t \sum_{s=1}^{S} S(SWE_t^s - SWE_{t-1}^s)} \quad if \quad accumulation \quad for \quad t \tag{5}$$





Note that the number of days in *accumulation* varies for each pixel and consequently the coefficient is function of time and
space. Thus, it is possible to determine the final output, i.e., a daily HR SWE time-series, by applying pixel-wise the following
rules:

$$SWE(t) = \begin{cases} 0 & \text{if} \quad \textit{snow-free} \\ SWE(t-1) & \text{if} \quad \textit{equilibrium} \\ SWE(t-1) - M_{t-1,t} & \text{if} \quad \textit{ablation} \\ SWE(t-1) + A_{t-1,t} & \text{if} \quad \textit{accumulation} \end{cases} \quad (6)$$

It may happen that during *ablation* temperatures are low and the term $M_{t-1,t}$ is equal to 0, thus coinciding with the *equilibrium*
state. Note that $M$ may also be greater than 0 but if the state is different from *ablation*, that melting is not encountered. It is in
fact possible that temperatures present some inaccuracies or increase without causing a real melting. In other words, we assume
that *ablation* is only possible when we have contemporary: i) absence of snowfalls, ii) a decrease of SAR backscattering, and
iii) high temperature. The term $A_{t-1,t}$ is instead always positive. It is worth stressing the fact that even though possible
redistribution caused by wind and gravitational transport is not explicitly taken into account in Eq. 6, its consequences are
implicitly appreciated by observing a longer persistence of snow on the ground. Indeed, the potential melting is distributed
over space and time by considering the snow cover duration. This implicitly takes into account the difference in the energy
inputs due to both the topographic and the redistribution effects providing a good estimation of the total SWE. Moreover, by
providing an approximation of the accumulation events we also consider late snowfalls that may occur during the main melting
season and that are a large source of error in the state-of-the-art methods (Slater et al., 2013a).

## 3  Study Areas and Dataset Description

To assess the performance of the proposed method, we consider two different test areas. The first one is the South Fork catch-
ment located in California, USA, in the Sierra Nevada. For this test site, we considered three hydrological seasons spanning
from the 1st of October 2018 to 30th of September 2021. The considered basin has an area around 970 km$^2$ and a mean
elevation of 3070 m, ranging from a minimum elevation of 1930 m up to a maximum elevation of 4150 m. For this catch-
ment, a spatialized SWE product with a resolution of 50 m is available acquired by the Airborne Snow Observatory (ASO).
ASO couples imaging spectrometer, laser scanner and a physical model that provides an estimate of the snow density to de-
rive accurate SWE maps (Painter et al., 2016). One snow pillow for continuous SWE measurements and one AWS providing
air temperature are available inside the catchment. Moreover, we considered 6 snow pillows and 10 stations with continuous
temperature measurements within a radius of around 15 km from the catchment (see 4a). These data were downloaded from
the United States Department of Agriculture (USDA) Natural Resources Conservation Service (NRCS) Snowpack Telemetry
(SNOTEL) network (see https://www.wcc.nrcs.usda.gov/snow/) and from the California Data Exchange Center (CDEC) (see
https://cdec.water.ca.gov/).





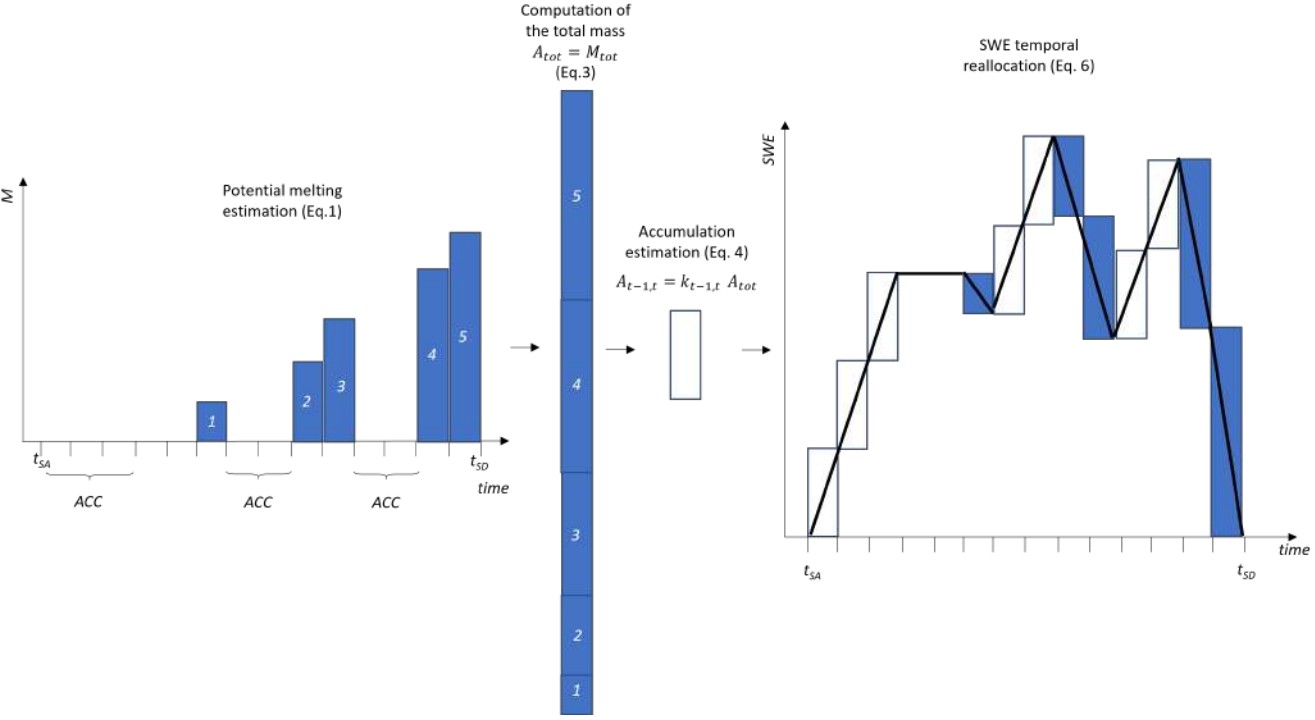

**Figure 3.** Illustration of the reconstruction and temporal reallocation of the SWE for a given pixel. Starting from the left side of the figure, the catchment state is identified for each day of the snow season (delimited by $t_{SA}$ and $t_{SD}$) and the potential melting is estimated according to Eq.1. The sum of all the potential melting at the different days represent the total amount of SWE for that pixel. This is redistributed during the *accumulation* day using Eq.4. For this illustrative example, a constant $k$ is considered. As one can notice the reconstructed SWE can represent *accumulation* (even as late spring snowfall), *ablation* and *equilibrium* conditions.

The second catchment is the Schnals (Senales in Italian - for brevity we will report the German name only) located in the Vinschgau (Venosta) Valley in South Tyrol, Italy, in the Alps. For this catchment we analyzed two hydrological seasons spanning from the 1st of October 2019 to the 30th of September 2021. The considered area has an extension of about 220
km² and a mean elevation of 2370 m, ranging from a minimum elevation of 590 m up to a maximum elevation of 3550 m. For this test site, manual SWE measurements are available (collected by Avalanche Centre of the Bolzano Province - Lawinenwarndienst - see https://lawinen.report/weather/snow-profiles - and by Eurac Research, Institute for Earth Observation). Additionally, we considered the operating temperature and SD sensors of the Province of Bozen (see https://data.civis.bz.it/ it/dataset/misure-meteo-e-idrografiche). An overview of the Schnals catchment and the location of available measurements is
provided in Fig. 4b.

The HR daily SCA time-series is derived through the method proposed by Premier et al. (2021). The input data used for the reconstruction are S2, Landsat-8 and MODIS data. The method requires as input a long time-series of HR images. Hence, we downloaded a total of around 400 scenes for the South Fork catchment and 700 scenes for the Schnals valley from





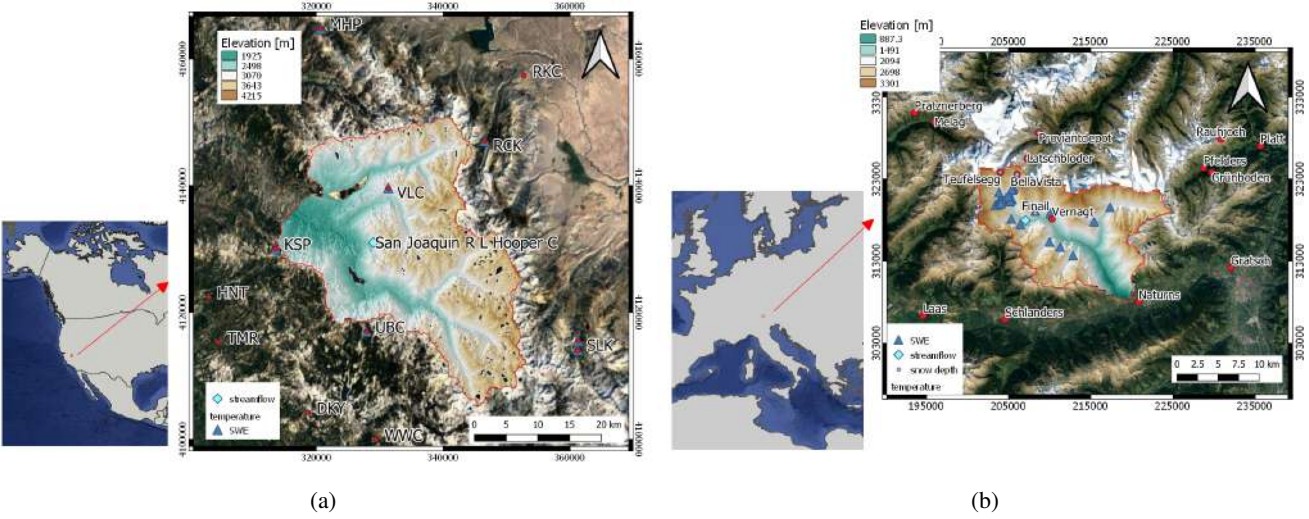

(a)                  (b)

**Figure 4.** Overview of the two test sites: a) South Fork catchment, California, USA, and b) Schnals catchment, South Tyrol, Italy. Background image ©Google Maps 2022.

https://earthexplorer.usgs.gov. The following steps are applied to opportunely pre-processed the data: i) conversion from digital
number to Top of the Atmosphere (ToA) reflectance values, ii) cloud masking through the algorithm s2cloudless available at
https://github.com/sentinel-hub/sentinel2-cloud-detector (Zupanc, 2017), iii) SCF detection through an unsupervised statistical
learning approach (Barella et al., 2022), and iv) binarization of the classification results.

The daily MODIS data are needed for those hydrological seasons in analysis only. The ready-to-use MOD10 version 6.1
are distributed by the National Snow and Ice Data Center (see https://nsidc.org/data/MOD10A1) (Hall and Riggs, 2021). The
NDSI values are converted to SCF by using the algorithm proposed by Salomonson and Appel (2004).

The S1 data are downloaded from https://search.asf.alaska.edu/ and pre-processed (i.e., precise orbit application, thermal
noise removal, border noise removal, beta nought calibration, tile assembly, co-registration, multi-temporal filtering, terrain
correction, geo-coding and sigma nought calibration). These steps are performed using SNAP (Sentinel Application Platform)
and some custom tools. Three tracks are available for each test site, i.e. track 64, 137 and 144 for the South Fork catchment and
track 15, 117 and 168 for the Schnals catchment, with a total number of around 480 and 350 downloaded images respectively.
The backscattering is then daily interpolated and a multi-temporal analysis is carried out for the three tracks separately. If at
least one track shows a drop of at least 2 dB in the signal (Nagler and Rott, 2000) w.r.t. a moving average of the 12 previous
days, that day is considered to be in *ablation*.

## 4 Experimental Results

In this section, we present the results obtained for the South Fork and for the Schnals catchment.

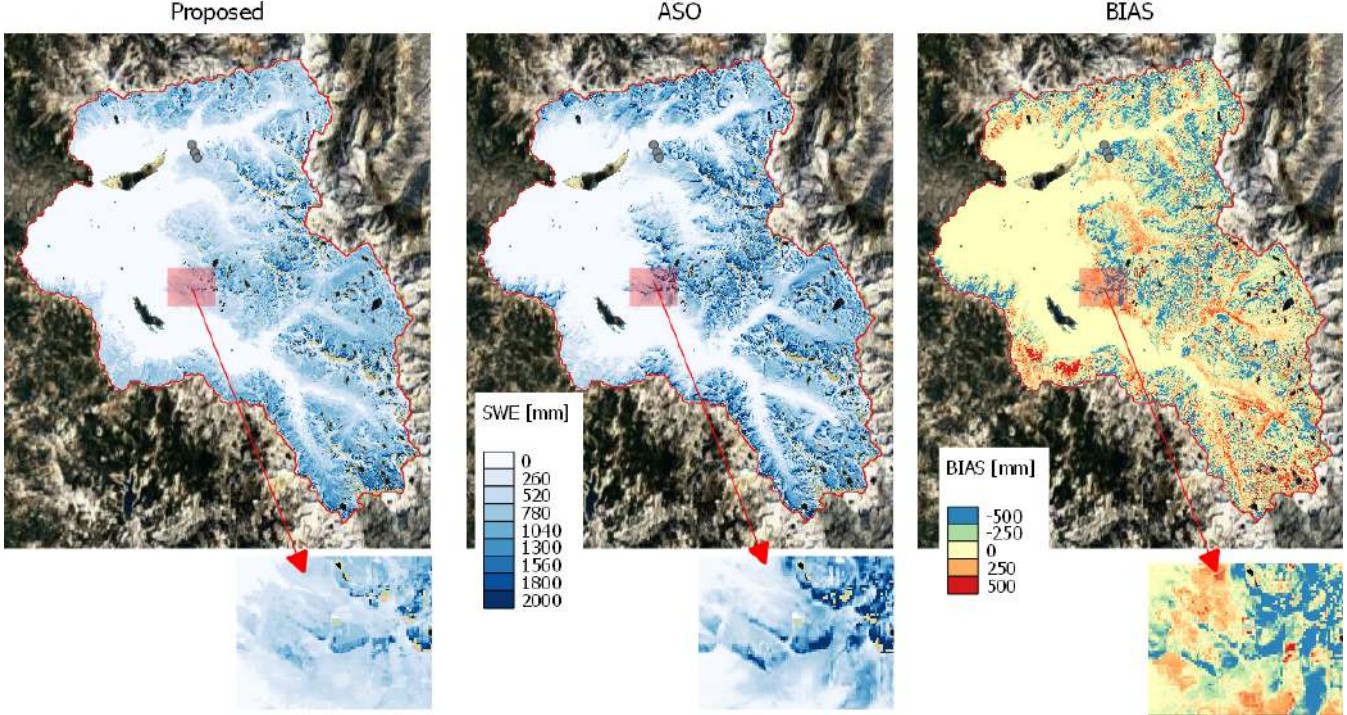

**Figure 5.** Proposed SWE map (on the left), ASO SWE map (in the centre) and bias map calculated as difference between the proposed and ASO SWE (on the right) for the 9th of June 2019. A zoom is shown under the correspondent maps. A transect is shown with three green dots in the North area of the catchment. Background image ©Google Maps 2022.

## 4.1 South Fork catchment

The proposed SWE maps are aggregated at a resolution of 50 m and compared with the corresponding ASO maps, for a total number of 12 dates.

From a qualitative inspection of the results, a general good agreement between the two SWE maps is visible. For sake of
brevity we propose here a detailed analysis of the 9th of June 2019, for which we reported in Fig. 5 the SWE map obtained with the proposed approach, the SWE produced by ASO and the bias map calculated as pixel-wise difference between the proposed and the reference ASO map. In general, it is possible to notice that the proposed method is able to reproduce spatial patterns similar to the ones detected by ASO. This result shows that the use of HR input data achieve an unique spatial detail, which represents one of the main advantages of the proposed method. By looking at a zoomed area, it is possible to better appreciate
this similarity. Nonetheless, it is possible to notice a tendency of the proposed product to underestimate SWE especially in some North exposed areas. This may be due to either i) an error introduced by the DD model, that only considers temperature without accounting for radiation differences linked to different exposures or ii) an error introduced by the reference SD map used in the ASO for defining the amount of snow at the beginning of the season in permanently snow covered areas.



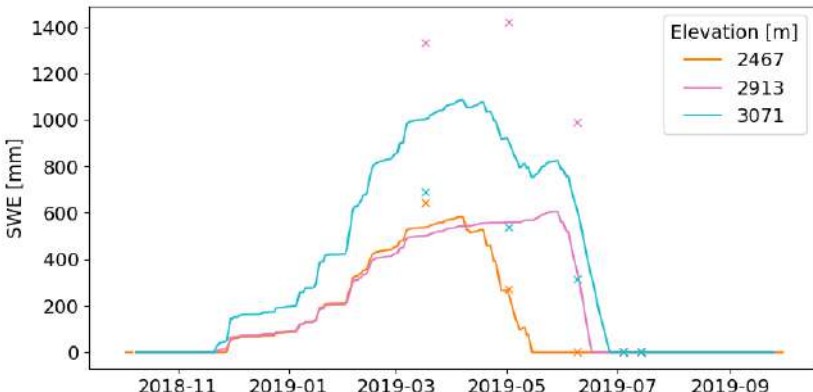

**Figure 6.** SWE temporal trends for: the proposed approach (continuous lines) and ASO (crosses) for the considered transect reported in Fig. 5.

In Fig. 6 the temporal trend of three points for a selected transect is reported. First of all, it is possible to observe that the
proposed product presents an expected behaviour, i.e., longer snow persistence and increasing SWE for higher elevation. On the
other hand, ASO shows higher SWE for the mid elevation point. It is worth noting that even though we use ASO as reference
product, some inconsistencies may be present due to a possible inaccurate estimation of the snow density by the model.

The Volcanic Knob (VLC) monitoring site, which is located inside the analyzed area, is also used for evaluating the obtained
results showing a very good agreement (see Fig. 7). Note that the first year 2018/19 is also used to set up the constant $a$ used
for the DD model, according also to values found in literature (Hock, 2003). The validity of the chosen value is confirmed by
a good agreement of the results for the following two seasons. Interestingly, even though the stations are used to identify the
*accumulation* state the temporal SWE trend does not necessarily present everywhere the same shape as for the station (see in
Fig. 6 and 7). In fact, this final result is influenced also by the persistence of the snow as well by the potential melting, that
varies depending especially by the elevation as it is calculated with a kriging with external drift. Hence, the good agreement
with the station trend is also confirming the validity of the proposed method.

The results of a quantitative global analysis are reported in Table 1. The evaluation shows a generally good correlation
between the two products, i.e., 0.729 on average. The average bias is -40 mm and the average RMSE is 216 mm. The highest
bias and RMSE values are generally encountered in the mid-winter acquisitions. This is due to the fact that the snow cover in
that period of the year is higher, and consequently the SWE, thus generating a potential larger error. Moreover, it is possible
to have an inaccurate detection of the exact location and duration of the snowfalls. This may be due to the fact that the
accumulation in the proposed method is driven by punctual measurements and may be not representative of the entire catchment
as described in Sec. 2.1. For the South Fork area, only one station is inside the catchment and this is not enough to capture the
high variability of precipitations as expected.





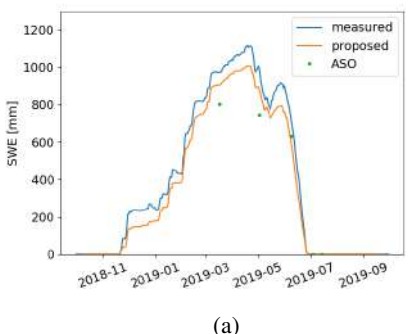
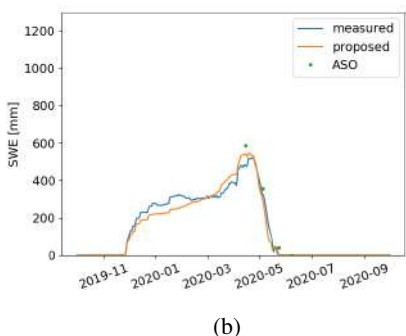
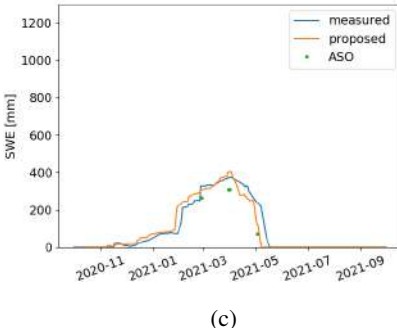

| (a) | (b) | (c) |

**Figure 7.** SWE obtained by the proposed approach (in orange) against the measured SWE (in blue) at the Volcanic Knob test site for the hydrological seasons a) 2018/19, b) 2019/20 and c) 2020/21.

**Table 1.** Results of the comparison between the proposed SWE and ASO products. Bias, RMSE and correlation are calculated pixel-wise. SWE tot is the SWE integrated over space.

| Date | BIAS [mm] | RMSE [mm] | Correlation [-] | SWE tot ASO [Gt] | SWE tot proposed [Gt] |
|---|---|---|---|---|---|
| 17/03/2019 | -88 | 311 | 0.73 | 734 | 653 |
| 02/05/2019 | -66 | 308 | 0.84 | 640 | 580 |
| 09/06/2019 | -71 | 299 | 0.86 | 482 | 417 |
| 04/07/2019 | -59 | 197 | 0.85 | 167 | 113 |
| 14/07/2019 | -51 | 166 | 0.80 | 100 | 53 |
| 15/04/2020 | -70 | 239 | 0.64 | 367 | 303 |
| 05/05/2020 | -65 | 225 | 0.66 | 235 | 165 |
| 23/05/2020 | -95 | 234 | 0.65 | 156 | 69 |
| 08/06/2020 | -20 | 152 | 0.66 | 44 | 26 |
| 26/02/2021 | -8 | 130 | 0.65 | 175 | 167 |
| 31/03/2021 | 59 | 168 | 0.70 | 180 | 235 |
| 03/05/2021 | 49 | 166 | 0.70 | 67 | 123 |

In Fig. 8, the temporal trend of the total SWE for the three considered hydrological seasons is shown. We notice a general
good agreement between the total amount of SWE estimated through the two approaches. The plot also gives an idea of the large differences that can be encountered for different seasons and that are well captured by the proposed method. The first hydrological season 2018/2019 shows the highest amount of SWE, while the others are drier. The tendency is of a slight underestimation of SWE for the proposed method w.r.t. ASO results for the first two seasons, while the last is overestimated as also shown in Table 1. The possible reasons will be discussed in Sec. 5.

A more detailed analysis is presented in Fig. 9, where we show the trend of the maximum of SWE versus elevation, slope and aspect. The results show an increasing trend of SWE with elevation that is inverted for highest quotes, since these usually





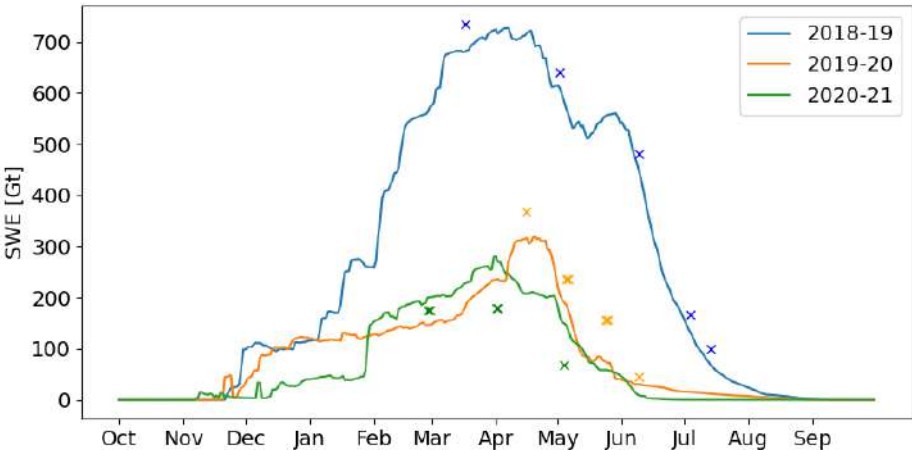

**Figure 8.** Temporal trends of the total SWE for the South Fork catchment over the three analyzed seasons. Crosses represent the ASO reference.

present very steep slopes and consequently a marked tendency of snow to be subjected to gravitational transport. This is also confirmed by the second graph, where steeper slopes present less SWE. Another coherent result is also the bigger amount of SWE for North exposed areas. These results confirm that the method is suitable to be exploited for hydrological applications.

For more results derived by the comparison between the proposed method and the ASO reference product we refer the reader to the Appendix C (see Fig. C1). Here we report a detailed analysis on the total SWE for all the available ASO SWE information considering different elevation, aspect and slope belts (see Fig. C2 and C3).

**4.2   Schnals catchment**

For the Schnals valley, reference spatialized data of SWE are not available. However, manually collected SWE measurements
for the hydrological season 2020/21 were collected also along spatial transects. For more details about the measurement location, please refer to the Appendix A, Fig. A1. The results show a bias of 38 mm and a RMSE of 209 mm, indicating a general good agreement. For sake of brevity, we report in Fig. 10 only few examples of reconstructed SWE evaluated against the manual measurements. It is possible to notice, in accordance with the results obtained for the South Fork catchment, that the variability from an area to another of the catchment is properly caught by the proposed method. This is due to the different
persistence of snow on ground that modulates the spatial variability of the potential melting estimated by the DD model. It is possible to notice in the figure that the SWE tends increases with the elevation.

In Fig. 11 we report the map of the SWE maximum for the two analyzed hydrological seasons. It is possible to see that the season 2020/21 is characterized by a higher amount of SWE. However, the two years show similar patterns that are coherent with the morphology of the study area. In detail, we can notice that interestingly there is a higher amount of SWE especially in



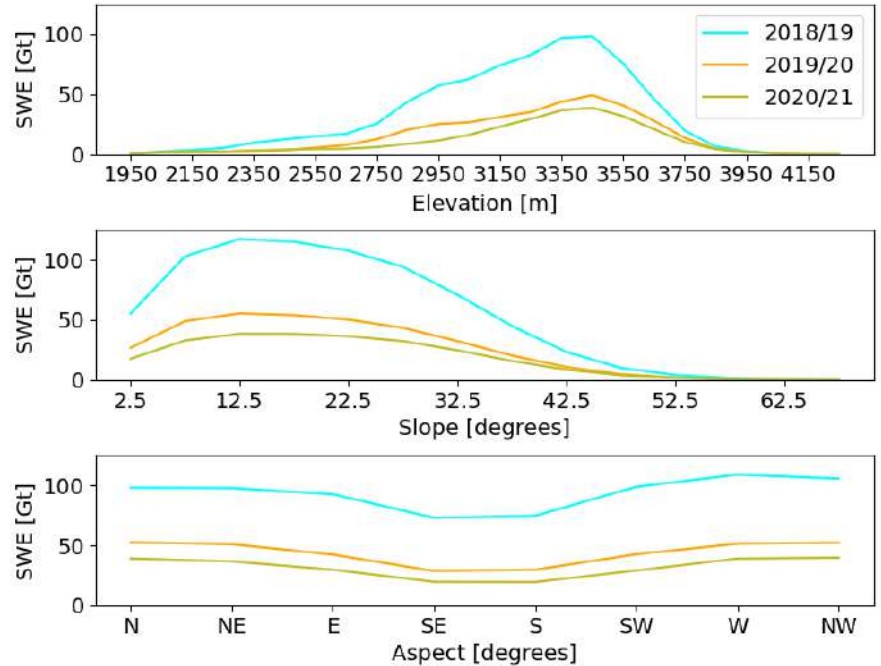

**Figure 9.** Trends of the maximum of SWE for the hydrological seasons 2018-2021 w.r.t. the elevation (up), slope (centre) and aspect (low) for the South Fork catchment.

the East part of the catchment that corresponds to the glacierized area of the Roteck/Monte Rosso mountain. We found a longer persitence of snow for these North exposed slopes and consequently a larger amount of reconstructed SWE. The consistency of the SWE patterns is also confirmed by Fig. 12, from which similar considerations as for the South Fork catchment can be carried out. Another qualitative indicator of the goodness of the results is given in Fig. 13. The trend of the SWE maximum over time is represented together with the discharge measured at Schnalserbach - Gerstgras. In the second year, higher SWE

amounts correspond to a higher peak in terms of measured discharge as expected. Moreover, it is possible to appreciate for the two seasons an increase of discharge that happens in correspondence of the SWE decrease.

## 5   Discussion

We have presented in the previous section the quantitative and qualitative results for the two study areas. Notwithstanding the good agreement with the observed measures as well as with the reference ASO product, we will draw in this section a critical

analysis of the proposed approach. In detail, each main step, i.e., catchment state identification, SCA regularization and SWE reconstruction, is analysed in the following paragraphs, focusing on the main sources of errors.



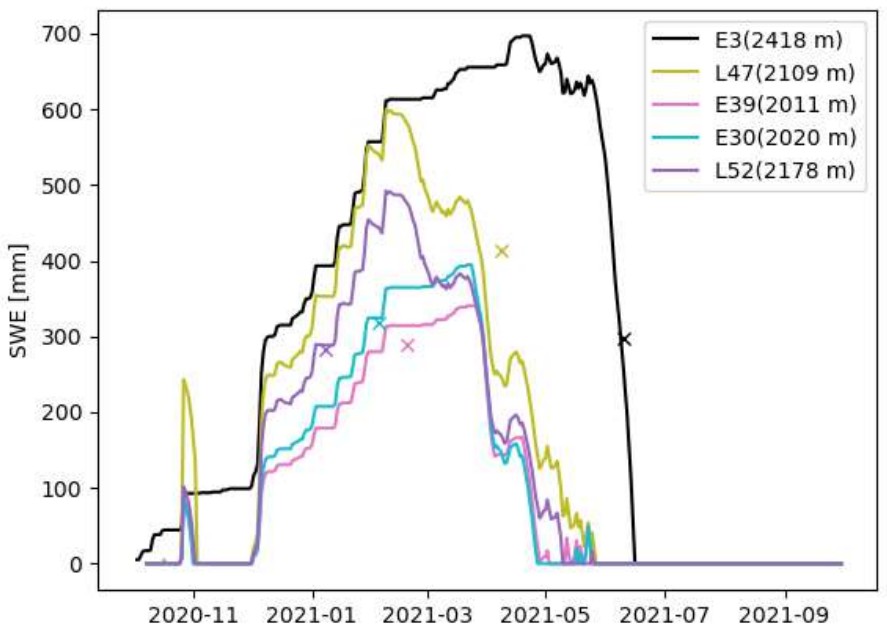

**Figure 10.** SWE proposed (continuous line) against the manual measured SWE (crosses) at some locations in the Schnals catchment (see Fig. A1).

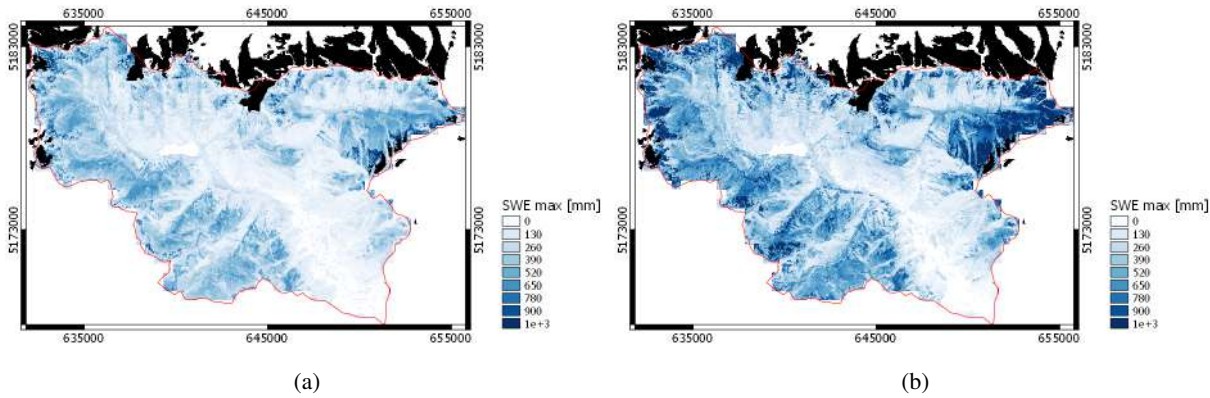

**Figure 11.** Maximum of SWE for the Schnals catchment for the hydrological seasons a) 2019/20 and b) 2020/21. Black areas represent glaciers that are masked out ©Randolph Glacier Inventory (RGI 6.0).

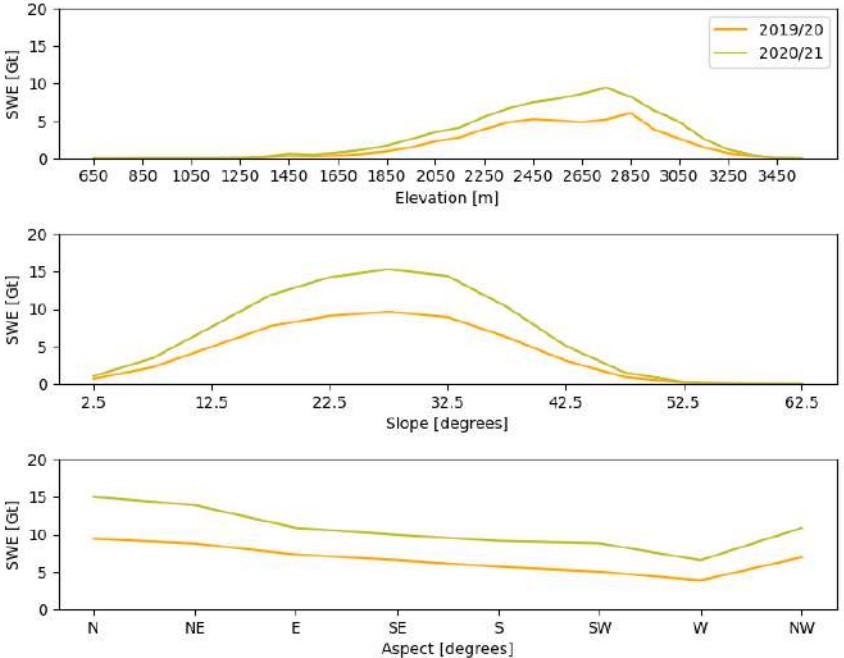

**Figure 12.** Trends of the maximum of SWE for the hydrological seasons 2019-2021 w.r.t. the elevation (up), slope (centre) and aspect (low) for the Schnals catchment.

## 5.1 Error in the determination of the catchment state

In Sec. 2.1 we have described how to define the *accumulation* state starting from in-situ SWE, SD or precipitation measurements. We suggested that a network of AWS is needed to cope with the possible heterogeneity of the snowfalls inside the

catchment. However, we would like to discuss more in detail the topology of such a network. In Sec. 2.1 we assumed that the *accumulation* state refers to all the catchment covered by snow without distinguishing between areas that are interested by a snowfall and areas that are in *equilibrium* or *ablation*, e.g., this is the case of contemporary rain-on-snow and snowfall inside the catchment. We introduced this simplifying hypothesis since it is difficult to precisely determine where the snowfall occurs with a HR detail. By strictly taking this assumption, and therefore exploiting only one in-situ measurement, the results

are still affected by wind and gravitational redistribution that influence the AWS observations. This in turn may affect the final SWE results that are biased by wrong state identification. Moreover, in some remote areas automatic measurements can be completely missing. Hence, there is the need to identify the *accumulation* state at HR as done for the *ablation* state using the SAR information. A possible solution is to spatialize the AWS information considering for example elevation bands where the hypothesis of a constant state is more reliable (i.e., a similar concept as the hydrological response unit). Nonetheless, this



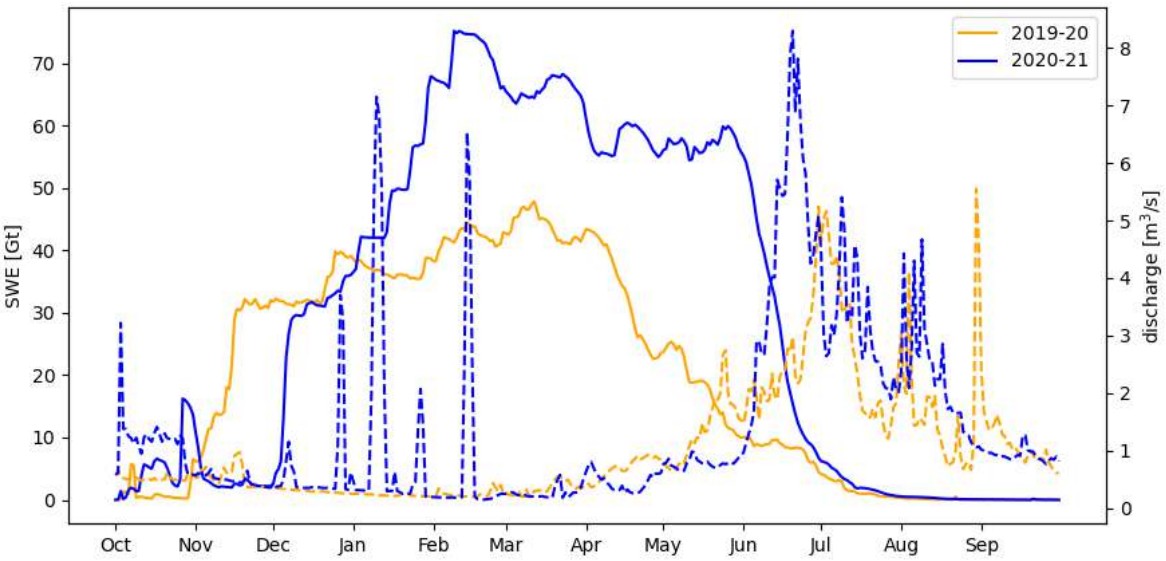

**Figure 13.** SWE for the Schnals catchment for the hydrological seasons 2019-2021. Continuous lines represent SWE, dashed lines represent the discharge measured at Schnalserbach - Gerstgras.

is a challenging task that requires a well distributed AWS network or to couple our approach with a physically based model able to spatialize correctly the AWS information. Even if the modularity of the proposed approach allows the separation of the different steps that can be easily interchanged with other possible solutions, this is is out of the scope of this paper.

  Another possible solution is to exploit once again the information provide by the satellite missions. Indeed, the use of satellite information collected over larger and remote areas represents an interesting enhancement of the proposed method.

The prediction of the catchment state could be provided for example by satellite information about SCA variations that are connected to SWE changes. Especially at the beginning of the season, an important snowfall is well represented by a strong increment in terms of SCA. However, there are some ambiguities that need to be solved. For example, when observing constant SCA or when SCA is 100% we cannot determine the state of the catchment. In this sense, the size of the catchment plays an important role that need to be further investigated. As interesting alternative we also mention the use of SAR data. In this work,

we exploited S1 to detect the snowpack melting. However, the signal seems to be also sensitive to the presence of fresh snow, showing an increase of the backscattering in correspondence of a snowfall (e.g., Lievens et al., 2022; Tsang et al., 2021). The poor temporal resolution represents however a strong limitation for a practical application.



## 5.2 Error in the SCA derivation and regularization

It is worth discussing here the SCA regularization more in detail. In Fig. 14 and 15 we show the SCA before and after the reg-
ularization for the South Fork and Schnals catchments, respectively. As explained in Sec. 5.2, due to the snow cover detection
algorithms employed in the proposed method the raw reconstructed SCA presents: i) strong decreasing peaks in correspon-
dence of the HR acquisitions at the beginning of the season that indicate an underestimation of SCA in forested areas, and
ii) small increasing peaks in correspondence of the HR acquisitions in the late melting phase that indicate the presence of
snow patches that are missed by LR sensors. The regularized SCA is instead more stable and the spurious oscillations, present
especially during the most cold winter period, are corrected. The effectiveness of the correction is also visible by looking at a
corrected image. Fig. 16 represents a common situation when foggy and mixed pixels are classified erroneously. The proposed
correction improves the snow detection especially in these complicate cases.

An evident case where an overestimation of the SCA is introduced is in the season 2020/21 in May/June for the South Fork
catchment (see Fig. 14c). This error is due to the fact that the AWS do not indicate an *accumulation* in correspondence of the
peaks that happen in the late melting phase. Hence, the label is corrected according to the majority rule in *ablation* in the case
of an old snowfall (see Sec. 2.2 and Algorithm 1). Many pixels are considered as TP, leading to the propagation of the *snow*
pixels backward and the consequent overestimation of SCA. This in turn leads to an overestimation of the SWE as shown in
the results (see Sec. 4.1). On the other hand, an underestimation of the SCA is introduced for example in May 2019 for the
South Fork catchment (see Fig. 14a). By an accurate inspection of the conditions that lead to the flattening of the SCA during
the late snowfall, it has been showed that the stations indicate a long period as *accumulation*, i.e., from 16th until the 29th of
May, while the peak starts decreasing in the original SCA time-series starting from the 21st of May. According to the majority
rule for a recent snowfall, the pixels are marked as FP. In fact, the most frequent label is *snow-free* since they are *snow-free* for
sure from $t$ onward and this implies the replacement with *snow* backward until the day of last ablation. It is possible that the
AWS present some sensor errors, but this could also be the case of a mixed state inside the catchment. In other words, the AWS
reveal a snowfall but this is most likely happening at high elevations, while the SCA is decreasing due to an ongoing melting
especially in the lower elevation belts of the catchment (SCA from $\sim 100\%$ to $\sim 80\%$ means that low quotes are getting snow-
free). It may be possible that at low quotes it rained while at higher quotes the AWS correctly detect a snowfall. However, we
expect that such an ephemeral snowfall is not affecting the total amount of SWE, as we have seen in the Sec. 4.1. As future
perspective, it is worth noting that the availability of more HR images (e.g., Landsat-9) will help to mitigate the errors in the
daily SCA reconstruction.

## 5.3 Error in the SWE reconstruction

In Sec. 4 we have presented the results of the proposed method. We have shown the differences that arise when comparing
the results of our method with a spatialized product (ASO). We would like to discuss here the sources of error and the major
weaknesses of the proposed method. The potential source of errors are represented by i) the temperature data, that in turn affects





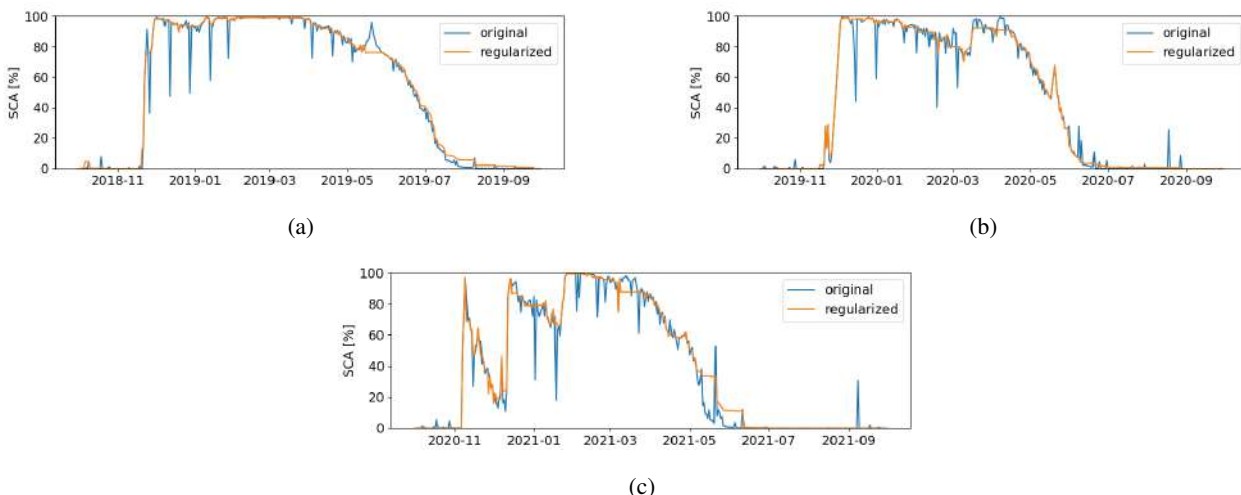

**Figure 14.** Trends of the SCA in the South Fork catchment for the hydrological seasons a) 2018/19, b) 2019/20 and c) 2020/21. Original input version (in blue) and corrected version (in orange) after the application of the proposed regularization.

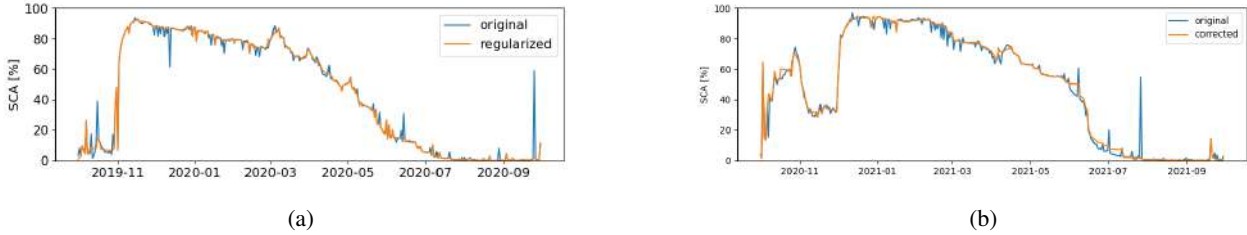

**Figure 15.** Trends of the SCA in the Schnals catchment for the hydrological seasons a) 2019/20 and b) 2020/21. Original input version (in blue) and corrected version (in orange) after the application of the proposed post-filter.

the DD model, and ii) the wrong identification of $t_{SA}$ and $t_{SD}$, that depends on the accuracy of both the SCA time-series and the catchment state. Since the input data are subjected to several degrees of preprocessing, it is difficult to carry out a specific sensitivity analysis of the problem. The most important factors are discussed in the work of Slater et al. (2013b). Instead, we perform here a critical analysis for better defining future developments to improve the proposed method.

The implications of a wrong detection of $t_{SD}$ are much more important than the wrong identification of $t_{SA}$. At the end of the ablation period, $M$ is high and therefore the total SWE for a pixel, which is given by the potential melting integrated over time for days in ablation, can be strongly altered by introducing either an overestimation or an underestimation, depending on the case. The error can also be propagated for several days during the melting season. On the contrary, in the period of the year when most consistent snowfalls happen at the beginning of the season, the days in *ablation* are few, temperatures are low,

consequently $M$ is negligible (see Eq. 1) and does not change so much from one day to the other.





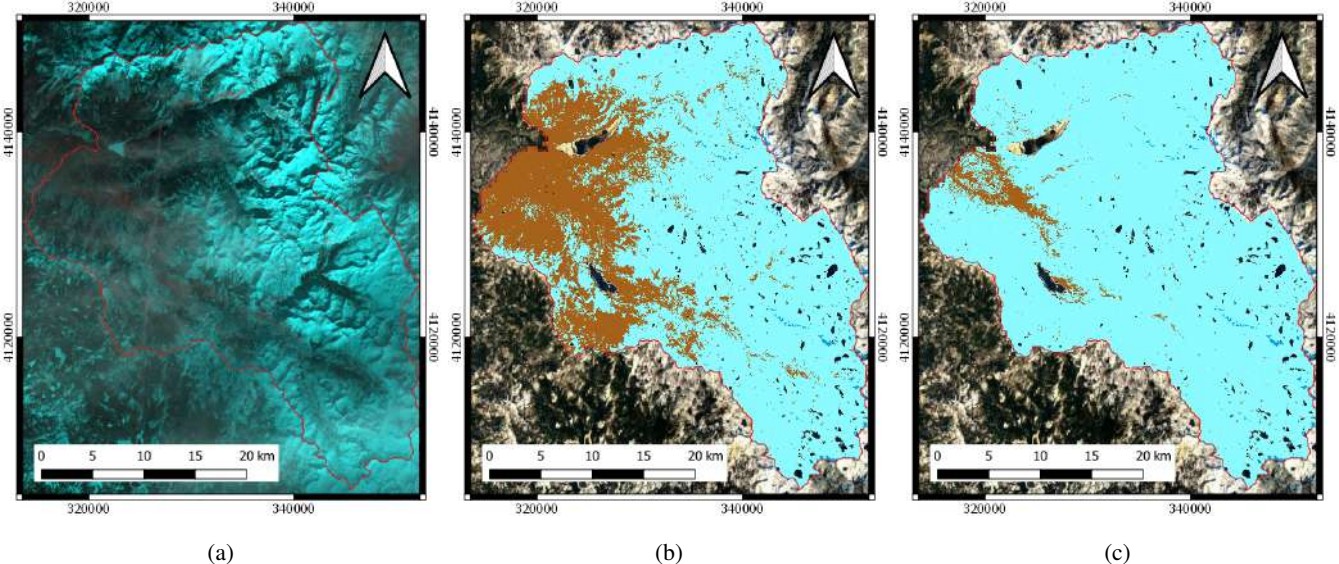

(a)            (b)            (c)

**Figure 16.** Example of a regularized snow map on the date 12/01/2020 for the South Fork catchment: a) false-color composition (R:SWIR,G:NIR,B:RED), b) input snow cover map, and c) regularized snow cover map. Background image ©Google Maps 2022.

The use of a simple DD model for computing the potential melting is another source of error. This simple model does not take into account different energy inputs due to either geomorphometrical features or different periods of the year, since we used a constant DD factor. This may be the case of the overestimation of SWE happening during the hydrological season 2020/21 as shown in Sec. 4.1. In fact, this year presents an anticipated melting. Hence, we expect a lower energy input w.r.t.

the season when the DD factor has been calibrated. Even though we are aware of the limitations of such a simple model, we did not consider the use of a more complex model, since this is not the focus of this work. An interesting further development is to test the approach with a more sophisticated potential melting estimation. Since the potential melting is calculated separately, this step can be easily replaced.

As potential correction of the DD model, we proposed the use of S1 for identifying the timing of the melting phase. In fact, it

may be that the DD detects high temperatures but the energy is still not sufficient to cause the snowpack melting. Even though we do not expect big differences since temperatures are low in the first phase of the melting and consequently the potential melting is low, the spatialization of temperature may also be affected by possible errors. S1 represents a spatialized way to identify the melting, but it presents as major disadvantage a poor temporal resolution, i.e., few days. As future development, we aim at exploiting S1 and the new HR land surface temperature acquired by the next Sentinel generation as a proxy of the

potential melting estimation, thus enlarging the applicability of the proposed method in remote areas.

Finally it is worth mentioning that, the uncertainty introduced in the method are also linked to the high spatial heterogeneity of the SWE. It is possible to encounter a very large variability also within a pixel with size of 25 m as shown by the SWE





measurments acquired in Schnalstal by Warscher et al. (2021). This results in an intrinsic difficulty to evaluate the output with an appropriate reference data.

## 6 Conclusions

In this work we presented a novel approach to reconstruct daily HR SWE maps for a mountainous catchment. We started by determining the state of the catchment, i.e. if it is subjected to *accumulation* (increase of SWE), *ablation* (decrease of SWE) or *equilibrium* (constant SWE). The state was identified exploiting both i) in-situ SWE or SD data, which provide information about the snowfall; and ii) multi-temporal SAR information to decide if the pixel is melting. Moreover, a novel daily HR SCA time-series was used to determine the dates of snow appearance and disappearance, which represents an information with unprecedented spatial and temporal detail. The SCA time-series was opportunely regularized according to the catchment state. Furthermore, the state was also used as necessary information for the SWE reconstruction, i.e. whether SWE is added or removed. The potential melting was estimated by mean of a simple DD model after having spatialized in-situ temperature observations. The SWE was reconstructed for the entire snow season without the need of precipitation data as input. The results obtained for two different test sites, i.e., the South Fork catchment and the Schnals catchment demonstrated the effectiveness of the proposed approach to estimate the HR SWE. For the first catchment, the results were evaluated against the ASO SWE product at 50 m, showing an average bias of -40 mm. For the second site, the results were evaluated against manual measurements showing a bias of 38 mm. The obtained results were extensively discussed also considering possible hydrological applications of such a product. In this sense, we have seen that the results are very promising, since they i) are able to well capture the typical spatial variability of a HR product, ii) show spatial patterns that are consistent with the reference product, as well as with the geomorphology of the study area, iii) provide a reliable global balance at a catchment scale, and iv) reproduce the variability of different hydrological seasons.

Finally, we can state that the use of HR SCA for SWE reconstruction is more adequate to sample the variation of SWE due to the complex topography of mountainous catchments. Some technological limitations are present (e.g., necessity of merging LR and HR sensors given the absence of daily optical HR acquisitions, scarce temporal resolution of SAR acquisitions). Even though the proposed approach tries to overcome all these limitations, we expect that further improvements will be introduced also by future satellite missions. This will open opportunities not only to improve the proposed method but also to obtain a near-real time predictor of SWE for large hydrological and ecological applications.

*Code and data availability.* The implemented code can be made available upon request to the authors. All the raw input data are freely available as indicated in Sec. 3. The output dataset related to this article will be made available in an open repository after publication or asked directly by the authors.





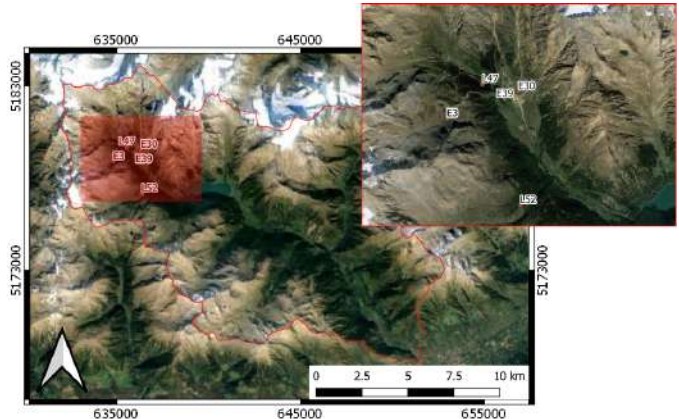

**Figure A1.** Overview of the measurements. Background image ©Google Maps 2022.

## Appendix A: In-situ snow measurements

The state characterization is done by observing available SWE or SD in-situ measurements. For the South Fork catchment, SWE continuous measurements are available for 7 different sites. As shown in Fig. 4a, only one station is inside the considered area, i.e. the Volcanic Knob station (VLC). However, we considered also the stations in the surrounding area to smooth out possible sensor errors or redistribution effects.

In the Schnals catchment, SWE continuous records are available only for the location Bella Vista. For this reason, the decision on the catchment state is based on the SD records available for 5 stations (see Fig. 4b). Three stations are inside the study area and the remaining ones are very close to the study area.

Note that for the same catchment, all the stations show a coherent pattern, meaning that we can consider the catchment subjected to the same major weather forcings.

In Fig. A1 we also report the location of some of the manual SWE observations collected internally by Eurac Research (Institute for Earth Observation) for the Schnals catchment. These few points have been used in Fig. 10 to show some examples of reconstructed SWE trend and its behaviour against the manual measurements. For an overview of the location of all the collected manual observations used to compute the performances of the approach, see Fig. 4b.

## Appendix B: Degree day estimation

The degree day (DD) is estimated starting from the available in-situ temperature observations for both catchments. Regarding the South Fork catchment, the temperature is available in 11 different stations located within a radius of around 15 km from the study area (see Fig. 4a). Due to the presence of some gaps in the data, we excluded from the computation of the DD the station "RKC" for season 2018/19, "WWC" for 2019/20 and "DKY","VLC", and "UBC" for 2020/21. For the Schnals catchment, the temperature is available in 15 different locations within a maximum distance of around 10 km (see Fig. 4b). Once the DD is





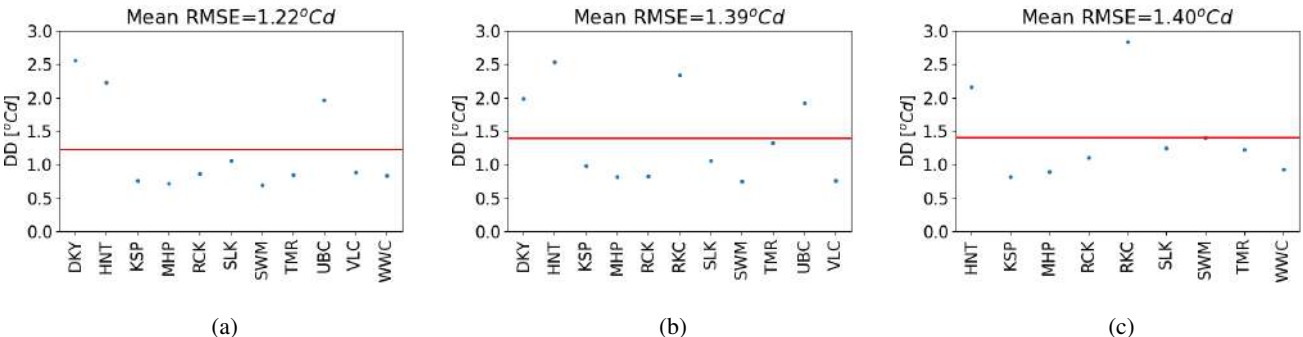

**Figure B1.** Leave-one-out cross validation results for the South Fork catchment for the hydrological seasons a) 2018/19, b) 2019/20 and c) 2020/21.

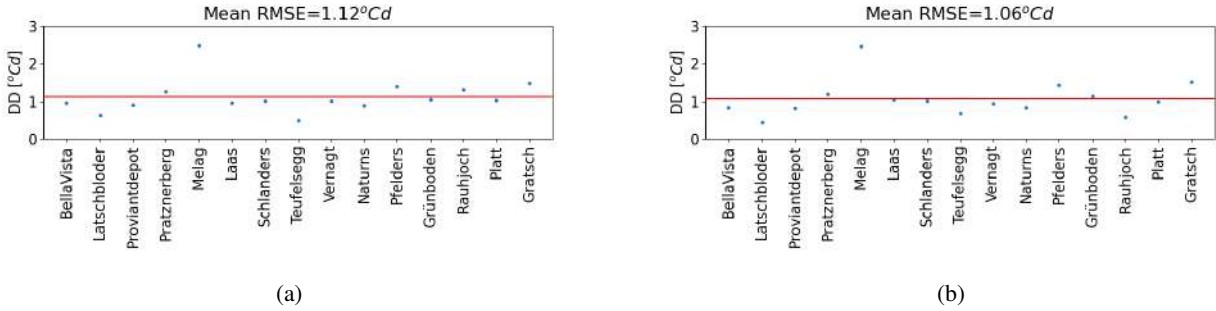

**Figure B2.** Leave-one-out cross validation results for the Schnals catchment for the hydrological seasons a) 2019/20 and b) 2020/21).

computed for each station through Eq. 2, the DD is spatially interpolated with the kriging routine as explained in Sec. 2.3. The goodness of the spatial interpolation is tested through a leave one out (LOO) cross validation. The results in terms of root mean square error (RMSE) are reported in Fig. B1 for South Fork and in Fig. B2 for Schnals, showing a mean RMSE that never

555 exceeds $1.5^{o}Cd$.

## Appendix C: SWE results for the South Fork catchment

Fig. C1 shows the SWE maps derived with the proposed method, the ASO reference product and the bias calculated as difference between the proposed and the reference maps.

Fig. C2 shows the total amount of SWE calculated for different elevation, slope and aspect bands for the 12 dates when also

560 the ASO product is available. We can notice a general good agreement showing that the proposed method is able to represent the typical geomorphological variability of the snow processes.

Fig. C3 shows the boxplots relative to the total amount of SWE calculated for different elevation, slope and aspect bands for the 12 dates when also the ASO product is available. We remind that a boxplot represents the Interquantile Range (IQR) and is





(a) 17th March 2019.

(b) 2nd May 2019.

(c) 9th June 2019.

**Figure C1.** Proposed SWE (left), ASO SWE (centre) and bias (right) for the 12 analyzed dates over the three hydrological season (2018-2021) for the South Fork catchment. Background image ©Google Maps 2022.





(d) 4th July 2019.

(e) 14th July 2019.

(f) 15th April 2020.

**Figure C1.** Proposed SWE (left), ASO SWE (centre) and bias (right) for the 12 analyzed dates over the three hydrological season (2018-2021) for the South Fork catchment. Background image ©Google Maps 2022. (cont.)



(g) 5th May 2020.

(h) 23th May 2020.

(i) 8th June 2020.

**Figure C1.** Proposed SWE (left), ASO SWE (centre) and bias (right) for the 12 analyzed dates over the three hydrological season (2018-2021) for the South Fork catchment. Background image ©Google Maps 2022. (cont.)



(j) 26th February 2021.

(k) 31st March 2021.

(l) 3rd May 2021.

**Figure C1.** Proposed SWE (left), ASO SWE (centre) and bias (right) for the 12 analyzed dates over the three hydrological season (2018-2021) for the South Fork catchment. Background image ©Google Maps 2022. (cont.)



(a) 17th March 2019.

(b) 2nd May 2019.

(c) 9th June 2019.

(d) 4th July 2019.

(e) 14th July 2019.

(f) 15th April 2020.

**Figure C2.** Total SWE [Gt] distributed for each elevation, slope and aspect belt. The proposed product (in yellow) is evaluated against ASO (in blue) for the 12 analyzed dates over the three hydrological season (2018-2021) for the South Fork catchment.





(g) 5th May 2020.

(h) 23th May 2020.

(i) 8th June 2020.

(j) 26th February 2021.

(k) 31st March 2021.

(l) 3rd May 2021.

**Figure C2.** Total SWE [Gt] distributed for each elevation, slope and aspect belt. The proposed product (in yellow) is evaluated against ASO (in blue) for the 12 analyzed dates over the three hydrological season (2018-2021) for the South Fork catchment (cont.).



composed by the median (orange line), the first quartile Q1 (or 25th percentile) and the third quartile Q3 (or 75th percentile). The whiskers show the "minimum" (Q1-1.5*IQR) and the "maximum" (Q3 +1.5*IQR). In this representation we omit the outliers.

*Author contributions.* VP and MC designed the research; VP carried out the experiments and processing; RB provided the HR snow maps from S2; all the authors contributed to the analysis and interpretation of the results; VP wrote the paper based on inputs and feedbacks from all coauthors.

*Competing interests.* The authors declare that they have no conflict of interest.

*Acknowledgements.* This work was supported by the Swiss National Science Foundation (SNF) project "Snowtinel: Sentinel-1 SAR assisted catchment hydrology: toward an improved snow-melt dynamics for alpine regions" Contract No. 200021L205190.
We also would like to thank the NASA Airborne Snow Observatory (ASO) for providing free-of-charge data.

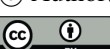



(a) 17th March 2019.

(b) 2nd May 2019.

(c) 9th June 2019.

(d) 4th July 2019.

(e) 14th July 2019.

(f) 15th April 2020.

**Figure C3.** Boxplots of the bias calculated as the SWE generated by the proposed approach minus the ASO product for each elevation, slope and aspect belt. The results are represented for the 12 analyzed dates over the three hydrological season (2018-2021) for the South Fork catchment.



(g) 5th May 2020.

(h) 23th May 2020.

(i) 8th June 2020.

(j) 26th February 2021.

(k) 31st March 2021.

(l) 3rd May 2021.

**Figure C3.** Boxplots of the bias calculated as the SWE generated by the proposed approach minus the ASO product for each elevation, slope and aspect belt. The results are represented for the 12 analyzed dates over the three hydrological season (2018-2021) for the South Fork catchment (cont.).



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
