# Peer review of "Exploring the Use of Multi-source High-Resolution Satellite Data for Snow Water Equivalent Reconstruction over Mountainous Catchments"

_The Cryosphere, 2022_

## Referee Comment (RC2)

[referee-annotated manuscript omitted]

---

## Community Comment (CC1)

**Community comment on tc-2022-146**

**Major comments**

**Missing information on the SAR data**

Using the Sentinel-1 SAR imagery to correct the daily snow cover maps is a very creative and effective solution, and probably one of the main contributions of this paper. However, even though most of the methodology is based on the SAR-derived catchment state and it is practically treated as ground truth, there is a lack of background information and uncertainty estimation:

- It is not mentioned how forests are dealt with in the identification of the catchment state. Marin et al. (2020), from which this part of the methodology is adapted, mention that "the response to the wet snow becomes more complex in case of the snowpack in forest", but leave this out of the scope of their study. Therefore, if forests are included in this study (which I assume from the elevation ranges of the catchments), it should be elaborated how effective the methodology is in forested areas. It would also be useful to know what percentage of the catchments is covered by forests.
- The temporal and spatial resolution of the SAR imagery is never explicitly mentioned.
- The uncertainty as quantified by Marin et al. (2020) is not mentioned either. They estimate the RMSE of the start of the moistening phase to be 6 days. An error of 6 days in the ablation identification could lead to large differences in the SWE reconstruction (equation 3), and should therefore be acknowledged or taken into account.
- In L186 and L342 it is mentioned that ablation only occurs with a drop of the SAR backscattering signal, but during the runoff phase the backscattering signal is increasing (Marin et al., 2020). I assume that this increase should also be included in the definition of the ablation days.
- Sentinel-1 is only mentioned as S1 and never fully spelled out.

**SWE loss during moistening and ripening phase**

In the methodology the assumption is made that during an ablation day with a non-zero DD there will always be a loss of SWE. However, if I understood it correctly these ablation days include all three phases of snow melt (moistening, ripening and runoff), even though during the moistening and ripening phases there is rarely any SWE loss, as also shown by Marin et al. (2020). It would perhaps be more physically accurate to limit the SWE loss to the runoff phase. This would not change the total amount of calculated melt energy and therefore accumulation, but it would possibly change the peak SWE estimate and likely change the timing of the SWE loss. In figure 3 for instance, if the first ablation phase would consist only of moistening or ripening there would be no loss of SWE and the peak SWE would be higher after the second accumulation phase. Moreover, if the runoff is limited to the runoff phase, the SWE loss would be delayed and more concentrated. This would have significant hydrological implications.

Since the classification between the three snow melt phases can easily be determined from the SAR imagery, it would seem that adding this information to the SWE reconstruction would not lead to a large increase in complexity of the methodology. It would, on the other hand, possibly lead to an increased physical basis of the methodology, especially given the known issues around a constant DD factor (Magnusson et al., 2015). I'd be interested to know your thoughts on this.

**Figure C3 and C2 are more informative than figure 9**

Showing how the modeled SWE behaves as a function of the elevation, slope and aspect is indeed very insightful, but only when compared to the behavior of the observed SWE. Therefore, I find figure C2 more informative than figure 9, even if ideally the maximum SWE would indeed be the best moment for comparison. However, figure C2 does not remove the bias and therefore does not allow for a relative comparison between the model and the observations per terrain parameter. Figure C3 does this, but the high number of figures does not allow for an easy general comparison. A compressed version of figure C3, for example with averaged biases, could potentially be more informative and would be easier to include in the main text as well. Other than that, I believe figure C2 would be clearer with lines instead of bars, and figure 12 could be left out of the manuscript.

**Figure 13**

The comparison between the modeled SWE and the discharge is very insightful, but the accompanying explanation is lacking in depth. Indeed the peak SWE correctly matches the peak discharge among the two years, but the timing in both years shows very different behavior. In 2020-2021 the response is much more direct, while in 2019-2020 it's more delayed. Is this because the soil in 2020-2021 is already saturated after the rainfall events in Jan-March? And which part of the discharge after July originated from snow melt and which part from rainfall? On a side note, it would perhaps be insightful to show the rainfall rates on the inverted y-axis, and have the SWE loss and discharge on the same y-axis with the same units (e.g. m3/day).

**Minor comments**

- The resolution of the figures is often not high enough to be able to distinguish important details, especially in the spatial plots. Unless this is a result of the compiling of the preprint, increasing the resolution or saving the figures in vector format (for the graphs) would benefit the manuscript.
- The legend nor the caption in figure 4 explain what the red points stand for (which I assume to be temperature stations).
- Line 289 ("Note that the number of days in accumulation varies for each pixel and consequently the coefficient is function of time and space.") contradicts L104 ("According to the state, that is assumed to be homogeneous for all the pixels of the catchment,… ").
- Even if it's clear from the text, the caption of table 1 should perhaps mention of which catchment these results are
- L372: "The highest bias and RMSE values are generally encountered in the mid-winter acquisitions." As I see it, this is not reflected by the RMSE and bias values we see in table 1, unless mid-winter means march-may.
- In the author contributions, MC should be CM
- L433 I'm guessing that you're talking about MODIS, but I believe it would be good for clarity if you mentioned this
- L464 "replacement with snow", I assume this should be "replacement with snow-free".
- Figure 14 and 15 are rather small, and I would perhaps have appreciated to see the accumulation and ablation phases reflected in the background of the plots

Magnusson, J., Wever, N., Essery, R., Helbig, N., Winstral, A., and Jonas, T.: Evaluating snow models with varying process representations for hydrological applications, Water Resour Res, 51, 2707–2723, https://doi.org/10.1002/2014wr016498, 2015.

Marin, C., Bertoldi, G., Premier, V., Callegari, M., Brida, C., Hürkamp, K., Tschiersch, J., Zebisch, M., and Notarnicola, C.: Use of Sentinel-1 radar observations to evaluate snowmelt dynamics in alpine regions, Cryosphere, 14, 935–956, https://doi.org/10.5194/tc-14-935-2020, 2020.

---

## Author Response (AR1)

Dear Editor,

We would like to express our gratitude to you and the reviewers for providing us with meaningful and constructive comments on our manuscript. We have considered your feedback and made major revisions to the manuscript.

Firstly, we have rewritten the methodological section to provide a clearer definition of the state at the pixel level, which makes it more applicable and easier to understand. We have also discussed the limitations of defining the state as homogeneous for the entire catchment during accumulation, and the advantages of using spatialized information such as Sentinel 1 for determining the ablation state. We have included two separate paragraphs that provide a detailed description of accumulation and ablation retrieval. We have also used this new structure to better clarify the use of Sentinel-1 in our work.

In addition, we have thoroughly revised the results section, taking into account all of the feedback and suggestions provided. We have added an analysis of an additional dataset, the Western United States daily snow reanalysis, which has a spatial resolution of 500 m. This further validates the good performance of the proposed approach.

Furthermore, the discussion section has been deeply revised and restructured by inserting a new subsection where we present and discuss the results of a sensitivity analysis. This analysis gives insight into the main sources of errors of the proposed approach.

Besides these major changes, we have also revised the text and the figures.

Thank you for your time and consideration.

Sincerely,

Valentina Premier on behalf of all co-authors

**Answer to the Community Comment #1 – Manuscript tc-2022-146**

Congratulations on this great work, I believe it is an important contribution to the field of SWE modeling. I find your solutions very original and at the same time intuitive and straightforward. The manuscript is very readable and detailed, most of the questions that came up while reading were answered later on. Having said that, I do have a number of comments, which I have gathered in the attached document. I hope they are clear and that they will be useful to you, and my apologies if I interpreted the manuscript incorrectly.

We thank Pau Wiersma for his constructive and deep feedbacks and comments on our manuscript. We went through each point and took advantage of this community comment to improve the quality of the manuscript. Our answers are reported in blue.

*Missing information on the SAR data*

Using the Sentinel-1 SAR imagery to correct the daily snow cover maps is a very creative and effective solution, and probably one of the main contributions of this paper. However, even though most of the methodology is based on the SAR-derived catchment state and it is practically treated as ground truth, there is a lack of background information and uncertainty estimation:

- It is not mentioned how forests are dealt with in the identification of the catchment state. Marin et al. (2020), from which this part of the methodology is adapted, mention that "the response to the wet snow becomes more complex in case of the snowpack in forest", but leave this out of the scope of their study. Therefore, if forests are included in this study (which I assume from the elevation ranges of the catchments), it should be elaborated how effective the methodology is in forested areas. It would also be useful to know what percentage of the catchments is covered by forests.

  Thank you for pointing this out. The percentage of forested area is around 25% for the Schnals catchment and 32% for the South Fork catchment (data source Masanobu et al., 2014). This information can be included in the description of the study areas in the manuscript. Consequently, masking out these areas would have resulted in ignoring a quite big portion of the catchment, as happens for most of the catchments located in our Alps.
  Indeed, the mechanisms that drive the SAR backscattering in forested areas covered by snow become more complex. The radar signal interacts with both the wet snow, which covers the ground and/or the top of the canopy, and the various parts of the trees. The different forest density, the type of the trees and the distribution of the snow above or below the canopy may generate temporal signature that are different from the U-shape described in Marin et al., 2020. Without trying to solve all the possible complex backscattering interactions explicitly, in the paper we propose to apply the method only for those pixels in the scene where the characteristic U-shape signature appears. For doing this, we set up the rule described at L341 of the manuscript, i.e. "If at least one track shows a drop of at least 2 dB in the signal (Nagler and Rott, 2000) w.r.t. a moving average of the 12 previous days, that day is considered to be in ablation." We have observed from our experimental results that this simple rule applied to the daily interpolated backscattering time series is effective in detecting if the multitemporal backscattering signal of a given pixel present the characteristic U shape or not. Following the proposed approach, we then consider in ablation all the days from this moment until the snow disappearance (this can be better clarified in the next version of the manuscript). For the pixels that do not present the characteristic U-shape, we only rely on the degree day model for the identification of the ablation. As an example, we add here a figure showing the forest mask and the pixels where S-1 information has been used to detect the ablation phase. The plot refers to the South Fork catchment for the season 2020/21. For brevity, we omit the other catchment and other seasons that show similar results. It is possible to see that our proposed algorithm to detect the U-shape of the backscattering fails right over forested areas (in white) while works elsewhere.

[Figure]

[Figure]

forest
DD timing detection
S1 timing detection

However, we should be aware that forested areas are challenging not only for what concerns S1 backscattering, but also for the degree day model as well as the snow cover detection. For example, the interception by canopy influences the melting phase, due to the impact of trees on the heat conduction and radiation balance that are not considered in the degree day model. Furthermore, snow under canopy is difficult to detect by optical satellites. Notwithstanding, we decided not to mask out forests given their quite significant percentage in the analyzed catchments and the obtained result that do not show a bias in correspondence of canopy presence. In any case, we also believe that this represents a hot topic that needs further research.

- The temporal and spatial resolution of the SAR imagery is never explicitly mentioned.

  The spatial resolution is 25 m, the same as the final time-series. We will add this information to the manuscript. The temporal resolution is 6 days considering the same geometry of acquisition, however all the available S1 acquisition tracks over the catchment have been considered. For example, in Schnals catchment up to 3 tracks are acquired in the timeframe of 6 days (i.e., $t_0$, $t_{0+1}$, $t_{0+5}$). It is worth stressing the fact that we applied a linear interpolation in time to obtain daily data as stated in L341.

- The uncertainty as quantified by Marin et al. (2020) is not mentioned either. They estimate the RMSE of the start of the moistening phase to be 6 days. An error of 6 days in the ablation identification could lead to large differences in the SWE reconstruction (equation 3), and should therefore be acknowledged or taken into account.

  Thanks for rising this point. We consider this information very relevant also for this paper and we will recall it in the revised version of the manuscript. Beside the uncertainty introduced by S1, we observed and discussed in the paper that the possible errors of the proposed method are mitigated by the fact that the potential melting calculated via the degree day model is low in the early melting i.e., in the moistening and ripening phases due to the low air temperatures (see answer below for further reasoning on this important part of the proposed approach).
  Finally, it is worth mentioning the fact that the uncertainty provided in Marin et al., (2020) has been calculated w.r.t. the output of a snow model, which is still prone to error (even if particular attention has been dedicated to the calibration and to the selection of the test site, a residual error may be still present).

- In L186 and L342 it is mentioned that ablation only occurs with a drop of the SAR backscattering signal, but during the runoff phase the backscattering signal is increasing (Marin et al., 2020). I assume that this increase should also be included in the definition of the ablation days.

  Thanks for pointing out this issue. We identify the ablation as the period from the day when a − 2dB drop of S1 backscattering is detected until the snow identified by optical-derived time series disappears. This will be better clarified in the final version of the manuscript.

- Sentinel-1 is only mentioned as S1 and never fully spelled out.

    Thank you for this observation. We will add this in the manuscript (L169).

*SWE loss during moistening and ripening phase*

In the methodology the assumption is made that during an ablation day with a non-zero DD there will always be a loss of SWE. However, if I understood it correctly these ablation days include all three phases of snow melt (moistening, ripening and runoff), even though during the moistening and ripening phases there is rarely any SWE loss, as also shown by Marin et al. (2020). It would perhaps be more physically accurate to limit the SWE loss to the runoff phase. This would not change the total amount of calculated melt energy and therefore accumulation, but it would possibly change the peak SWE estimate and likely change the timing of the SWE loss. In figure 3 for instance, if the first ablation phase would consist only of moistening or ripening there would be no loss of SWE and the peak SWE would be higher after the second accumulation phase. Moreover, if the runoff is limited to the runoff phase, the SWE loss would be delayed and more concentrated. This would have significant hydrological implications. Since the classification between the three snow melt phases can easily be determined from the SAR imagery, it would seem that adding this information to the SWE reconstruction would not lead to a large increase in complexity of the methodology. It would, on the other hand, possibly lead to an increased physical basis of the methodology, especially given the known issues around a constant DD factor (Magnusson et al., 2015). I'd be interested to know your thoughts on this.

We appreciate your comment that gives us the opportunity to better clarify this crucial point. We agree that considering the SWE loss happening during the runoff phase, which coincides with the minimum of the backscattering, is physically more accurate than considering the moistening, i.e., the day when a significant drop in the backscattering (at least of 2 dB) is observed. However, as described previously in the third answer, the uncertainty in the phase detection through S1 also plays a key role. For example, as illustrated in Marin et al., (2020) in Table 5 the difference in terms of days regarding the runoff onset had reached +19 days in one unlucky case.

This delays can introduce considerable errors since this phase is characterized by relative hot temperatures and consequently high melting rates. On the other hand, anticipating this phase does not lead to large errors due to lower temperatures and lower melting rates. Given our experimental results, we think that it is more conservative to make a mistake in the anticipate the detection of the runoff onset. Moreover, the backscattering signal may be difficult to interpret especially in case of alternated moistening and refreezing events or multiple runoff events. These situations lead to multiple local minima in the signal, as also shown in Marin et al. (2020) in figure 4c or 4g that we report also here, making the definition of a detection rule involved.

[Figure]

Nevertheless, it is important to stress the fact that the information provided by S1 provides more detailed information about the spatial distribution of the melting process than the DD derived by the spatialized temperature. In this work a first exploitation of this information has been proposed but several future developments are needed for a complete exploitation of the S1 time series as discussed in the paper. This should start from a normalization of the backscattering acquired for each of the available tracks for the effects of the topography. This may help in the identification of the complex variations described in the example before (e.g., Small, D. (2011)).

*Figure C3 and C2 are more informative than figure 9*

Showing how the modeled SWE behaves as a function of the elevation, slope and aspect is indeed very insightful, but only when compared to the behavior of the observed SWE. Therefore, I find figure C2 more informative than figure 9, even if ideally the maximum SWE would indeed be the best moment for comparison. However, figure C2 does not remove the bias and therefore does not allow for a relative comparison between the model and the observations per terrain parameter. Figure C3 does this, but the high number of figures does not allow for an easy general comparison. A compressed version of figure C3, for example with averaged biases, could potentially be more informative and would be easier to include in the main text as well. Other than that, I believe figure C2 would be clearer with lines instead of bars, and figure 12 could be left out of the manuscript.

We agree that figures C2 and C3 contain a large number of subfigures, but we believe that it is important to show in detail how the proposed approach performs w.r.t. a reference product and to differentiate among the different periods of the year when the ASO product is available. So, we would keep both figures in the Appendix, as already done. On the other hand, Figures 9 and 12 show the temporal stability of the method. They also represent an easy and direct comparison with, for example, the output of a third-party hydrological modeling run on the same basin. A compressed version of C3, as suggested, would show how the errors are distributed over the different topographic parameters but provides less details, which may be useful from a hydrological point of view. Even if we are in favor of adding this plot to the paper, we will wait for other reviewers' opinion on this matter if any.

*Figure 13*

The comparison between the modeled SWE and the discharge is very insightful, but the accompanying explanation is lacking in depth. Indeed the peak SWE correctly matches the peak discharge among the two years, but the timing in both years shows very different behavior. In 2020- 2021 the response is much more direct, while in 2019-2020 it's more delayed. Is this because the soil in 2020-2021 is already saturated after the rainfall events in Jan-March? And which part of the discharge after July originated from snow melt and which part from rainfall? On a side note, it would perhaps be insightful to show the rainfall rates on the inverted y-axis, and have the SWE loss and discharge on the same y-axis with the same units (e.g. m3/day).

We thank you for this meaningful comment that stimulated our investigation. We have reported in the manuscript only the comparison between the total SWE calculated for the entire catchment area and the discharge measured at Schnalserbach – Gerstgras, since this was born as a very qualitative analysis. To answer this question and further investigate our results, we decided to deepen the analysis and consider the SWE related to the subcatchment closed at the outlet point Schnalserbach – Gerstgras, as shown in the figure below. This makes the two variables more comparable, despite the analysis remains qualitative.

[Figure]

We present here these two plots for the two seasons, where we report the SWE variations that are associated with a runoff (i.e., only when they are associated with a decrease of SWE hence generating a runoff) and the discharge expressed in m3/day and with the same scale as suggested, and on the other axis the daily precipitation expressed in mm.

[Figure]

[Figure]

While a proper answer to your question would require a complete hydrological analysis (that is out from the purpose of this paper) and a hydraulic characterization of the watershed properties, we can observe that there is a good agreement in terms of both timing and quantity among snow generated runoff and discharge. The riverine discharge starts increasing in correspondence with the snowmelt and it starts decreasing when also the snowmelt is reduced for both periods. However, it is true that during the first year there is a delay while the response is more direct for the second season. More than differences in terms of precipitation that do not show any evidence, we ascribe this situation to a different snowmelt rate. Indeed, the season 2019-20 shows a longer distributed snowmelt period, interrupted by periods with lover SWE output (as end of March-beginning of April, beginning of May or middle of June). This situation may favor ground infiltration with a predominance of subsurface runoff w.r.t. surface runoff, that contributes slowly to the discharge. On the other hand, the season 2020-21 shows a long and high intensity SWE release (end of May-end of June) that may cause a sudden saturation of the soil, with predominant surface runoff that contributes more directly to the discharge. However, other contributions should be considered for a proper analysis, as for example the storage of water in the two snow reservoirs that are present in the territory. We can add this discussion to the manuscript.

**Minor comments**

- The resolution of the figures is often not high enough to be able to distinguish important details, especially in the spatial plots. Unless this is a result of the compiling of the preprint, increasing the resolution or saving the figures in vector format (for the graphs) would benefit the manuscript.
  We agree with this observation. Unfortunately, due to the large size of the manuscript, we compressed it without taking care of the result. But we will absolutely consider this problem in the next version.

- The legend nor the caption in figure 4 explain what the red points stand for (which I assume to be temperature stations).
  Thank you for noticing this. The red points represent the temperature stations. We will add this information to the legend.

- Line 289 ("Note that the number of days in accumulation varies for each pixel and consequently the coefficient is function of time and space.") contradicts L104 ("According to the state, that is assumed to be homogeneous for all the pixels of the catchment,... ").
  We would like to explain the sentence "the number of days in accumulation varies for each pixel" more clearly. It is true that the state is assumed to be homogeneous for all the pixels of the catchment, but only for those pixels that are snow covered. The necessary condition to have an accumulation, is obviously that the pixel must be snow covered. This information is provided by remote sensing observation. So, even though the state is assumed to be homogeneous, the snow duration varies for each pixel resulting in a different number of days in accumulation for each pixel. Consequently, the coefficient k that is the subject of L289, is a function of time and space. We propose to rephrase the second sentence with "According to the state, that is assumed to be homogeneous for all the snow-covered pixels of the catchment,..." to state this more clearly.

- Even if it's clear from the text, the caption of table 1 should perhaps mention of which catchment these results are.
  Thanks for the comment. We will add this in the caption.

- L372: "The highest bias and RMSE values are generally encountered in the mid-winter acquisitions." As I see it, this is not reflected by the RMSE and bias values we see in table 1, unless mid-winter means march-may.
  Yes, we meant the first dates acquired before the full melting period in March-May. We propose to rephrase the sentence with "The highest bias and RMSE values are generally encountered for the first available ASO acquisitions during the period March-May."

- In the author contributions, MC should be CM.
  Thanks for the comment. We will correct this.

- L433 I'm guessing that you're talking about MODIS, but I believe it would be good for clarity if you mentioned this.
  We are talking about remote sensing products in general. The two examples we reported in the paper (L435) are: i) the SCA variations, which can be derived by optical data as MODIS, Sentinel-2 or Landsat and can be used to identify an accumulation or melting event, and ii) the snow depth, which can be derived from S1 at a resolution of 1 km (Lievens et al., 2022), can be used to identify the accumulation events.

- L464 "replacement with snow", I assume this should be "replacement with snow-free".
  Yes, you are right. Thank you for noticing this mistake.

- Figure 14 and 15 are rather small, and I would perhaps have appreciated to see the accumulation and ablation phases reflected in the background of the plots.
  Thank you for this nice suggestion. Here we add the updated plots with a cyan background for days identified to be in ablation and pink background for day identified to be in accumulation.

The following figures refer to the Schnals catchment.

[Figure]

The following figures refer to the South Fork catchment.

[Figure]

**References**

Lievens, H., Brangers, I., Marshall, H.-P., Jonas, T., Olefs, M., and De Lannoy, G.: Sentinel-1 snow depth retrieval at sub-kilometer resolution over the European Alps, The Cryosphere, 16, 159–177, 2022

Masanobu Shimada, Takuya Itoh, Takeshi Motooka, Manabu Watanabe, Shiraishi Tomohiro, Rajesh Thapa, and Richard Lucas, "New Global Forest/Non-forest Maps from ALOS PALSAR Data (2007-2010)", Remote Sensing of Environment, 155, pp. 13-31, December 2014. doi:10.1016/j.rse.2014.04.014.

Small, David. "Flattening gamma: Radiometric terrain correction for SAR imagery." IEEE Transactions on Geoscience and Remote Sensing 49.8 (2011): 3081-3093.

**Answer to Anonymous Referee #1 – Manuscript tc-2022-146**

I would like to congratulate the authors to this very interesing work, as they have presented a novel methodology and useful contribution to snow science. The presented approach might open new doors for reconstructing snow water resources also on larger scales, by employing multiple data sources. Especially, the usage of SAR backscattering signals to both, detect snow ablation, and indirectly correct daily high resolution SCA maps is an original methodology. The simple, yet efficient basic concept of the reconstruction approach, together with the parsimonious use of in-situ data is highly appealing. In my opinion, the organization of the work is good and the presentation of the results is rather clear. However, many sentences could be restructured to make for an easier read. I have made various comments on the manuscript that I hope will help to improve the paper.

The authors thank the anonymous referee for the positive comments. In the following, we address each point raised by the referee and take advantage of them to improve the quality of the manuscript. Our answers are shown in blue.

There are still some important details that remain unclear to me, especially concerting the determination of the catchment state:

- It would be helpful to see a more detailed description about how the ablation state is derived from Sentinel 1 data and how this would translate to the three snowpack phases described in Marin et al. (2020) (i.e. moistening, ripening, runoff). In line 185 only a "relevant drop" in backscattering is mentioned. Does this mean the catchment state ablation already starts when any liquid water is present in the snowpack? Since it seems possible (at least with S1A and S1B) to identify the snowpack runoff phase and a separation to previous moistening and ripening phases, the use of a DD-style melt model would be much more justified - as this is practically eliminating the need to track energy states (cold content) in the snowpack modelling. Please clarify if ablation is based on a drop in backscattering (melting phase) like line 185 suggests, or on the minimum of backscattering (runoff phase), as implied line 174. If the latter is the case, there is more physical grounds to employ a DD-style melt model and this should be brought forward in the text. However, if the ablation state is corresbonding to the moistening phase (i.e. a mere drop in backscattering), this decission should be also explained in more detail.

  We thank the reviewer for this comment. The same point was noted in the Community Comment #1, for which we provided an answer. As stated by the Reviewer, the choice to consider the runoff onset as the beginning of the ablation phase is more adequate from a physical point of view. In the current version of the manuscript, we identified the ablation as the period beginning from the day when a $-2dB$ drop in S1 backscattering is detected until the total disappearance of the snow, as identified by optical-derived time series. During this period, snowmelt is possible only if the stations do not identify any accumulation event and if the DD method provides an effective potential melting. In other words, we assumed that the ablation onset corresponds to the moistening onset. We decided to consider the moistening onset instead of the runoff because Sentinel-1 presents an uncertainty that is about 7 days for the runoff phase as quantified by Marin et al. (2020). Following the suggestion of the Reviewer to perform a sensitivity analysis (see the answer below for all the details), we show that the error introduced in the final SWE reconstruction is higher when introducing a delay in detecting the runoff onset rather than anticipating it. In fact, temperatures are lower and consequently, the potential melting calculated through the DD method is lower in the early moistening/runoff phase. On the other hand, a delay may cause a significant loss in terms of SWE, being the temperatures higher, and consequently also the potential melting, as the melting season advances. For these reasons, we decided to implement a conservative approach based on the moistening onset i.e., anticipating the runoff onset to be sure to not delay it and thus introduce significant errors.

  However, we agree that from a physical point of view, the start of runoff should be considered as the initial day of ablation, especially if we consider that the revisit time of SAR satellite missions may improve greatly in the future. To this purpose, and to provide a more quantitative analysis, we recalculated the metrics with both approaches. The new results differ from those reported in the initial manuscript since during the time of the revision we improved the accuracy of the snow cover maps. For the catchment in the Sierra Nevada by considering the three hydrological seasons (2018-2021), we obtained a bias of $-22$ mm, an RMSE of 212 mm, and a correlation of 0.74 when considering the runoff, and a bias of $-20$ mm, an RMSE of 216 mm, and a correlation of 0.75 when considering the moistening. The results refer to the evaluation of our product against the ASO product. For the Schnals catchment, the evaluation against the manual measurements for the season 2020/21 showed a bias of -5 mm, an RMSE of 191 mm, and a correlation of 0.35 when considering the runoff phase while a bias of 19 mm, an RMSE of 218 mm, and a correlation of 0.36 when considering the moistening phase. It is worth noticing that this is a limited amount of point observations taken using an SWE coring tube that may not be fully representative of a 25x25 m pixel. Given the negligible difference in this specific case, we report in

the final version of the manuscript only the version with the runoff onset identification. We introduced both a dedicated subsection (2.1.2) where we better explained how the ablation state is identified, and a dedicated subsection in the Discussion where the sensitivity analysis is reported

- If I understood correctly, the catchment states ablation and equilibrium can exist both at the same time-step (different pixels have different states), but in accumulation all pixels have this state. Line 104 and 189 contradict themselves in this regard. Please clarify.

Thank you for pointing out this aspect that may generate misunderstandings. By considering the constructive comments received by all the reviewers, we introduced important changes in the method section. We explained that ideally the state should be identified for each pixel (and not at the catchment level as done in the previous version of the manuscript). The possible states are (see Figure 1): i) accumulation that represents an SWE increase ($\Delta SWE>0$), ii) ablation that represents an SWE reduction ($\Delta SWE<0$), and iii) equilibrium that represents a stable SWE ($\Delta SWE=0$).

| State | $\Delta SWE$ | Class transition t-1 | Class transition t | Description |
|---|---|---|---|---|
| Accumulation | >0 | ■ | □ | Snow on bare ground |
| | | □ | □ | Snow on snow |
| Ablation | <0 | □ | ■ | Snowpack disappearance |
| | | □ | □ | Snowpack reduction |
| Equilibrium | =0 | □ | □ | Stable snowpack |
| | | ■ | ■ | Bare ground |

Legend: □ = snow ■ = snow-free

*Figure 1 Definition of the three possible states: accumulation, ablation and equilibrium. The possible class transitions at pixel level associated with the state are described.*

The phenomena that cause a SWE variation are several, as snowfall, melting, sublimation, human activities or redistribution due to wind or gravitational transport, e.g., avalanches. However, we refer mainly to snowfalls if accumulation and to melting if ablation. In fact, we propose to estimate the SWE to be added considering a quantity proportional to the snow depth/SWE variations, thus, in an ideal case, including only fresh snow as the main driver. Similarly, the amount of SWE to be subtracted is calculated using a DD model and, therefore, it only represents melted snow. Trivially, SWE remains constant when equilibrium.

The state varies pixel-wise due to the topography and meteorology of the study area. However, it is difficult to extrapolate this information with the necessary spatial detail. For the accumulation identification, the sources of information that can be exploited are few. In the paper, we propose that the accumulation can be retrieved by a network of automatic weather stations (AWS) that measure snow depth/SWE. It is important to note that a station is representative of a limited area whose extension is highly variable depending on the complexity of the terrain. However, by considering only the snowfall events, one can think that from a network of stations distributed with elevation, it is possible to divide the catchment into different elevation belts that can be considered homogenous. However, in many basins, this is quite far from reality. As a common configuration for snow monitoring, we have a single station located at a high point of the catchment, that is informative enough to identify the accumulation events but not their extent. In such a situation, as described in the paper, we considered that the snowfalls occur throughout the snow-covered area of the catchment. We are aware that this assumption may be erroneous, especially in the case of mixed conditions. For example, snowfall may be observed at high elevations, together with rain-on-snow at low elevations that causes snowmelt. However, it has been shown in the literature that the estimation of the snowfall limit may be very challenging (see, e.g., Fehlmann et al., 2018). For this reason, we believe that introducing an approach based on temperature thresholds to define the snowfall limit may still represent a strong simplification that does not necessarily add value to our approach.

On the other hand, the ablation state can be retrieved by using i) temperature index models (it is generally easier to spatialize temperature data w.r.t. snow depth observations), but also ii) multi-temporal SAR observations derived by Sentinel-1. Marin et al. (2020) investigated the relationship between the SAR backscattering and the three melting phases, i.e., the moistening, ripening, and runoff phase. In detail, they showed that if the SAR backscattering is interested in a decrease of at least 2 dB, the snowpack is assumed to get moistened (Nagler and Rott, 2000). First, this decrease

affects only the afternoon signal (beginning of the moistening phase). When it also affects the morning signal, the ripening phase starts. Finally, the backscattering increases as soon as the SWE starts to decrease, which corresponds to the beginning of the runoff phase. This moment represents the first contribution of the snowpack to the release of water. The multi-temporal analysis of the SAR backscattering represents a novel way to identify the ongoing melting in a spatialized manner. By integrating this information into a degree-day model, it is possible to exclude false early melting dates.

We included all these explanations in the manuscript. Currently, we are working toward a solution that exploit remote sensing information only i.e., natively spatialized information (as already mentioned in the manuscript, for example by considering surface temperature measurements from satellites, meteorological radar or SAR derived snow depth/SWE information). However, this needs further research and it left as a future development in the current manuscript.

- Although bringing the reconstructed SWE time-series in context to the catchment discharge might provide some insights into the estimated snow cover dynamics, however, the way this information is presented in Figure 13 and interpreted in the text (line 409 -411) is not suitable for this purpose. There are basically two statements emerging from this analysis: i) there was more snow in one year than the other, ii) SWE decreases and subsequently discharge increases at some point. As it can be seen in Figure 13 there are very different discharge responses in the spring freshet between the two years. I do not see this analysis to be much helpful in the current state and would advise either to remove it from the manuscript, or expand the analysis (giving more information about precipitation, hydrological characteristics of the catchment, and changing the units in the figure (e.g. to mm)).

The authors thank the reviewer for this comment. This is also in line with a comment provided by the community (see CC#1). As suggested by Pau Wiersma (CC#1), we deepened the analysis on the relationship between the SWE runoff and the measured discharge. We considered the SWE related to the subcatchment closed at the outlet point Schnalserbach – Gerstgras, as shown in Figure 2. This makes the two variables more comparable, although the analysis remains qualitative.

[Figure]

*Figure 2 Overview of the subcatchment whose outlet point corresponds to the location of the discharge measurement.*

We presented two new plots (Figures 3 and 4) for the two seasons that may replace Figure 13 in the manuscript. The analyzed variables are: i) the SWE variations that are associated with a runoff (i.e., only when they are associated with a decrease of SWE), ii) the discharge; and iii) the precipitation measured at Vernagt expressed in mm/day. We better analyze what we can observe from these plots. In detail, we can observe that there is a good agreement in terms of both timing and quantity among snow-generated runoff and discharge confirming that the catchment is snowmelt dominated. The discharge starts increasing in correspondence with the snowmelt and it starts decreasing when also the snowmelt is reduced for both periods. We can observe that the first year shows a delay while the response is more direct for the second season. More than differences in terms of precipitation, we ascribe this situation to a different snowmelt rate. Indeed, the season 2019-20 shows an earlier, weaker, and longer distributed snowmelt period, interrupted by periods

with low SWE output (such as the end of March-beginning of April, beginning of May, or middle of June). This situation may favor ground infiltration with a predominance of subsurface runoff w.r.t. surface runoff, that contributes slowly to the discharge. On the other hand, the 2020-21 season shows a long and high intensity SWE release (end of May-end of June) that may cause a sudden saturation of the soil, with predominant surface runoff that contributes more directly to the discharge. This hypothesis may also be confirmed by recent literature, showing that when snowmelt is earlier, it is also less intense and the runoff response could be reduced, with strong implications for future climate change impacts (Musselman et al., 2017). However, other contributions should be also considered, as for example, the storage of water in the two snow reservoirs that are present in the territory. While a proper analysis requires a complete hydrological study and a hydraulic characterization of the watershed properties, we believe that this simplified analysis shows the potentiality of the presented results in a real application. For this, we think that adding the information provided here to the Reviewer also to the manuscript may be interesting to the reader and stimulate works that will exploit data derived from the proposed approach.

[Figure]

*Figure 3 Snow generated runoff, discharge and precipitations for a subcatchment in the Schnals valley for the season 2019/20.*

[Figure]

*Figure 4 Snow generated runoff, discharge and precipitations for a subcatchment in the Schnals valley for the season 2020/21.*

- The authors discuss various sources of uncertainty in the methodology and state that due to a number of preprocessing steps a formal sensitivity analysis is difficult to perform. However, it might be still very valuable for the reader to get a feeling about how possible errors might translate to the SWE reconstruction. Please consider the possibility to provide a simplified version of part of the problem, by e.g. perturbing the values of tSD and tSA (and perhaps keeping the states

constant during this time) and showing the consequences in terms of peak SWE. This could also help to underline the statements in line 480f.

We thank the Reviewer for this suggestion. As mentioned, it might be difficult to really estimate all the uncertainty sources and provide a proper sensitivity analysis on the parameters that play a role in the proposed approach. However, we follow the suggestion of the Reviewer and we carried out a simplified sensitivity analysis. For the sake of clarity, we propose to investigate how the parameters affect the final SWE reconstruction by considering the pixel where the station Volcanic Knob provides continuous SWE measurements in the Sierra Nevada catchment. The parameters that we believe play an important role in the methodology are i) the degree day factor, ii) the SWE threshold used to identify the states, iii) the time of snow disappearance (tSD), iv) the time of snow appearance (tSA), and v) the time of first ablation detected by S1 (we call it here tS1). We vary each of these parameters separately keeping the others constant and equal to the optimal case (i.e., the one with the lowest RMSE). The test is carried out for one season (2018/19). Although we are aware that this analysis is not exhaustive, it can give an overview of the most important sources of error.

[Figure]

*Figure 5 tSD = 27/06/2019, tSA=22/11/2018, tS1=22/04/2019 (as detected by the station), SWE threshold=2mm, a varies from 3 to 6 mm/(°Cd) by steps of 0.2.*

It is possible to notice in Figure 5 that the error increases linearly when a moves away from the optimal value, that is a=4.5 (as it was set in the manuscript).

[Figure]

*Figure 6 tSD = 27/06/2019, tSA=22/11/2018, tS1=22/04/2019 (as detected by the station), a=4.8 mm/(°Cd), SWE threshold varies from 0 to 20 mm by steps of 1 mm.*

It is possible to see in Figure 6 that, as expected, the higher the threshold, the greater the error. In fact, for too large thresholds, the method fails to detect the accumulation states. A snow threshold of 2 mm, as set in the manuscript, is acceptable.

[Figure]

*Figure 7 tSA=22/11/2018, tS1=22/04/2019 (as detected by the station), a=4.8 mm/(°Cd), SWE threshold 2 mm, tSD 27/06/2019 +- 15 days.*

It is possible to notice in Figure 7 that both underestimating and overestimating tSD introduce important errors in the reconstruction. In fact, at the end of the melting season the temperature is high and consequently the potential melting. A difference of +-5 days (which corresponds to the S2 repetition time) already introduces around 50 mm of RMSE.

[Figure]

*Figure 8 tSD 27/06/2019, tS1=22/04/2019 (as detected by the station), a=4.8 mm/(°Cd), SWE threshold 2 mm, tSA=22/11/2018 +- 15 days.*

It is possible to see in Figure 8 that the shift of tSA does not strongly affect the RMSE as tSD does. For negative shifts, the accuracy RMSE is constant since no SWE is added to the reconstruction. In fact, for those days, we find that the coefficient k (see Eq. 4) is 0 since it is calculated from the AWS. In other words, it means that the accumulation is not really happening before at least one station detects an increase in SWE.

[Figure]

*Figure 9  tSD = 27/06/2019, tSA=22/11/2018, a=4.8 mm/(°Cd), SWE threshold 2 mm, tS1=22/04/2019 +- 15 days.*

In this case, it is also possible to see in Figure 9 that the shift of tS1 does not strongly affect the RMSE as does tSD. The RMSE for negative shifts remains constant after a certain point, since for those days the DD model returns 0 potential melting, so there are no differences. This means that it is in general better to make an error anticipating the melting phase than postponing it.

Even though we are aware that this represents a very simplified analysis and might be not exhaustive, we can summarize that we expect that the error that most strongly affects the results is a shift in the date of snow disappearance. For this reason, we believe that the SWE reconstruction can fully benefit from the introduction of an accurate daily HR time series.

**Additional Comments**

Abstract/1 Introduction:

- 1 reconsider recasting the first sentence.

  We propose to change the sentence with: "The hydrological cycle is strongly influenced by seasonal snow accumulation and release. For this reason, mountains are often claimed as the "water towers" of the world."

- 3 how about ablation processes, or are you specifically talking about peak SWE?

  We meant here the peak of SWE, but for completeness and clarity it is better to include all the ablation processes. We change with "..the complex snow accumulation, redistribution, and ablation processes."

- 11 At this point, the reader might not follow what you mean with "time-series regularization from impossible transitions". Mentioned again at 105, 111, and 127 before finally explained in section 2.2 at 194.

  In the abstract we specify "… from impossible transitions, i.e. the erroneous change of the pixel class from snow to snow-free when it is expected to be in accumulation or, vice versa, from snow-free to snow when ablation." We also report the definition in L105 and 127.

- 17 I'm not particularly fond of using present perfect (throughout the manuscript). But that might be personal preference. Please consider using present tense (or simple past) consistently.

  Thank you. We also prefer the present tense.

- 21 not only on local hydrology, as many regions of the world rely e.g. on the spring freshet hundreds of km downstream

  Thank you for noticing this. We change with "…local and global hydrology"

- 23 „from at least 50%"please check again with Vivironi et al (2003). Although snowmelt is a major contributor of mountain water resources, as far as I know, the numbers given by Vivironi and others include mountain waters in general.

  Thank you for pointing out this. You are right. We changed the sentence with "Contribution of snow-dominated catchments to streamflow ranges from 40% of the total flow to sometimes more than 95%, depending on the region (Viviroli et al., 2003)"

- 27 precipitation variability is affected by orography, interpolation is affects by sampling density among other factors, and observations can be erroneous due to e.g. undercatch

  Thank you for this clarification. We propose to rephrase the sentences in this way: "In fact, precipitation data used as input for physically-based models are often affected by uncertainties, thus limiting the spatial accuracy of snow accumulation and melt models (Engel et al., 2017; Günther et al., 2019). The main challenges to obtain an accurate precipitation field arise from the strong spatial variability of the variable related to the orography, a generally scarce sampling density of the phenomenon that strongly influences the interpolation results, and possible inaccuracies in the measurements caused, for example, by undercatching (e.g., Prein and Gobiet, 2017)."

- 65 "accumulation and melt = „accumulation and ablation" (i.e. in this sense ablation includes erosion and evaporation etc.)

  Thank you. We accept your suggestion.

- 66 consider topography vs geomorphology (throughout the text)

  Thank you. We accept your suggestion.

- 80 they range from DD to complete energy balance models (as used in Bair et al 2016), and these do not require calibration

  We agree with the Reviewer, also according to the comment of Reviewer#2, that this sentence is not correct. With "calibration", we erroneously meant the tuning of parameters such as the degree day (or empirical melt factor) that can affect the final results but cannot be considered as a proper calibration parameter. We removed this sentence.

- 114 (Italy) like (USA)

  Thank you. It has been replaced.

2 Proposed approach to HR SWE reconstruction

- 120 first sentence is obsolete iMo.

  Thank you. We accept your suggestion and deleted it. Also, we introduced changes in the Methodology section according to the comments of the other reviewers.

- 132 please specify „too vast".

  We thank the reviewer for pointing out this issue. We noticed that this is also generating confusion based on the comments of the other reviewers. In the new manuscript, we propose to define the state at the pixel level. We explained that the method does not necessarily require working at the catchment level. However, the limitation is represented by how we define the accumulation. Given the scarcity of AWS within our study areas, we preferred to consider the accumulation state to occur for the entire snow-covered area of the considered catchment. For this reason, in the previous version we talked about a not "too vast" area. In this sense, we mean that the catchment is subject to similar meteorological forces, so the assumptions made for the accumulation state identification hold. In other words, this implies that the basin should be subjected to similar snowfall events. Fortunately, this is the case in many situations, except the catchment is not "too vast". Hence, rather than suggesting a perfect size we suggest analyzing the climatic and hydrologic characteristics of each subcatchment when dealing with a large area, i.e. to check if a correlation exists among the subcatchments in terms of climatic variables (temperature, precipitation, discharge).

- 136 Maybe this would be clearer: „in detail, the catchment state is characterized by the change in SWE, but is also associated with possible changes in SCA". Or similar.

  Thank you. We revised the whole section according to the other comments, so please refer to the new version where we talk about a pixel state and associated pixel class transition.

- 141 „extension" = extent

  Thank you. It has been replaced.

- 143 „...dSWE < 0 due to melt water drainage".

  We removed this part from the new version of the manuscript.

- 145 snow depth, not height. Anyway, better to talk here in terms of mass/SWE. e.g. „...if the snowpack is melting only partially".

Thank you. It has been replaced.

- 167-168 maybe better placed in the discussion section

  Thank you for your suggestion. We removed the sentence from Sec. 2.1 and we moved it in Sec. 5.1.

- 169 potentiality = potential to detect the presence of a melting snowpack…

  Thank you. It has been replaced.

- 174 most important in terms of what? Certainly not in terms of melt water production as melt rates increase towards later season. Peak SWE is not necessarily the peak of melt water runoff. Please clarify.

  Thank you for pointing this out. Yes, we agree that the SWE peak does not correspond to the melt water runoff peak. We meant here that this is the moment when water starts to be released from the snowpack. We propose to change the sentence in this way: "This moment represents the first contribution to the water release…"

- 181 replace "quotes" with "elevations", "elevation bands" or similar (throughout the manuscript)

  Thank you. All this part has been rephrased, so please refer to the new manuscript version.

- 189 ablation and equilibrium classes can exist at the same time, but 104 states that the state is assumed homogeneous for all pixels of the catchment. Please clarify!

  Thank you for pointing out this aspect that may generate misunderstandings. As explained in a previous answer, in the new version of the manuscript, we propose to introduce the state concept not at the catchment level, but at the pixel level. Ideally, each pixel may have a different state. Since the accumulation cannot be identified pixel-wise, we consider all the pixels to have the same state when the AWS detects a snowfall. On the other hand, ablation can be identified pixel-wise thanks to the use of S-1 and a spatialized degree day model. Hence, ablation and equilibrium can coexist.

- 196 contamination = obstruction?
  Thank you. It has been replaced.

- 276 which variable is used as external drift, elevation?

  Exactly. It has been added to the new version of the manuscript.

- 296 contemporary = simultaneously ?

  Thank you for correcting it. It has been replaced.

- Figure 3, I don't see this figure referenced anywhere in the text,

  Thank you for noticing this. We added a reference at the beginning of this paragraph.

3 Study Areas and Dataset Description

- You acknowledge forest canopy as an important source of uncertainty but do not give any information if, and how much of the area is forested.

  Thank you for pointing this out. The percentage of forest is around 25% for the Schnals catchment and 32% for the South Fork catchment (data source Shimada et al., 2014). We added to the description of the study areas (Section 3).

- 321 Are manual SWE observation only available for one of the two seasons?

  Yes, they are available only for the season 2020/21. We added this information to the text. The reason is that it was not possible to collect measurements during the previous season due to Covid restrictions.

- 336 S1 is not introduced in the text (unlike sentinel-2)

  Thank you for noticing this. We added this to the text.

- Figure 4: resolution should be improved; it is very hard to read.

  Thank you for noticing this. We improved the quality of the image.

- Figure 4b: Why show all SWE observations if you only use a few of them (e.g. in Fig 10)? Or is the mean performance metrics based on all of them? Please clarify.

  Thanks for pointing this out. The metrics are evaluated on the whole dataset, while the plot in Fig. 10 reports only a few examples. We clarified this in the text in this way "The results show a bias of  -5 mm, an RMSE of 191 mm, and a correlation of 0.35, indicating a generally good agreement. The overall performances are calculated w.r.t. the complete dataset, for sake of brevity we report in Fig. 10 the reconstructed SWE evaluated against the manual measurements only for a subset of the collected points. For more details on the measurement location of the selected subset, please refer to Appendix C, (see Fig. C1)." We also specified this in the Appendix. Moreover, for completeness, we also added Figure 10 (below) where we show the scatterplot between the measured and proposed SWE for the complete dataset as required by Reviewer#2. However, we think that it is interesting to keep Fig. 10 of the manuscript since it gives an overview of the SWE trend for different pixels selected from different locations, showing a variability that seems to be properly caught by the proposed method.

[Figure]

*Figure 10 Observed and proposed SWE in the Schnals catchment for the h.y. 2020/21.*

4 Results

- Figure 6: image resolution should be increased. "Trend" could be recast as "time-series"

  Thank you for noticing this. We check the quality of all the figures for the next manuscript version. We also replaced "trend" with "time-series" throughout the manuscript.

- Figure 7: rather small, also image resolution could be increased, caption does not mention ASO.

  We included ASO in the caption.

- Figure 8 / Table 1. Although it can make sense to specify total catchment wide SWE in Gt, I would much rather prefer it to be given in mm as well.

  We calculated the total SWE in mm and replaced Fig. 8 and Table 1.

- Figure 9 might be more helpful when ASO observations are included. I also think the plot does not support your interpretation that the drop in total SWE in very high elevations is due to gravitational redistribution (L 387), nor that steeper slopes present less SWE, nor that there is more snow on north facing slopes. Total SWE amounts are presented (in Gt), so this strongly depends on the area covered by this class (i.e. elevation bands). Same is true for slope and aspect bins. You need to scale with the area of a class to allow for these relative comparisons (express SWE in mm). Otherwise, you carrying the information about the catchment topography (e.g. hypsometry). x-axis: Aspect is not in degrees but in cardinal and ordinal directions

  We thank the reviewer for this suggestion. However, we cannot add the ASO observations since we are plotting the SWE maximum against elevation, slope, and aspect and the maximum of the season is not available for ASO. We propose to remove Fig. 9 from the main text. We added to the Appendix the figures below, where SWE is reported in mm for the different elevation, slope and aspect band (similar to Fig. C2) for all the dates when ASO was available. However, we believe that also Fig. C2 expressed in Gt is still informative. In fact, the two plots represent something different. We originally expressed the SWE in Gt because we wanted to show how the total cumulated SWE is behaving w.r.t. the reference. For example, high slope areas show to agree when expressing SWE in Gt since these areas are very small and do not count that much when considering an overall balance. On the other hand, the analysis expressed in mm highlights relevant differences linked to the biases we encounter when analyzing the spatial maps of Fig. C1. In the figure below, it is possible to notice differences, especially for steep slopes, where the proposed method underestimates SWE w.r.t. ASO. However, we would expect less SWE for these steeper slopes that promote gravitational transport. This point can be better investigated and discussed in the revised version of the manuscript.

[Figure]

*Figure 11 17/03/2019.*

[Figure]

*Figure 12 02/05/2019.*

[Figure]

*Figure 13 09/06/2019.*

[Figure]

*Figure 14 04/07/2019.*

[Figure]

*Figure 15 14/07/2019.*

[Figure]

*Figure 16 15/04/2020.*

[Figure]

*Figure 17 05/05/2020.*

[Figure]

**Figure 18 23/05/2020.**

[Figure]

*Figure 19 08/06/2020.*

[Figure]

*Figure 20 26/02/2021.*

[Figure]

[Figure]

*Figure 21 31/03/2021.*  *Figure 22 03/05/2021.*

Finally, we propose to include these figures in the Appendix, keep Fig. C2 and include one of the dates as an example to show also in the main text.

- 397 what is meant by "only few examples"? Are there manual SWE observations for other years as well, or is this just a subset of the locations shown in Fig 4b?

  Thank you for noticing this. As specified in the previous comment, we clarified this in the text.

- 401 tends to increase

  Thank you for noticing this. It has been corrected.

- Figure 12, see fig 9. Since no spatial observations are available in Schnals, I don't consider this figure very helpful.

  According to the previous comment and the CC1 comment, we removed both Fig. 9 and 12.

- What about an evaluation against the automated snow depth sensor in Schnals?

  Thank you for raising this point. However, since we do not produce snow depth maps with our method and since we do not estimate the snow density, this comparison cannot be done.  We could simulate SWE in the stations through a snow model (e.g., SNOWPACK). However, the station in Schnals is disturbed by wind erosion/accumulation (wind speed is not measured at the station) so we believe that this might affect the accuracy of the final simulation making the comparison difficult.

5 Discussion

- 413 maybe better "…quantitative and qualitative evaluation of the proposed SWE reconstruction over two study areas"

  Thank you for your suggestion. We modified the text accordingly.

- Figure 16: caption: use the same naming (i.e. "original snow cover map") as in the legends and captions of the previous Figures (14, 15)

  Thank you for your suggestion. We modified the caption accordingly.

- Section 5.1 only focusses on errors in predicting the accumulation state. Please expound upon the uncertainties associated with predicting the ablation/equilibrium state.

We apologize for this lack. We believe that we can integrate this part also by adding the previous discussion about the uncertainty and how the errors propagate. In this sense, we discuss the importance of predicting an accurate time of snow disappearance. On the other hand, errors in predicting the start of the ablation through Sentinel-1 seem not to strongly affect the performances. However, we would like to stress the fact that the value of the max SWE for a given pixel is driven by the number of days in ablation. Therefore, we expect that especially in the last melting phase it is very important to accurately identify the ablation state.

- 480-485 Please clarify/recast: "M" and "potential melting" are used in the same sentence and might confuse readers. Suggestion: "Since potential melting values at the end of the ablation period are high, an erroneous estimation of tSD strongly affects the reconstruction of peak SWE."

  Thank you for your suggestion. We modified the text coherently.

Appendix:

- Figure A1: caption does not state that this is Schnals

  Thank you for your suggestion. We modified the caption coherently.

- Figure C2: x-axis: Aspect is not in degrees but in cardinal and ordinal directions; As a line plot it would be easier to compare observation and the proposed approach.

  You are right. Apologies for this oversight. We also replaced the bars with a line plot.

**References**

Bair, E. H., Rittger, K., Davis, R. E., Painter, T. H., & Dozier, J. (2016). Validating reconstruction of snow water equivalent in California's Sierra Nevada using measurements from the NASA Airborne Snow Observatory. Water Resources Research, 52(11), 8437-8460.

Engel, M., Notarnicola, C., Endrizzi, S., & Bertoldi, G. (2017). Snow model sensitivity analysis to understand spatial and temporal snow dynamics in a high-elevation catchment. Hydrological processes, 31(23), 4151-4168.

Günther, D., Marke, T., Essery, R., & Strasser, U. (2019). Uncertainties in snowpack simulations—Assessing the impact of model structure, parameter choice, and forcing data error on point-scale energy balance snow model performance. Water Resources Research, 55(4), 2779-2800.

Marin, C., Bertoldi, G., Premier, V., Callegari, M., Brida, C., Hürkamp, K., ... & Notarnicola, C. (2020). Use of Sentinel-1 radar observations to evaluate snowmelt dynamics in alpine regions. The Cryosphere, 14(3), 935-956.

Musselman, K. N., Clark, M. P., Liu, C., Ikeda, K., & Rasmussen, R. (2017). Slower snowmelt in a warmer world. Nature Climate Change, 7(3), 214-219.

Prein, A. F., & Gobiet, A. (2017). Impacts of uncertainties in European gridded precipitation observations on regional climate analysis. International Journal of Climatology, 37(1), 305-327.

Shimada, M., Itoh, T., Motooka, T., Watanabe, M., Shiraishi, T., Thapa, R., & Lucas, R. (2014). New global forest/non-forest maps from ALOS PALSAR data (2007–2010). Remote Sensing of environment, 155, 13-31.

Viviroli, D., Weingartner, R., & Messerli, B. (2003). Assessing the hydrological significance of the world's mountains. Mountain research and Development, 23(1), 32-40.

**Answer to the Referee #2 – Manuscript tc-2022-146**

My overall impression of the work is that it potentially represents a novel contribution and one that takes the SWE reconstruction approach into a very new direction with potential value. I am excited by this potential and think the authors should be applauded for taking on this work. However, I found some aspects of the paper extremely difficult to follow (identification of catchment state) and in general think the quality of the writing and use of english to be quite problematic. I think the paper is valuable but I would strongly suggest more editorial consideration in the context of sentence structure and grammar as nearly every-other sentence suffers from some type of grammatical error. Most of these errors were quite small and did not interfere with my understanding of the points being made but some errors were more considerable. These errors were far too numerous for me to spend the time to point them all out or to correct them all. My comments are included in the comments margin of the attached PDF. Most of these are broader-context comments that align with my perspective that the paper needs very major revisions. Thank you, Noah Molotch

The authors thank the Reviewer Noah Molotch for his meaningful comments and suggestions. We agree that some parts of the manuscript need to be revised and restructured following the constructive comments from the referees. In particular, the identification of the catchment state has been rewritten to make this part easier for readers. In addition, we revised the document and tried to correct all grammar mistakes. We also believe that the English-language copy-editing that the journal offers in case of acceptance will further help improving the quality of the writing. We went through each comment reported in the PDF. Our answers are reported in blue.

L81 I don't believe this statement is true. I am unaware of any "calibration" in the application of SWE reconstruction models.

We agree with the Reviewer that this sentence is not correct. With "calibration", we erroneously meant the tuning of parameters such as the degree day (or empirical melt factor) that can affect the final results. Since there are no calibration parameters in the SWE reconstruction, we removed this sentence from the revised version of the manuscript.

L83 This is not true. Presumably 30-m resolution would not be considered low resolution in this context and there are many papers in the literature that have applied SWE reconstruction using Landsat data - examples that I am aware of are below but there may now be additional papers.

Molotch, N.P., T.H. Painter, R.C. Bales, and J. Dozier, Incorporating remotely sensed snow albedo into a spatially distributed snowmelt model, Geophysical Research Letters, VOL. 31, doi:10.1029/2003GL019063, 2004.

Molotch, N.P., and R.C. Bales, Scaling snow observations from the point to the grid-element: implications for observation network design, Water Resources Research, VOL. 41, doi: 10.1029/2005WR004229, 2005.

Molotch, N.P., and R.C. Bales, Comparison of ground-based and airborne snow-surface albedo parameterizations in an alpine watershed: impact on snowpack mass balance, Water Resources Research, VOL. 42, doi:10.1029/2005WR004522, 2006.

Molotch, N.P., and S.A. Margulis, Estimating the distribution of snow water equivalent using remotely sensed snow cover data and a spatially distributed snowmelt model: a multi-resolution, multi-sensor comparison, Advances in Water Resources, 31, 2008.

We agree with the Reviewer that there are several works that exploit Landsat data, which are considered high-resolution data in this context. However, we meant with this sentence that there are no daily high-resolution acquisitions. This is the reason why we consider the work presented by Premier et al., 2021 and the proposed SCA correction based on the state determination as the foundations of the proposed SWE reconstruction. We believe that the use of such daily time-series, which is regularized coherently with the catchment state, represents also an important novelty for the SWE reconstruction. However, we agree with the Reviewer that the section needs to be better rephrased. We propose the following changes:

"To estimate SCA, many works presented in the literature exploit low-resolution (LR) images, since the large swath allows a high repetition time, i.e., with daily or sub-daily acquisitions. This allows the mitigation of the cloud obstruction and proper sampling of the SCA. However, the LR images do not provide sufficient spatial detail on the variability of the snow cover evolution in the mountains, which is on the order of a few dozen meters. Moreover, the use of LR sensors results in a non-linear combination of the different land cover type present within the pixel, and this should be properly considered by the snow classification approaches to avoid large errors, especially in complex terrains. On the other hand, the use of HR snow maps introduces important benefits both in determining SWE as well as in streamflow forecasting (Molotch and Margulis, 2008b; Li et al., 2019). Landsat products were

exploited to retrieve SCA in many works in the literature (e.g., Molotch et al., 2004; Molotch and Bales, 2005, 2006). However, they acquire an image every 16 days (at the equator). With the introduction of the Copernicus Sentinel-2 (S2) mission, HR images are made available with an improved temporal resolution of 5 days (at the equator). This opens up new opportunities to monitor the heterogeneous snow conditions in the mountains. However, due to cloud coverage, useful acquisitions may be reduced by up to 50% in the Alps (Parajka and Blöschl, 2006). Therefore, even if the Landsat images are exploited together with the S2 images, only a few acquisitions are available per month. Recently, we proposed an approach to the reconstruction of daily HR snow cover maps. [...]"

L85 what is this? "Shannon"?

The Nyquist-Shannon theorem establishes the sample rate that allows us to capture information from a continuous signal. In detail, the sample rate must be twice the highest frequency contained in the signal. In this case, we meant that the variation in space of the snow cover (i.e., the continuous signal) must be sampled at least every few meters, especially in complex alpine terrain. To avoid loss of information when using MODIS data, with a resolution of 500 m, the SCF must be considered (i.e., a quantization) although: i) the actual location of the snow/snow free cannot be reconstructed inside the pixel; and ii) the SCF error can be large under certain conditions (e.g., see Aalstad et al., 2020 for quantitative comparison). Vice-versa if the phenomenon is sampled with the correct sample rate (both in time and space), aggregation at a lower resolution is then always possible without losing information.

Even though all modern communication systems are founded on the Nyquist-Shannon theorem, this is a theorem specific to the signal processing field, and therefore to avoid misunderstandings we removed the sentence from the paragraph (see comment above for the revised version of the paragraph).

L104 This assumption needs to be discussed at length in the paper as there are many instances – for example in the Sierra Nevada and other maritime mountain ranges – when the higher elevations are still in the accumulation state whereas the lower elevations are in the ablation state.

We agree with the Reviewer that this represents a strong assumption. By considering the constructive comments received by all the reviewers, we introduced important changes in the method section. We explained that ideally the state should be identified for each pixel (and not at the catchment level anymore). The possible states are (see Figure 1): i) *accumulation* that represents an SWE increase (ΔSWE>0), ii) *ablation* that represents an SWE reduction (ΔSWE<0), and iii) *equilibrium* that represents a stable SWE (ΔSWE=0).

| State | ΔSWE | Class transition t-1 | Class transition t | Description |
|---|---|---|---|---|
| Accumulation | >0 | ■ | ☐ | Snow on bare ground |
| | | ☐ | ☐ | Snow on snow |
| Ablation | <0 | ☐ | ■ | Snowpack disappearance |
| | | ☐ | ☐ | Snowpack reduction |
| Equilibrium | =0 | ☐ | ☐ | Stable snowpack |
| | | ■ | ■ | Bare ground |

Legend: ☐ = snow ■ = snow-free

*Figure 1 Definition of the three possible states: accumulation, ablation and equilibrium. The possible class transitions at pixel level associated with the state are described.*

The phenomena that cause a SWE variation are several, as snowfall, melting, sublimation, human activities or redistribution due to wind or gravitational transport, e.g., avalanches. However, we refer mainly to snowfalls if accumulation and to melting if ablation. In fact, we propose to estimate the SWE to be added considering a quantity proportional to the snow depth/SWE variations, thus, in an ideal case, including only fresh snow as the main driver. Similarly, the amount of SWE to be subtracted is calculated using a DD model and, therefore, it only represents melted snow. Trivially, SWE remains constant when equilibrium.

The state varies pixel-wise due to the topography and meteorology of the study area. However, it is difficult to extrapolate this information with the necessary spatial detail. For the accumulation identification, the sources of information that can be exploited are few. In the paper, we propose that the accumulation can be retrieved by a network of automatic weather stations (AWS) that measure snow depth/SWE. It is important to note that a station is representative of a limited area whose extension is highly variable

depending on the complexity of the terrain. However, by considering only the snowfall events, one can think that from a network of stations distributed with elevation, it is possible to divide the catchment into different elevation belts that can be considered homogenous. However, in many basins, this is quite far from reality. As a common configuration for snow monitoring, we have a single station located at a high point of the catchment, that is informative enough to identify the accumulation events but not their extent. In such a situation, as described in the paper, we considered that the snowfalls occur throughout the snow-covered area of the catchment. We are aware that this assumption may be erroneous, especially in the case of mixed conditions. For example, snowfall may be observed at high elevations, together with rain-on-snow at low elevations that causes snowmelt. However, it has been shown in the literature that the estimation of the snowfall limit may be very challenging (see, e.g., Fehlmann et al., 2018). For this reason, we believe that introducing an approach based on temperature thresholds to define the snowfall limit may still represent a strong simplification that does not necessarily add value to our approach.

On the other hand, the ablation state can be retrieved by using i) temperature index models (it is generally easier to spatialize temperature data w.r.t. snow depth observations), but also ii) multi-temporal SAR observations derived by Sentinel-1. Marin et al. (2020) investigated the relationship between the SAR backscattering and the three melting phases, i.e., the moistening, ripening, and runoff phase. In detail, they showed that if the SAR backscattering is interested in a decrease of at least 2 dB, the snowpack is assumed to get moistened (Nagler and Rott, 2000). First, this decrease affects only the afternoon signal (beginning of the moistening phase). When it also affects the morning signal, the ripening phase starts. Finally, the backscattering increases as soon as the SWE starts to decrease, which corresponds to the beginning of the runoff phase. This moment represents the first contribution of the snowpack to the release of water. The multi-temporal analysis of the SAR backscattering represents a novel way to identify the ongoing melting in a spatialized manner. By integrating this information into a degree-day model, it is possible to exclude false early melting dates.

We included all these explanations in the manuscript. Currently, we are working toward a solution that exploit remote sensing information only i.e., natively spatialized information (as already mentioned in the manuscript, for example by considering surface temperature measurements from satellites, meteorological radar or SAR derived snow depth/SWE information). However, this needs further research and it left as a future development in the current manuscript.

L157 this should be discussed in that reliance on in situ data has issues in terms of transferability to areas without ground measurements – this is often a motivator behind classic SWE reconstruction approaches (as cited herein) which are designed to be widely transferable and independent of in situ observations.

Thank you for this comment. This part has been better discussed in the paper. In fact, we believe that working on a robust method for the determination of the accumulation state is one of the most attractive future developments of the proposed approach. Notwithstanding the approach depends on two information acquired by the in-situ stations: i) temperature, to determine the potential melting; and ii) snow depth/SWE variations, to identify the accumulation state. However, we are not considering a spatialized snow depth/precipitation input as commonly done for physical snow models, but only an indication of the snow depth variation, thus representing an advantage. As discussed in the previous answer, the accumulation state could be derived by exploiting remote sensing data, but this requires further research. Similar to classic SWE reconstruction, our proposed approach is dependent on temperature data that need to be spatialized. Therefore, this represents not only a limit for transferability but also a possible source of error. As discussed already in the current version of the paper (L497-500), S1 represents a spatialized way to identify the melting, but it presents as a major disadvantage a poor temporal resolution, i.e., a few days. As a future development, we aim at exploiting S1 and the new HR land surface temperature acquired by the next Sentinel generation as a proxy of the potential melting estimation, thus enlarging the applicability of the proposed method in remote areas.

L171 It has been shown that when the backscattering is interested by a decrease of at least 2 dB, the snowpack is assumed to get moisted

Thank you for highlighting this grammar mistake.

L181 quotes

Thank you for highlighting this mistake. We meant elevations. This mistake was corrected throughout the manuscript.

L211 so does this mean there is no "viewable gap fraction" correction? If so, that is problematic.

Yes, there is no "viewable gap fraction" correction since the classification algorithm that we are using proposed by Barella et al., 2022 is meant to detect viewable snow. We know this is problematic, and this is the reason why we encountered the problem mentioned in the old version of the manuscript in L 207-209. In detail, we refer to an underestimation of snow presence in forested

areas when the snow falls below the canopy and is no longer visible from the satellite point of view, i.e., snow on ground. However, we have proposed an SCA regularization that has the potential to detect snow under canopy (see Section 2.2 and in detail, Algorithm 1 which is discussed in the text L224-259). In fact, while an HR pixel may not show snow (on the ground) due to the dense canopy presence, it is likely that an LR pixel intercepts some open areas surrounded by forest that are covered by snow. In this sense, we proposed to infer the snow presence under canopy not through a correction (as it is traditionally done through a VGF correction that however still represents a hot research topic) but by considering what is happening in the surrounding area (from the LR acquisition) and by analyzing the multi-temporal snow presence (SCA regularization). In this way, the presence of snow on ground is detected in a relatively robust manner as demonstrated by the obtained results.

L305 The South Fork of what river, the San Joaquin? Many of us know our Geography of this region quite well but this nomenclature doesn't seem correct.

Thank you for letting us know the correct nomenclature. In fact, we mean the South Fork catchment of the San Joaquin river. We apologize for the confusion in the nomenclature, but we do not know the region well. We select this region for the availability of the high-quality spatialized reference data provided by ASO, which we believe will make this area famous in the snow hydrology community in the future.

L317 why use the German name when the catchment is in Italy and the journal language is English.

We agree that this may sound weird, but the province where Senales/Schnals is located is a bilingual province. Formally, we can use either German or Italian toponymy, (which was introduced after the first World War). It should be mostly complete, and politically correct, to add both names, but for brevity, we chose the German one since it is the most spread around Europe.

L332 how accurate is this algorithm for SCA detection and how do / will errors in the algorithm propagate in the model to cause errors in SWE estimation?

The algorithm for SCA detection, which is described in detail in Barella et al., 2022, presents an RMSE of 22.82 and an MBE of 6.95 when compared with a set of three reference snow maps derived at 1 m resolution over the European Alps during winter 2021 (covering approximately 250km2).

To evaluate the propagation of errors related to SCA detection, we conducted a simplified analysis as suggested by Reviewer#1. For the sake of clarity, we considered only the pixel where the station Volcanic Knob provides continuous SWE measurements in the Sierra Nevada catchment. The test is carried out for one season only (2018/19). While we are aware that this analysis is not exhaustive, it can provide an overview of the error propagation.

In detail, errors in the SCA detection influence the time of snow disappearance (tSD) and the time of snow appearance (tSA). To see the effects on the two parameters separately, one is kept constant, and the other is varied.

[Figure]

*Figure 2 tSA=22/11/2018, tSD 27/06/2019 +- 15 days.*

It is possible to notice in Figure 2 that both underestimating and overestimating tSD introduce important errors in the reconstruction. In fact, at the end of the melting season the temperature is high and consequently the potential melting. A difference of +-5 days (which corresponds to the S2 repetition time) already introduces around 50 mm of RMSE.

[Figure]

*Figure 3 tSD 27/06/2019, tSA=22/11/2018 +- 15 days.*

It is possible to see in Figure 3 that the shift of tSA does not strongly affect the RMSE as tSD does. For negative shifts, the accuracy RMSE is constant since no SWE is added to the reconstruction. In fact, for those days, we find that the coefficient k (see Eq. 4) is 0 since it is calculated from the AWS. In other words, it means that the accumulation is not really happening before at least one station detects an increase in SWE.

In addition, to complete the sensitivity analysis we also tested the sensitivity of the method to the degree day factor a, the SWE threshold used to identify the states and the time of first ablation detected by S1 (we call it here tS1). These parameters were kept constant in the previous cases (a=4.8 mm/(°Cd), SWE threshold 2 mm and tS1=22/04/2019).

[Figure]

*Figure 4 tSD = 27/06/2019, tSA=22/11/2018, tS1=22/04/2019 (as detected by the station), SWE threshold=2mm, a varies from 3 to 6 mm/(°Cd) by steps of 0.2.*

It is possible to notice in Figure 4 that the error increases linearly when a moves away from the optimal value, that is a=4.5 (as it was set in the manuscript).

[Figure]

*Figure 5 tSD = 27/06/2019, tSA=22/11/2018, tS1=22/04/2019 (as detected by the station), a=4.8 mm/(°Cd), SWE threshold varies from 0 to 20 mm by steps of 1 mm.*

It is possible to see in Figure 5 that, as expected, the higher the threshold the greater the error. In fact, for too large thresholds, the method fails to detect the accumulation states. A snow threshold of 2 mm, as set in the manuscript, is acceptable.

[Figure]

*Figure 6  tSD = 27/06/2019, tSA=22/11/2018, a=4.8 mm/(°Cd), SWE threshold 2 mm, tS1=22/04/2019 +- 15 days.*

In this case, it is also possible to see in Figure 6 that the shift of tS1 does not strongly affect the RMSE as does tSD. The RMSE for negative shifts remains constant after a certain point, since for those days the DD model returns 0 potential melting, so there are no differences. This means that it is in general better to make an error anticipating the melting phase than postponing it.

Even though we are aware that this represents a very simplified analysis and might be not exhaustive, we can summarize that we expect that the error that most strongly affects the results is a shift in the date of snow disappearance. For this reason, we believe that the SWE reconstruction can fully benefit from the introduction of an accurate daily HR time series.

L354 I agree that this may be quite important but to really show this you would want to show a result with course resolution data and then another result when adding in the high resolution data to show the relative improvement associated with the addition of the high resolution data.

We thank the Reviewer for pointing out this important aspect of the proposed approach. The sentence at L354 points out the good agreement in the geometrical details between the SWE maps derived by the proposed approach and the reference ASO products, which have a comparable spatial resolution. Having such a reference dataset allows us to evaluate how the algorithm performs at that specific spatial detail. We agree with the reviewer that this is not enough to show the relative improvement associated with the use of HR data. However, quantify the improvement in a rigorous manner may be difficult since different source of errors are difficult to be isolated when using LR or HR images separately. First, a proper comparison should preferably refer to the obtained SCA time-series rather than the SWE time-series in order to isolate the errors coming from the reconstruction method. Having saying that, several works have already shown the benefits introduced using HR data (e.g., Aalstad et al., 2020 for citing the most recent). In fact, SCF retrieved by LR sensors presents some intrinsic limitations, e.g., variable viewing angle that changes the spatial resolution inside the image, high heterogeneity of land cover, illumination and atmospheric conditions inside the resolution cell especially over mountainous areas (see e.g., Rittger et al., 2016).  The development of a robust and accurate algorithm to SCF retrieval able to address all the aforementioned problems has been the main research topic of the last 20 years. On the other hand, as showed in our recent work (Premier et al., 2021), it is possible to reconstruct a daily HR SCA time-series that presents high accuracy by learning from the historical snow patterns that repeat inter-annually. In this work, we used the information provided by LR sensors more as an indication than as an absolute truth, being aware that it presents an uncertainty i.e., we added an uncertainty value to the retrieved LR SCF (see (Premier et al., 2021), for more details). In this sense, the LR SCF is also corrected during the process. Hence, when using the corrected daily LR dataset to reconstruct the SWE we obtain results comparable with the ones derived by the high resolution approach presented in this paper and aggregated at 500m. In other words, if the LR time series is accurate and filtered out from the above-mentioned problem, the real benefits provided by the time series at HR is only an incontrovertible geometric detail, which might be of paramount importance for some applications (e.g., hydrology, avalanche forecasting, ecological studies, etc) but does not change the total amount of reconstructed SWE. This is also in line with the findings of Bair et al., 2022.  It is finally worth stressing the fact that also the different temporal resolution plays an important role in the final evaluation of the SWE reconstruction. Therefore, considering partial time series (e.g., only HR images) instead of the

completed ones (e.g., HR and LR images) can introduce artificial errors that do not allow one to properly quantify the advantage of using LR or HR data in the SWE reconstruction.

However, we take the opportunity raised by this comment to evaluate the results obtained by the proposed approach against another dataset that is available at 500 m resolution (Fang et al., 2022), but that is derived by assimilating Landsat data. The metrics obtained for the two methods are shown in the Table. Also, the figures report the results for the three hydrological seasons.

| Date | BIAS [mm] | | RMSE [mm] | | Correlation [-] | |
|---|---|---|---|---|---|---|
| | proposed | NASA | proposed | NASA | proposed | NASA |
| 17/03/2019 | -121 | 36 | 242 | 292 | 0.80 | 0.66 |
| 02/05/2019 | -61 | -4 | 208 | 307 | 0.90 | 0.77 |
| 09/06/2019 | -25 | -32 | 182 | 302 | 0.93 | 0. 79 |
| 04/07/2019 | -49 | -14 | 129 | 201 | 0.90 | 0.71 |
| 14/07/2019 | -51 | -20 | 125 | 163 | 0.84 | 0.65 |
| 15/04/2020 | -73 | -26 | 159 | 169 | 0.80 | 0.78 |
| 05/05/2020 | -59 | 5 | 151 | 154 | 0.82 | 0.80 |
| 23/05/2020 | -95 | -27 | 179 | 150 | 0.79 | 0.78 |
| 08/06/2020 | -25 | 3 | 96 | 100 | 0.72 | 0.70 |
| 26/02/2021 | 5 | 96 | 92 | 157 | 0.75 | 0.65 |
| 31/03/2021 | 74 | 119 | 124 | 202 | 0.85 | 0.72 |
| 03/05/2021 | 65 | 54 | 121 | 121 | 0.89 | 0.66 |

[Figure]

*Figure 7 Results for the South Fork catchment of the San Joaquin river for the h.y. 2018/19.*

[Figure]

*Figure 8 Results for the South Fork catchment of the San Joaquin river for the h.y. 2019/20.*

[Figure]

*Figure 9 Results for the South Fork catchment of the San Joaquin river for the h.y. 2020/21.*

It is possible to see that the two time-series show a similar trend. Generally, we encounter lower BIAS for the NASA product while lower RMSE and higher correlation are observed for our proposed product when compared with ASO data.

Table 1 percent bias would also be helpful

Thank you for this suggestion. We added this information in the revised version of the manuscript. The new table is also reported here. The metrics slightly differ from the previous version, since during the time of the revision we improved the accuracy of the snow cover maps for this new version. Furthermore, according to the feedback provided by Reviewer#1, we decided to consider the runoff onset instead of the moistening onset ad date of first possible ablation.

| Date | BIAS [mm] | PBIAS [%] | RMSE [mm] | Correlation [-] | SWE tot ASO [mm] | SWE tot proposed [mm] |
|---|---|---|---|---|---|---|
| 17/03/2019 | -82 | -11 | 315 | 0.73 | 778 | 699 |
| 02/05/2019 | -33 | -5 | 299 | 0.84 | 679 | 647 |
| 09/06/2019 | -7 | -1 | 268 | 0.88 | 511 | 504 |
| 04/07/2019 | -37 | -24 | 185 | 0.86 | 177 | 143 |
| 14/07/2019 | -41 | -58 | 164 | 0.80 | 106 | 67 |
| 15/04/2020 | -71 | -21 | 242 | 0.63 | 389 | 321 |
| 05/05/2020 | -53 | -26 | 224 | 0.67 | 250 | 199 |
| 23/05/2020 | -91 | -112 | 235 | 0.65 | 166 | 78 |
| 08/06/2020 | -18 | -58 | 161 | 0.66 | 48 | 30 |
| 26/02/2021 | 12 | 6 | 125 | 0.67 | 186 | 198 |
| 31/03/2021 | 81 | 29 | 168 | 0.73 | 191 | 271 |
| 03/05/2021 | 74 | 50 | 160 | 0.80 | 72 | 145 |

400 what is meant by this sentence? I think there are many arguments to avoid a degree day model so I am curious as to the point being made here but I do not find the statement to be decipherable.

We meant with this sentence that we have two factors that affect the SWE determination, i.e., the degree day that we use to calculate the potential melting and also the snow duration on the ground. If for example, two close pixels have a very similar temperature but differ in aspect, the resulting potential melting does not differ as it is calculated based on temperature only. However, the different snow persistence on the ground will result in a different amount of SWE for those pixels.

Figure 9 two points: first: what are the units "Gt" on the vertical axis. Second: I do not think it is appropriate to use the wrt abbreviation.

Gt is gigatons. However, according to Reviewer#1 and Community Comment#1, we replaced the unit with mm (see also Table above). We also removed the abbreviation.

Figure 10 Respectfully, I think it is problematic to only show "some locations". There should be more transparency around the model performance for ALL available evaluation data.

We agree with what the Reviewer says. In fact, the metrics were reported for all the locations (L396). The purpose of selecting a few points in Fig. 10 was more for visualization reasons and to show that different point correctly present different trends. However, we added a scatterplot (Figure 10) with all the observations for completeness.

[Figure]

*Figure 10 Observed and proposed SWE in the Schnals catchment for the h.y. 2020/21.*

L418 I think the very lengthy section (early in the paper) on defining the accumulation, equilibrium, or ablation state would be greatly improved if more context were given to the reader about the relevance/importance of that procedure. For example, it is a big difference from previous reconstruction works, and thus it requires a delicate and well-organized presentation - and some emphasis placed on why it is so important in light of the differences of your modeling approach relative to other SWE reconstruction approaches.

Thank you for this suggestion. According to the comment of Reviewer#3, we stressed the importance of the state already in the introduction: "The concept of snow state of a pixel is also introduced in this work, representing the direction of the SWE change for that pixel, i.e., accumulation, ablation or equilibrium. This allows SWE reconstruction without the need for spatialized precipitation data. In fact, the method redistributes the amount of melting by exploiting the information about the state rather than quantifying the precipitation input. Moreover, the state allows us to set up the novel regularization of the SCA time-series. "

Furthermore, we revised Section 2.1 in the new version of the manuscript as described in the previous answer by defining the state at a pixel level and not at a catchment level. We believe that this makes the concept more understandable.

L421 interested

Thank you for highlighting this grammar mistake.

L454 i am not sure that the methods section is clear enough about the regularization (for example the "majority rule") such that this discussion is meaningful. I think the methods section needs work to make this clearer. Perhaps by providing some clear examples.

Thank you for this suggestion. We have revised the method section, as explained in the previous answer. We hope that the state definition at pixel level will make the method clearer. However, for sake of clarity, we would like to report here some examples, that are related to the most frequent errors that we observed from our analysis of the obtained results: i) an underestimation of snow presence in forested areas when the snow falls below the canopy and is no longer visible from the satellite point of view, i.e., snow on ground, and ii) the missed identification of snow-patches at the end of the season. Please, note that in these examples the equilibrium is omitted but, in that case, both transitions are not possible.

Let's consider the first case (i.e., transition snow to snow-free is not allowed). A pixel at time *t* has been erroneously classified as snow free when its state is accumulation.

[Figure]

In this example, the last previous accumulation happened at t-1, so we are in the case of a "recent snowfall". According to the proposed method we check the ±5 adjacent dates and correct the pixel by counting for the most frequent label, which is snow. So we corrected the classification to snow. The possible reasons of this error can be due to a misclassification of snow under canopy or snow in shadow, or an error introduced during the SCA reconstruction.

In the second example (i.e., transition snow to snow-free is not allowed see figure below), the most frequent label is snow-free. In this case, we will replace *t-1* with snow-free.

[Figure]

In the third example (see figure below), we need to replace not only t-1 as snow-free, but also t-2. In fact, according to the proposed correction algorithm, to ensure coherence the replacement has to be done until the date of the last previous ablation.

[Figure]

Let's look now at a more complex example, to understand what may happen during the main ablation phase. You can notice that the time-series presents some snow dates in correspondence of HR acquisition (t=7, t=11, t=15) while the other dates are snowfree. This is a typical error since in the SCA reconstruction, the LR may fail to detect the snow patches i.e., low SCF. By going through the example, let's start from t=0, when the last accumulation happens and then only ablation follows for all the remaining dates. At t=5, it looks like the snow is disappearing. Since we are in ablation, this is coherent with the state. However, at t=7 we have a transition snow-free to snow that is not allowed. At t=7 we are still in the case of a recent snowfall, so by applying the rule that we set up, we obtain that the majority of the labels is snow. Hence, t=6 will be set as snow. The next incorrect transition is t=11. Now we are in the case of an old snowfall. If we consider the time window ±5 days, we would observe a majority of snow-free pixels. But in this case, we would make a mistake, since the LR acquisition is not able to correctly see the snowpatches as said before. Hence, we proposed in our correction algorithm, to consider only the HR acquisitions available from the last accumulation (t=0 in this case), up to a maximum number of 5 pixels, and count the occurrences of snow labels. In this case, this results in a majority of snow pixels, so we replace t=8, t=9, t=10 with snow. The same procedure is repeated when we encounter t=15. We proposed not to consider the HR acquisitions after the time to be corrected in the majority rule since the snow patches may disappear quickly.

[Figure]

These examples are not exhaustive but are provided to better understand the proposed approach.

It is finally stressing the fact that these empirical rules have been introduced to cope with the lack of optical observations acquired with appropriate spatial and temporal resolution to sample the snow cover. However, it is interesting to note how simple rules together with snow state knowledge can be used to try to correct a SCA time series.

L486 i think it would be appropriate to discuss the ways in which a DD model might insufficiently characterize potential melt. E.g. a uniformly applied degree day model (and associated melt coefficient) would underestimate melt for south facing slopes and would over-estimate melt for north facing slopes and this would result in corresponding biases in reconstructed SWE. The topic of discussion would need to consider the issue of how well the melt coefficient mathematically transposes degree-days into melt flux relative to all the terms of a physically based melt calculation (i.e. all radiative and turbulent fluxes). Hence, spatial variability in solar radiation, albedo, wind speed, vapor pressure, temperature, atmospheric and surface emissivity - these would all be relevant to this discussion and have been covered / discussed in many SWE reconstruction papers (notably those authored or co-authored by Bair, Dozier, Rittger, Molotch, etc).

We thank the reviewer for this suggestion. We will deepen the discussion on this topic. However, we did not notice from our analysis patterns in the bias maps that can be attributed to this issue. This is also the reason why we did not investigate the use of other modified temperature index models. Please, refer to Figures 11-22 that we produced as suggested by Reviewer#1. The figures are similar to Figure C2 but with SWE expressed in mm. It is possible to notice that we usually can observe an underestimation for North facing slope when comparing our product with ASO. This is not in line with the correct observation of the reviewer that we would expect an overestimation for North exposed area since we are using a constant degree day model. Hence, we cannot directly ascribe the errors that we observe to the use of a degree day model.

[Figure]

[Figure]

*Figure 11 17/03/2019.*

*Figure 12 02/05/2019.*

[Figure]

[Figure]

*Figure 13 09/06/2019.*

*Figure 14 04/07/2019.*

[Figure]

*Figure 15 14/07/2019.*

[Figure]

*Figure 16 15/04/2020.*

[Figure]

*Figure 17 05/05/2020.*

[Figure]

*Figure 18 23/05/2020.*

[Figure]

*Figure 19 08/06/2020.*

[Figure]

*Figure 20 26/02/2021.*

[Figure]

*Figure 21 31/03/2021.*

[Figure]

*Figure 22 03/05/2021.*

**References**

Aalstad, K., Westermann, S., & Bertino, L. (2020). Evaluating satellite retrieved fractional snow-covered area at a high-Arctic site using terrestrial photography. Remote Sensing of Environment, 239, 111618.

Bair, E. H., Dozier, J., Rittger, K., Stillinger, T., Kleiber, W., & Davis, R. E. (2022). Does higher spatial resolution improve snow estimates?. The Cryosphere Discussions, 1-20.

Barella, R., Marin, C., Gianinetto, M., & Notarnicola, C. (2022, July). A novel approach to high resolution snow cover fraction retrieval in mountainous regions. In IGARSS 2022-2022 IEEE International Geoscience and Remote Sensing Symposium (pp. 3856-3859). IEEE.

Fang, Y., Y. Liu, and S. A. Margulis. (2022). Western United States UCLA Daily Snow Reanalysis, Version 1 [Data Set]. Boulder, Colorado USA. NASA National Snow and Ice Data Center Distributed Active Archive Center. https://doi.org/10.5067/PP7T2GBI52I2. Date Accessed 12-12-2022.

Fehlmann, M., Gascón, E., Rohrer, M., Schwarb, M., & Stoffel, M. (2018). Estimating the snowfall limit in alpine and pre-alpine valleys: A local evaluation of operational approaches. Atmospheric Research, 204, 136-148.

Li, D., Lettenmaier, D. P., Margulis, S. A., & Andreadis, K. (2019). The value of accurate high-resolution and spatially continuous snow information to streamflow forecasts. Journal of Hydrometeorology, 20(4), 731-749.

Marin, C., Bertoldi, G., Premier, V., Callegari, M., Brida, C., Hürkamp, K., ... & Notarnicola, C. (2020). Use of Sentinel-1 radar observations to evaluate snowmelt dynamics in alpine regions. The Cryosphere, 14(3), 935-956.

Molotch, N. P., Painter, T. H., Bales, R. C., & Dozier, J. (2004). Incorporating remotely-sensed snow albedo into a spatially-distributed snowmelt model. Geophysical Research Letters, 31(3).

Molotch, N. P., & Bales, R. C. (2005). Scaling snow observations from the point to the grid element: Implications for observation network design. Water Resources Research, 41(11).

Molotch, N. P., & Bales, R. C. (2005). Scaling snow observations from the point to the grid element: Implications for observation network design. Water Resources Research, 41(11).

Molotch, N. P., & Margulis, S. A. (2008). Estimating the distribution of snow water equivalent using remotely sensed snow cover data and a spatially distributed snowmelt model: A multi-resolution, multi-sensor comparison. Advances in water resources, 31(11), 1503-1514.

Nagler, T., & Rott, H. (2000). Retrieval of wet snow by means of multitemporal SAR data. IEEE Transactions on Geoscience and Remote Sensing, 38(2), 754-765.

Parajka, J., & Blöschl, G. (2006). Validation of MODIS snow cover images over Austria. Hydrology and Earth System Sciences, 10(5), 679-689.

Premier, V., Marin, C., Steger, S., Notarnicola, C., & Bruzzone, L. (2021). A Novel Approach Based on a Hierarchical Multiresolution Analysis of Optical Time Series to Reconstruct the Daily High-Resolution Snow Cover Area. IEEE Journal of Selected Topics in Applied Earth Observations and Remote Sensing, 14, 9223-9240.

**Answer to the Reviewer #3 – Manuscript tc-2022-146**

This paper presents a reconstruction method to produce time series of high-resolution (HR) snow water equivalent (SWE) maps from different sources of remotely sensed data, with specific focus on the applicability on mountain areas. The method's description and results in two selected catchments are discussed, and the potential sources of errors are addressed in the context of general applicability and future improvements.

The topic is relevant for the scientific community in snow-dominated areas due to the lack of HR mapping on a time scale that allows monitoring of quick changes in the snowpack extension and mass change, and the scarcity of direct SWE measurements and methods to provide a dense network of monitoring stations for spatial interpolation approaches. The innovation and relevance of these objectives are sufficiently addressed in the manuscript. However, in its current version, some issues are found that require further assessment before considering further review and potential publication in this journal. Please, find below these major items. I hope that these comments are useful to improve the work and help to further comprehend its context and applicability further than the present results.

The authors thank the Reviewer for his/her constructive feedbacks and comments on the manuscript. We went through each point and took advantage of the comments to improve the quality of the manuscript. Our answers are reported in blue.

- The Introduction section contains good points but requires some structure to get more focused on the specific goals' context. I would also recommend to present this earlier in the narrative. Lines 100-108 can be easily moved/merged to/with section 2 for the sake of clarity.

  Thank you for your suggestion. By taking into account also the comments provided in the open discussion, we structure the introduction as follows: first, we talk about the importance of snow for hydrological applications and why it is important to get SWE maps. Then we introduce the available methods to monitor snow and, among them, we present remote sensing as an alternative that provide natively spatialized observations. In detail, we explain that it is not possible to derive SWE estimates regularly from the current satellite missions. However, important information about the melting phases can be extracted by multitemporal SAR images and snow cover depletion curves from optical sensors. Hence, we propose to exploit this information sources to retrieve SWE.

  We propose to modify the introduction accordingly in the revised version of the manuscript.

- The general objective should be better elaborated in line 99, i.e. not only state what but also what for and some specific scope. For example, the target type of catchment is relevant but it is not declared until lines 130-131 that size is limiting the potential further applicability of the method. Moreover, the order of magnitude of "a not too vast catchment" must be assessed.

  We thank the reviewer for pointing out this issue. We would like to explain better that the approach is not necessarily designed for a catchment. This will be better stated in the new version of the manuscript, where we introduce the state concept at a pixel level and not at a catchment level (see next point). However, we make the necessary assumption to consider the accumulation happening for the entire area of interest. In fact, in our study cases, we lack information to consider better detail. In this sense, considering a "too vast" catchment would make this already strong assumption much less reliable.

  In fact, this assumption works (as also shown by our obtained results) when the catchment is subjected to similar meteorological forcings. Hence, rather than suggesting a perfect size we suggest analyzing the climatic and hydrologic characteristics of each subcatchment when dealing with a large area, i.e., to check if a correlation exists among the subcatchments in terms of climatic variables (temperature, precipitation, discharge). This suggestion has been reported in the manuscript.

  In this way we propose to rephase L99 as follows: "This work proposes a new multi-source approach to reconstruct HR daily SWE time-series at the pixel scale. We explore the applicability of the method to two mountainous catchments: i) the South Fork catchment of the San Joaquin river, located in the Sierra Nevada - California (USA), and ii) the Schnals catchment, located in the Alps - South Tyrol (Italy). The main sources of error are discussed to provide insights about the

main advantages and disadvantages of the approach, which may be of great interest for several hydrological and ecological applications."

- Section 2.1 is determinant in the methodological approach. In the explanation, it is not clear whether the catchment state is identified for each pixel or for the whole catchment area; this needs a revision to be clear throughout the text. Moreover, the spatial definition of the "total delta-SWE" is missing, which is required, and additionally the use of this variable should be uniform for the three states (i.e. is also total in line 149?). In line 149, I am not sure about the meaning of "no changes WITHIN the catchment", do you mean really that or rather no change when considered as a whole?

Thank you for pointing out this aspect that may generate misunderstandings. By considering the constructive comments received by all the reviewers, we introduced important changes in the method section. We explained that ideally the state should be identified for each pixel (and not at the catchment level as done in the previous version of the manuscript). The possible states are (see Figure 1): i) accumulation that represents an SWE increase (ΔSWE>0), ii) ablation that represents an SWE reduction (ΔSWE<0), and iii) equilibrium that represents a stable SWE (ΔSWE=0).

| State | ΔSWE | Class transition t-1 | Class transition t | Description |
|---|---|---|---|---|
| Accumulation | >0 | ■ | □ | Snow on bare ground |
| | | □ | □ | Snow on snow |
| Ablation | <0 | □ | ■ | Snowpack disappearance |
| | | □ | □ | Snowpack reduction |
| Equilibrium | =0 | □ | □ | Stable snowpack |
| | | ■ | ■ | Bare ground |

Legend: □ = snow ■ = snow-free

*Figure 1 Definition of the three possible states: accumulation, ablation and equilibrium. The possible class transitions at pixel level associated with the state are described.*

The phenomena that cause a SWE variation are several, as snowfall, melting, sublimation, human activities or redistribution due to wind or gravitational transport, e.g., avalanches. However, we refer mainly to snowfalls if accumulation and to melting if ablation. In fact, we propose to estimate the SWE to be added considering a quantity proportional to the snow depth/SWE variations, thus, in an ideal case, including only fresh snow as the main driver. Similarly, the amount of SWE to be subtracted is calculated using a DD model and, therefore, it only represents melted snow. Trivially, SWE remains constant when equilibrium.

The state varies pixel-wise due to the topography and meteorology of the study area. However, it is difficult to extrapolate this information with the necessary spatial detail. For the accumulation identification, the sources of information that can be exploited are few. In the paper, we propose that the accumulation can be retrieved by a network of automatic weather stations (AWS) that measure snow depth/SWE. It is important to note that a station is representative of a limited area whose extension is highly variable depending on the complexity of the terrain. However, by considering only the snowfall events, one can think that from a network of stations distributed with elevation, it is possible to divide the catchment into different elevation belts that can be considered homogenous. However, in many basins, this is quite far from reality. As a common configuration for snow monitoring, we have a single station located at a high point of the catchment, that is informative enough to identify the accumulation events but not their extent. In such a situation, as described in the paper, we considered that the snowfalls occur throughout the snow-covered area of the catchment. We are aware that this assumption may be erroneous, especially in the case of mixed conditions. For example, snowfall may be observed at high elevations, together with rain-on-snow at low elevations that causes snowmelt. However, it has been shown in the literature that the estimation of the snowfall limit may be very challenging (see, e.g., Fehlmann et al., 2018). For this reason, we believe that introducing an approach based on temperature thresholds to define the snowfall limit may still represent a strong simplification that does not necessarily add value to our approach.

On the other hand, the ablation state can be retrieved by using i) temperature index models (it is generally easier to spatialize temperature data w.r.t. snow depth observations), but also ii) multi-temporal SAR observations derived by Sentinel-1. Marin et al. (2020) investigated the relationship between the SAR backscattering and the three melting

phases, i.e., the moistening, ripening, and runoff phase. In detail, they showed that if the SAR backscattering is interested in a decrease of at least 2 dB, the snowpack is assumed to get moistened (Nagler and Rott, 2000). First, this decrease affects only the afternoon signal (beginning of the moistening phase). When it also affects the morning signal, the ripening phase starts. Finally, the backscattering increases as soon as the SWE starts to decrease, which corresponds to the beginning of the runoff phase. This moment represents the first contribution of the snowpack to the release of water. The multi-temporal analysis of the SAR backscattering represents a novel way to identify the ongoing melting in a spatialized manner. By integrating this information into a degree-day model, it is possible to exclude false early melting dates.

We included all these explanations in the manuscript. Currently, we are working toward a solution that exploit remote sensing information only i.e., natively spatialized information (as already mentioned in the manuscript, for example by considering surface temperature measurements from satellites, meteorological radar or SAR derived snow depth/SWE information). However, this needs further research and it left as a future development in the current manuscript.

- I have doubts on the simplification done on the potential combinations of positive/zero/negative values of delta-SWE and delta-SCA in this section. First, it seems that both variables have different spatial definitions, since pixel changes in terms of SCA are assessed. Additionally, some situations are discarded, for example, accumulation is not allowed to happen with negative delta-SCA values, but this is not infrequent in mountain areas in some regions in the world. Other situations are not included in the three potential states. In general, the assumptions are difficult to be validated in semiarid regions with snow relevance or during patchy snow periods in steep slopes, especially if the catchment state is defined uniformly in space. These issues should have been assessed and their discarding justified or at least clarified in terms of the applicability of the method.

As stated in the previous answer, we hope that our revised version will solve these doubts. Notwithstanding, we agree with the Reviewers observation that the assumptions made cannot work for each situation. A possible situation is represented by a snowfall happening at high elevations while lower elevations experience melting due to rain on snow or higher temperature. If we are aware that this represents a common situation in our study area, we suggest either dividing the catchment into different elevation belts if more AWS are present or implementing a method to detect the snowfall limit (e.g., based on temperature). This limitation has been better highlighted in the discussion section.

- In section 2.3, two issues require further assessment. First, the use of day-degree modelling for melting rates' estimation is not the best choice if accurate HR maps are the goal, in my opinion. At least, some justification of this apparent lack of coherence should be included, together with the comparison of the error of SWE estimation associated to the use of such methods and the error from low resolution satellite products. Secondly, the adoption of the temperature threshold is one of the major sources of error in the SWE estimation in mountain areas, as many works have already shown; so, the selected value needs some justification. Thirdly, and more relevant, lines 280-283 involve that melting is the only process in the ablation of the snowpack, which means that sublimation is neglected (but nothing is said on this); this may result in non-negligible loss of mass in the closure of the balance equation, and it is a constraint for the applicability of the method in some regions or during some periods/under some atmospheric conditions. This must be addressed in the description of the methodological assumptions and their validity. Finally, some comments on the scale effects from the subdaily evolution, not operating in the method, should be included.

We thank the Reviewer for giving us the opportunity to better explain these relevant concepts and better constrain the applicability conditions of the method in real situations.

Regarding the use of a degree-day model, we agree with the referee that there is an apparent lack of coherence in targeting HR SWE maps and using a DD model. We are also aware that the temperature threshold is a critical parameter that can introduce errors in the method. This has been clearly stated in the new version of manuscript citing some literature works that can stimulate further research (e.g., Tobin et al., 2013). We agree that a full energy budget model is required to model in an appropriate way the snowpack. However, we decided to use a simple DD model because our aim is to propose an approach as simple and independent form detailed meteorological observations as possible, which maximizes the information content of the remote sensing data. Moreover, since our target is SWE reconstruction, we need a model which can be easily inverted in time, unlike complex energy-based models that involve the solution of differential equations. However, the same approach that we used to define the pixel state could be used also with more sophisticated snow models.

Despite these assumptions, we were also surprised by the goodness of obtained results using this simple model. However, the HR observations of daily SCA and the timing of melting derived by SAR data provide a good modulation of the potential melting that allows us to estimate SWE with high geometrical detail. Please note that in the new version of the paper we also added the results expressed in mm to better analyze the behavior with respect to the topography (see Figures 2-13). In the figures below, it is possible to note differences, especially for steep slopes, where the proposed method underestimates SWE w.r.t. ASO. However, we would expect less SWE for these steeper slopes that promote gravitational transport. This point can be better investigated and discussed in the revised version of the manuscript.

[Figure]

*Figure 2 17/03/2019.*

[Figure]

*Figure 3 02/05/2019.*

[Figure]

*Figure 4 09/06/2019.*

[Figure]

*Figure 5 04/07/2019.*

[Figure]

*Figure 6 14/07/2019.*

[Figure]

*Figure 7 15/04/2020.*

[Figure]

*Figure 8 05/05/2020.*

[Figure]

*Figure 9 23/05/2020.*

[Figure]

*Figure 10 08/06/2020.*

[Figure]

*Figure 11 26/02/2021.*

[Figure]

*Figure 12 31/03/2021.*

[Figure]

*Figure 13 03/05/2021.*

We also performed a sensitivity analysis, as suggested also by Reviewer#1 to assess the impact of the DD model to the final results. It might be difficult to really estimate all the uncertainty sources and provide a proper sensitivity analysis on the parameters that play a role in the proposed approach. However, we follow the suggestion of the Reviewer and we carried out a simplified sensitivity analysis. For the sake of clarity, we propose to investigate how the parameters affect the final SWE reconstruction by considering the pixel where the station Volcanic Knob provides continuous SWE measurements in the Sierra Nevada catchment. The parameters that we believe play an important role in the methodology are i) the degree day factor a, ii) the SWE threshold used to identify the states, iii) the time of snow disappearance (tSD), iv) the time of snow appearance (tSA), and v) the time of first ablation detected by S1 (we call it here tS1). We vary each of these parameters separately keeping the others constant and equal to the optimal case (i.e., the one with the lowest RMSE). The test is carried out for one season (2018/19). Although we are aware that this analysis is not exhaustive, it can give an overview of the most important sources of error.

[Figure]

*Figure 14 tSD = 27/06/2019, tSA=22/11/2018, tS1=22/04/2019 (as detected by the station), SWE threshold=2mm, a varies from 3 to 6 mm/(°Cd) by steps of 0.2.*

It is possible to notice in Figure 14 that the error increases linearly when a moves away from the optimal value, that is a=4.5 (as it was set in the manuscript).

[Figure]

*Figure 15 tSD = 27/06/2019, tSA=22/11/2018, tS1=22/04/2019 (as detected by the station), a=4.8 mm/(°Cd), SWE threshold varies from 0 to 20 mm by steps of 1 mm.*

It is possible to see in Figure 15 that, as expected, the higher the threshold, the greater the error. In fact, for too large thresholds, the method fails to detect the accumulation states. A snow threshold of 2 mm, as set in the manuscript, is acceptable.

[Figure]

*Figure 16 tSA=22/11/2018, tS1=22/04/2019 (as detected by the station), a=4.8 mm/(°Cd), SWE threshold 2 mm, tSD 27/06/2019 +- 15 days.*

It is possible to notice in Figure 16 that both underestimating and overestimating tSD introduce important errors in the reconstruction. In fact, at the end of the melting season the temperature is high and consequently the potential melting. A difference of +-5 days (which corresponds to the S2 repetition time) already introduces around 50 mm of RMSE.

[Figure]

*Figure 17 tSD 27/06/2019, tS1=22/04/2019 (as detected by the station), a=4.8 mm/(°Cd), SWE threshold 2 mm, tSA=22/11/2018 +- 15 days.*

It is possible to see in Figure 17 that the shift of tSA does not strongly affect the RMSE as tSD does. For negative shifts, the accuracy RMSE is constant since no SWE is added to the reconstruction. In fact, for those days, we find that the coefficient k (see Eq. 4) is 0 since it is calculated from the AWS. In other words, it means that the accumulation is not really happening before at least one station detects an increase in SWE.

[Figure]

*Figure 18  tSD = 27/06/2019, tSA=22/11/2018, a=4.8 mm/(°Cd), SWE threshold 2 mm, tS1=22/04/2019 +- 15 days.*

In this case, it is also possible to see in Figure 18 that the shift of tS1 does not strongly affect the RMSE as does tSD. The RMSE for negative shifts remains constant after a certain point, since for those days the DD model returns 0 potential melting, so there are no differences. This means that it is in general better to make an error anticipating the melting phase than postponing it.

Even though we are aware that this represents a very simplified analysis and might be not exhaustive, we can summarize that we expect that the error that most strongly affects the results is a shift in the date of snow disappearance. For this reason, we believe that the SWE reconstruction can fully benefit from the introduction of an accurate daily HR time series. However, also possible errors introduce in the potential melting calculated through the degree day play an

important role. An error of 150 mm is introduced when using a coefficient of 1 mm/°Cday higher than the optimum. A similar error is obtained when considering a tSD of +- 5 days (that is the acquisition frequency of Sentinel-2 images).

However, in our opinion, the most interesting way to improve the current results is to move from modeling to actual observations that are intrinsically spatialized. We are currently working toward a solution that may consider remote sensing information only (as mentioned also in the old version of the manuscript at L498-500), for example by considering surface temperature measurements from satellites, meteorological radar, or SAR-derived snow depth/SWE information. However, this needs further research and a community effort, and it is left in this work as future development.

Regarding sublimation, we do not implicitly consider it in the method as well as wind redistribution or gravitational transport. In detail, the SWE reconstructed by the proposed method represents the amount of snowmelt water, net of sublimation. This, together with the connected limitations in the applicability of the method, has been better stated in the Method Section. However, while sublimation, redistribution and gravitational transport cannot be explicitly considered in the method, we can still observe their effects on a different snow persistence on the ground, which is detectable in the HR SCA time-series. We believe that this represents one of the advantages of using a daily HR SCA time-series as input to the method.

As it was stated in L190 we do not consider subdaily variations: "However in this work, we do not consider subdaily variations but only changes that are sampled in the temporal resolution of the exploited HR SCA time-series, i.e., one day." We agree that accumulation and ablation happen within an hourly scale. However, our method does not aim to achieve such a detail. Given the general objective of the proposed method, the overpassing time of Sentinel, the many assumptions made, e.g., the use of a simple degree day model, the lack of a method to evaluate the snowfall limit, the scarcity of temperature measurements that would not allow an accurate interpolation with a subdaily detail. But this is an interesting direction for future works.

- In section 4, the results are shown as selected points/transect/ periods in the study catchments, and detailed datasets are included as appendixes. The selection must be justified in all cases. The associated figures and tables' captions must include the catchment name in all cases (see figures 5 to 7, and table 1). Some sentences lack a proper justification, for example, lines 389 and 399 contain comments that can't be rigorously concluded in general from what has been shown. Or line 408, regarding Fig. 13, has a mass balance closure test been done? Figure 9 caption, are these "trends"?

Thank you for these suggestions. For sake of brevity, we reported only some of the results in the main draft. While we think that it might be interesting to show them in the Appendix, putting all of them in the main manuscript would be too much. Additionally, the metrics reported in L396 are already calculated for all the references and give an idea of the accuracy of the method. However, for completeness, we also added Figure 19 where we show the scatterplot between the measured and proposed SWE for the complete dataset as required by Reviewer#2.

[Figure]

*Figure 19 Observed and proposed SWE in the Schnals catchment for the h.y. 2020/21.*

We included the catchment name in the Table/Figures.

We agree that the statement in L389 is not scientifically rigorous, hence we removed it. With L399 we that we have two factors that affect the SWE determination, i.e., the degree day that we use to calculate the potential melting and also the snow duration on the ground. If for example, two close pixels have a very similar temperature but differ in aspect, the resulting potential melting does not differ as it is calculated based on temperature only. However, the different snow persistence on the ground will result in a different amount of SWE for those pixels. This discussion has been added to the manuscript.

According to the other reviewers' comment, we removed Fig. 13 and we deepened the analysis on the relationship between the SWE runoff and the measured discharge. We considered the SWE related to the subcatchment closed at the outlet point Schnalserbach – Gerstgras, as shown in Figure 20. This makes the two variables more comparable, although the analysis remains qualitative and a mass balance closure test has not been performed.

[Figure]

*Figure 20 Overview of the subcatchment whose outlet point corresponds to the location of the discharge measurement.*

We presented two new plots (Figures 21 and 22) for the two seasons that may replace Figure 13 in the manuscript. The analyzed variables are: i) the SWE variations that are associated with a runoff (i.e., only when they are associated with a decrease of SWE), ii) the discharge; and iii) the precipitation measured at Vernagt expressed in mm/day. We better analyze what we can observe from these plots. In detail, we can observe that there is a good agreement in terms of both timing and quantity among snow-generated runoff and discharge confirming that the catchment is snowmelt dominated. The discharge starts increasing in correspondence with the snowmelt and it starts decreasing when also the snowmelt is reduced for both periods. We can observe that the first year shows a delay while the response is more direct for the second season. More than differences in terms of precipitation, we ascribe this situation to a different snowmelt rate. Indeed, the season 2019-20 shows an earlier, weaker, and longer distributed snowmelt period, interrupted by periods with low SWE output (such as the end of March-beginning of April, beginning of May, or middle of June). This situation may favor ground infiltration with a predominance of subsurface runoff w.r.t. surface runoff, that contributes slowly to the discharge. On the other hand, the 2020-21 season shows a long and high intensity SWE release (end of May-end of June) that may cause a sudden saturation of the soil, with predominant surface runoff that contributes more directly to the discharge. This hypothesis may also be confirmed by recent literature, showing that when snowmelt is earlier, it is also less intense and the runoff response could be reduced, with strong implications for future climate change impacts (Musselman et al., 2017). However, other contributions should be also considered, as for example, the storage of water in the two snow reservoirs that are present in the territory. While a proper analysis requires a complete hydrological study and a hydraulic characterization of the watershed properties, we believe that this simplified analysis shows the potentiality of the presented results in a real application. For this, we think that adding the information provided here to

the Reviewer also to the manuscript may be interesting to the reader and stimulate works that will exploit data derived from the proposed approach.

[Figure]

*Figure 21 Snow generated runoff, discharge and precipitations for a subcatchment in the Schnals valley for the season 2019/20.*

[Figure]

*Figure 22 Snow generated runoff, discharge and precipitations for a subcatchment in the Schnals valley for the season 2020/21.*

- The discussion in section 5 repeats many facts or comments that have been previously presented or commented. Moreover, the discussion is focused on the sources of error at each step of the proposed method. I miss the discussion on the goodness of the results when compared to other products/methods/data sources that provide less resolution, or other standard or alternative existing methods. This is important as HR SWE mapping is the target goal.

The discussion on the goodness of the results has been expanded. We added the scatterplots as rquired by the Reviewer and commented on that (see next answer). Furthermore, beside comparing the obtained results with ASO data, we propose a comparison with a product at 500 m (Fang et al., 2022), as reported in the answer to Reviewer#2. However, we believe that the ASO dataset remains the key reference having that product a comparable spatial resolution. However, it is worth discussing why we think that it is difficult to estimate in a quantitative manner the improvement due to the use of HR data. First, a proper comparison should preferably refer to the obtained SCA time-series rather than the SWE time-series in order to isolate the errors coming from the reconstruction method. Having said that, several works have already shown the benefits introduced using HR data (e.g., Aalstad et al., 2020 for citing the most recent). In fact, SCF retrieved by LR sensors presents some intrinsic limitations, e.g., variable viewing angle that changes the spatial resolution inside the image, high heterogeneity of land cover, illumination and atmospheric conditions inside the resolution cell especially over mountainous areas (see e.g.,

Rittger et al., 2016).   The development of a robust and accurate algorithm to SCF retrieval able to address all the aforementioned problems has been the main research topic of the last 20 years. On the other hand, as showed in our recent work (Premier et al., 2021), it is possible to reconstruct a daily HR SCA time-series that presents high accuracy by learning from the historical snow patterns that repeat inter-annually. In this work, we used the information provided by LR sensors more as an indication than as absolute truth, being aware that it presents an uncertainty (see (Premier et al., 2021), for more details). In this sense, the LR SCF is also corrected during the process. Hence, when using this corrected LR dataset to reconstruct the SWE we obtain results in line with the ones aggregated at 500m and first derived at HR by the approach presented in this paper. In other words, if the LR time series is accurate and filtered out from the above-mentioned problem, the real benefits provided by the time series at HR is an incontrovertible geometric detail, which might be of paramount importance for some applications (e.g., hydrology, avalanche forecasting, ecological studies, etc). This is also in line with the findings of Bair et al., 2022.  It is finally worth stressing the fact that also the different temporal resolution plays an important role in the final evaluation of the SWE reconstruction. Therefore, considering partial time series instead of the completed ones can introduce artificial errors that do not allow one to properly quantify the advantage of using LR or HR data.

We take the opportunity raised by this comment to evaluate the results obtained by the proposed approach against a further dataset that is available at 500 m resolution (Fang et al., 2022), but that is derived by assimilating Landsat data. The metrics obtained for the two methods are shown in the Table. Also, the figures report the results for the three hydrological seasons.

| Date | BIAS [mm] | | RMSE [mm] | | Correlation [-] | |
|---|---|---|---|---|---|---|
| | proposed | NASA | proposed | NASA | proposed | NASA |
| 17/03/2019 | -121 | 36 | 242 | 292 | 0.80 | 0.66 |
| 02/05/2019 | -61 | -4 | 208 | 307 | 0.90 | 0.77 |
| 09/06/2019 | -25 | -32 | 182 | 302 | 0.93 | 0. 79 |
| 04/07/2019 | -49 | -14 | 129 | 201 | 0.90 | 0.71 |
| 14/07/2019 | -51 | -20 | 125 | 163 | 0.84 | 0.65 |
| 15/04/2020 | -73 | -26 | 159 | 169 | 0.80 | 0.78 |
| 05/05/2020 | -59 | 5 | 151 | 154 | 0.82 | 0.80 |
| 23/05/2020 | -95 | -27 | 179 | 150 | 0.79 | 0.78 |
| 08/06/2020 | -25 | 3 | 96 | 100 | 0.72 | 0.70 |
| 26/02/2021 | 5 | 96 | 92 | 157 | 0.75 | 0.65 |
| 31/03/2021 | 74 | 119 | 124 | 202 | 0.85 | 0.72 |
| 03/05/2021 | 65 | 54 | 121 | 121 | 0.89 | 0.66 |

[Figure]

*Figure 23 Results for the South Fork catchment of the San Joaquin river for the h.y. 2018/19.*

[Figure]

*Figure 24 Results for the South Fork catchment of the San Joaquin river for the h.y. 2019/20.*

[Figure]

*Figure 25 Results for the South Fork catchment of the San Joaquin river for the h.y. 2020/21.*

It is possible to see that the two time-series show a similar trend. Generally, we encounter lower BIAS for the NASA product while lower RMSE and higher correlation are observed for our product.

- The error indicators in results cannot be properly valued since little information is included from the study catchment in terms of SWE regime, in section 3.

Thank you for this interesting comment. Given the availability of SWE records for the South Fork catchment starting from 2000 in the Volcanic Knob station, we performed an analysis of the SWE records. In the following plots, you can appreciate the average SWE for all the recorded years, the range of variation and the current SWE (referred to the seasons 2018/19, 2019/20 and 2020/21). It is possible to see that the first season presents high SWE, while the other two seasons are under the average regime. Unfortunately, SWE records are not available for the Schnals catchment. Additionally, we do not have long records of snow depth observations that allow us to perform a similar analysis. However, the two seasons are in average with the typical snow conditions in Schnal.

[Figure]

*Figure 26 SWE trend for the h.y. 2018/19.*

[Figure]

*Figure 27  SWE trend for the h.y. 2019/20.*

[Figure]

*Figure 28 SWE trend for the h.y. 2020/21.*

- The discussion/conclusions should also include more reference to what processes can be tracked from the time series obtained of these SWE maps, and what cannot due to the assumptions, etcetera in the approach. This is very relevant to address the further applicability of the method.

Thank you for pointing out this important aspect. We have expanded the discussion on possible applications of our time-series. In detail, we have seen that the time-series provides accurate results, especially when considering the catchment scale and the late melting phase. For this reason, it can be exploited in hydrological applications such as streamflow forecasting, especially in regions where complex topography generates complex snow accumulation patterns and differential melting times (Tappeiner et al., 2001), which controls, besides hydrological processes, also key ecological processes in mountain regions (Park et al., 2021; Seeber et al 2021).  Moreover, our approach could be particularly suitable for remote regions where poor data on precipitation amount are available, since it is independent from this information. Also, we recommend applying this approach in environments that are not characterized by strong sublimation processes as well as catchments that present, as discussed earlier, frequent mixed states as snowfalls at high elevations while rain-on-snow at lower elevations. Furthermore, the method can be exploited to provide also historical SWE reanalysis time-series (at least a LR sensor should be present for the SCA reconstruction) that may allow understanding the effects of climate changes.

Some additional comments:

In general, the English usage and edition is good, but some revision is recommended.

Thank you for the suggestion. We also believe that the English language copy-editing that the journal offers in case of acceptance will further help improving the quality of the writing

Please, review the use of some wording. For example, line 381, "while the others (seasons) are drier" really means snow-scarce, which can also be due to high temperature; or the use of "bias" in the work to define "difference" or absolute error.

Thank you for the suggestion. We considered the use of new wording.

When some references are included in a list, please, use a constant criteria to order (increasing or decreasing date)

Thank you for the suggestion. We ordered the references by considering an increasing date.

Reference in line 35 looks not recent enough to be a updated review for remote sensing products, at least, some others could have been included.

Thank you for the suggestion. We added the work presented by Dong (2018).

Line 65, please provide some reference, there are works on that (i.e. Pimentel et al., 2015;2017; or others).

Thank you for the suggestion. We added the following references:

Pimentel, R., Herrero, J., Zeng, Y., Su, Z., & Polo, M. J. (2015). Study of snow dynamics at subgrid scale in semiarid environments combining terrestrial photography and data assimilation techniques. Journal of hydrometeorology, 16(2), 563-578.

Pimentel, R., Herrero, J., & Polo, M. J. (2017). Subgrid parameterization of snow distribution at a Mediterranean site using terrestrial photography. Hydrology and Earth System Sciences, 21(2), 805-820.

Please, assess the error associated to the ASO product, taken as ground-truth to test the results.

As reported in the paper presented by Painter et al., 2016, the SWE uncertainty is dependent on both the uncertainty associated with the snow depth retrieval and the uncertainty associated with the snow density modeling. The authors report "Our best understanding thus far is that snow depth uncertainty at the 3 m resolution is unbiased with RMSE of 0.08 m, resulting in depth uncertainty of < 0.02 m at 50 m resolution. With the snow density uncertainties of 13–30 kg m− 3 described above, we can estimate scenarios of SWE uncertainty. For a snowpack of 0.5 m depth and 100 kg m− 3, the SWE uncertainty is about 1 cm relative to the 5 cm actual. For a snowpack of 4.5 m depth and 450 kg m− 3, the SWE uncertainty is 10 cm relative to the 203 cm actual." This information has been added to the manuscript.

Beyond the comparison of results and ASO in the appendixes, dispersion graphs are needed to further assess the performance of the method, and some selected cases should be included in the results' section.

Thank you for your suggestion. We revised the results section. Here we include the dispersion graphs. It is possible to notice a generally good correlation between the proposed and ASO SWE. However, there is a tendency of observing a high dispersion that is also confirmed by the computed RMSE value. However, most of the points that are distant from the diagonal are outliers (low density). The year 2019 does not show a specific trend and the shape of the" cloud" is generally more symmetric. In the year 2020, it is possible to see that our product tends to underestimate SWE, except for very high SWE values (mainly outliers). On the other hand, our product tends to overestimate SWE in the year 2021. The worse correlation in the season 2020 was encountered also from the average value reported in the manuscript. However, given the high number of compared pixels, it is difficult to understand if there is a systematic error that generates the outliers. This analysis has been added to the revised version of the paper.

[Figure]

Figure 29 17/03/2019

Figure 30 02/05/2019

Figure 31 09/06/2019

Figure 32 04/07/2019

Figure 33 14/07/2019

Figure 34 15/04/2020

Figure 35 05/05/2020

Figure 36 23/05/2020

[Figure]

*Figure 37 08/06/2020*

*Figure 38 26/02/2021*

*Figure 39 31/03/2021*

*Figure 40 03/05/2021*

**References**

Aalstad, K., Westermann, S., & Bertino, L. (2020). Evaluating satellite retrieved fractional snow-covered area at a high-Arctic site using terrestrial photography. Remote Sensing of Environment, 239, 111618.

Bair, E. H., Dozier, J., Rittger, K., Stillinger, T., Kleiber, W., & Davis, R. E. (2022). Does higher spatial resolution improve snow estimates?. The Cryosphere Discussions, 1-20.

Fang, Y., Y. Liu, and S. A. Margulis. (2022). Western United States UCLA Daily Snow Reanalysis, Version 1 [Data Set]. Boulder, Colorado USA. NASA National Snow and Ice Data Center Distributed Active Archive Center. https://doi.org/10.5067/PP7T2GBI52I2. Date Accessed 12-12-2022.

Fehlmann, M., Gascón, E., Rohrer, M., Schwarb, M., & Stoffel, M. (2018). Estimating the snowfall limit in alpine and pre-alpine valleys: A local evaluation of operational approaches. Atmospheric Research, 204, 136-148.

Dong, C. (2018). Remote sensing, hydrological modeling and in situ observations in snow cover research: A review. Journal of Hydrology, 561, 573-583.

Fehlmann, M., Gascón, E., Rohrer, M., Schwarb, M., & Stoffel, M. (2018). Estimating the snowfall limit in alpine and pre-alpine valleys: A local evaluation of operational approaches. Atmospheric Research, 204, 136-148.

Marin, C., Bertoldi, G., Premier, V., Callegari, M., Brida, C., Hürkamp, K., ... & Notarnicola, C. (2020). Use of Sentinel-1 radar observations to evaluate snowmelt dynamics in alpine regions. The Cryosphere, 14(3), 935-956.

Musselman, K. N., Clark, M. P., Liu, C., Ikeda, K., & Rasmussen, R. (2017). Slower snowmelt in a warmer world. Nature Climate Change, 7(3), 214-219.

Painter, T. H., Berisford, D. F., Boardman, J. W., Bormann, K. J., Deems, J. S., Gehrke, F., ... & Winstral, A. (2016). The Airborne Snow Observatory: Fusion of scanning lidar, imaging spectrometer, and physically-based modeling for mapping snow water equivalent and snow albedo. Remote Sensing of Environment, 184, 139-152.

Park, D. S., Newman, E. A., & Breckheimer, I. K. (2021). Scale gaps in landscape phenology: challenges and opportunities. Trends in Ecology & Evolution, 36(8), 709-721.

Pimentel, R., Herrero, J., Zeng, Y., Su, Z., & Polo, M. J. (2015). Study of snow dynamics at subgrid scale in semiarid environments combining terrestrial photography and data assimilation techniques. Journal of hydrometeorology, 16(2), 563-578.

Pimentel, R., Herrero, J., & Polo, M. J. (2017). Subgrid parameterization of snow distribution at a Mediterranean site using terrestrial photography. Hydrology and Earth System Sciences, 21(2), 805-820.

Premier, V., Marin, C., Steger, S., Notarnicola, C., & Bruzzone, L. (2021). A Novel Approach Based on a Hierarchical Multiresolution Analysis of Optical Time Series to Reconstruct the Daily High-Resolution Snow Cover Area. IEEE Journal of Selected Topics in Applied Earth Observations and Remote Sensing, 14, 9223-9240.

Seeber, J., Newesely, C., Steinwandter, M., Rief, A., Körner, C., Tappeiner, U., & Meyer, E. (2021). Soil invertebrate abundance, diversity, and community composition across steep high elevation snowmelt gradients in the European Alps. Arctic, Antarctic, and Alpine Research, 53(1), 288-299.

Tappeiner, U., Tappeiner, G., Aschenwald, J., Tasser, E., & Ostendorf, B. (2001). GIS-based modelling of spatial pattern of snow cover duration in an alpine area. Ecological Modelling, 138(1-3), 265-275.

Tobin, C., Schaefli, B., Nicótina, L., Simoni, S., Barrenetxea, G., Smith, R., ... & Rinaldo, A. (2013). Improving the degree-day method for sub-daily melt simulations with physically-based diurnal variations. Advances in water resources, 55, 149-164.

---

## Author Response (AR2)

I am pleased to see that the authors now present an improved version of the manuscript including not only a take on a clearer description of the methodology that is now easier to follow, but also providing additional analysis. Nevertheless, I would still recommend that parts of the manuscript be reworded to further improve clarity of method and readability. Below you can find some minor and technical comments.

We thank the Reviewer for the comments that have contributed to improving the clarity of the manuscript. Please find in blue our point-to-point answers.

Specific comments

L5-6: it is not clear to me if you used precipitation data, or SWE/ SD recordings to determine the "accumulation" state (see comments to section 2.1.1). In section 2.2.2 it is stated that the "ablation" state also depends on the degree day (DD > 0).

Thank you for the comment. With the aim to be as general as possible, we meant with "precipitation data" both the data recorded by pluviometers as well as snow depth/SWE sensors. Since the differentiation from liquid to solid precipitation may be difficult using pluviometer data, and since in this work we do not use this data, we simplified the sentence as follows "snow depth/SWE and temperature data" instead of "precipitation data". In fact, you are correct that also temperature data are used to determine the DD and consequently the ablation state.

This is the in-situ data that we are actually using in this work since they are available in our study areas. However, we discuss the possibility of using precipitation data – that needs to be converted to solid precipitation –  in section 2.1.1 also according to another comment raised by the Reviewer later on in this revision.

L168-191: In this section, the authors give a range of possibilities about how an "accumulation" state could be inferred from different data sources. Even though, a closing summary sentence is provided for the section, I still struggle to fully understand the specifics of the method. I suggest restructuring this section to clearly separate i) suggestions about what approaches could be used, and what are the pros and cons of these approaches, and ii) what is specifically used in this study.

Thank you for the comment. We restructured the session as suggested by the Reviewer. Here we report the restructured paragraph:

The *accumulation* identification can be retrieved from a network of automatic weather stations (AWSs) that provide continuous information about the occurrence and elevation of snowfall events, for example, direct SWE measurements or indirect precipitation/SD measurements. Continuous SWE measurements are unfortunately scarcely available. By mean pluviometers and temperature observations, it is possible to split precipitation between liquid and solid (Mair et al., 2013) and identify the state accordingly. However, these stations are rarely installed at high elevations. SD sensors are more suitable for our purpose, but their observations are often affected by wind and gravitational transport, leading to deposition/removal that can be falsely interpreted as *accumulation*/*ablation*. Hence, even if the AWSs are generally situated in locations undisturbed from the wind action, it is more convenient to dispose of a large number of AWSs that need to be filtered to exclude possible sensor errors or wind/gravitational redistribution. In fact, a station is usually representative of a limited area whose extension is highly variable depending on the complexity of the terrain. In the case of a well-monitored area with stations distributed with elevation, it is possible to divide the catchment into different elevation belts where the snowfall events can be considered nearly homogeneous. However, in many basins, this might be far from reality. As a common configuration for snow monitoring, we have a single station located at a high point of the catchment, which is informative enough to identify the accumulation events but not their extent. Furthermore, it has been shown in the literature that estimating the snowfall limit can be very challenging (e.g., Fehlmann et al., 2018).

Therefore, the method can be adapted based on the availability of the in-situ observations. In this study, we made use of SWE and snow depth measurements, since they were available in our analyzed basins. To consider that a snowfall is occurring, the increase in SWE/SD should be greater than a certain threshold SDmin/SWEmin that is fixed at 2 cm for SD according to the values found in the literature (Engel et al., 2017), resulting in a value of 2 mm for SWE when considering the typical density of fresh snow (100 kg/m3).  Unfortunately, the basins are poorly monitored in terms of these variables and consequently, the snowfall limit cannot be estimated in an appropriate manner. Hence, we consider that snowfalls occur throughout the snow-covered area of the catchment. We acknowledge that this assumption may result in less accurate SWE estimation, especially in the case of mixed states. For example, snowfall can be observed at high elevations, together with rain-on-snow at low elevations, causing

snowmelt. However, we assume that the effect of these events on the total SWE balance is small enough to be considered negligible as we will discuss in Sec. 5.2.

In summary, when the AWSs show an increment greater than a defined threshold, we identify the state as *accumulation.*

L194: DD models are not based on intuition (e.g. Ohmura, 2001) Ohmura, A., 2001: Physical Basis for the Temperature-Based Melt-Index Method. J. Appl. Meteor. Climatol., 40, 753–761, https://doi.org/10.1175/1520-0450(2001)040<0753:PBFTTB>2.0.CO;2.

Thank you, we rephrased with "..which is an empirical model that makes use of air temperature as a proxy of the melting" and added the reference.

L211: it is not clear to the reader why focussing on a "u-shaped" backscattering signal is logical consequence from the methodical issues in forests.

Thank you for noticing this important logical step. We need to clarify that it is still possible to have the "U-shape" in forest. However, the canopy interferes with the signal, and this can result in a "noisier" backscattering signal. A recent work proposed by Darychuk et al., 2023, has proposed an interesting analysis in this sense. They show that the "U-shaped" backscattering signal is more evident in open areas compared to mature forest. They also show that the timing of the runoff onset and duration were less reliable under canopy. However, the intensity of the scattered signal by canopy may depend on the structure, composition, and stem density of forest. Thus, to simplify the processing, we applied the simple rule to consider as valid the runoff onset detected when a clear "U-shape" is visible. We rephrased that part: "As shown by Darychuk et al., 2023, the characteristic "U-shape" signature of the backscattering signal is less evident in mature forest, also depending on canopy structure, composition and density. The signal can be noisier due to the scattered contribution by canopy. Hence, we propose applying the method only for pixels that present a clear "U-shape" backscattering", which can be also present for forested areas."

L290: in line 400 it is stated that the DD factor is held constant during all seasons, please clarify.

Yes, the DD factor is held constant during all seasons. We used the first season to "calibrate" it in the station and then we evaluated the performances for the other seasons. We rephrased the sentence to clarify this point: "It is worth noting that the first year, 2018/19, is also used to set up the DD factor *a* that is then kept constant for the other seasons."

L431 it is tough to see anything in figure 6, regarding the transect locations. Is there any topographic indication why the mid elevation point is so underestimated?

We changed the color of the points of the transect in Fig.6. The purpose of Fig. 8 was to show that also the reference might show an unexpected behavior. In fact, it is true that our proposed SWE at the mid-elevation is probably showing an underestimation, and this is most probably related to a too-late runoff onset estimation. Note that the peak starts decreasing very late if compared both with ASO and the WUS-SR dataset. Further investigations reveal that the mid-elevation point is in a high-density forest area. We will add this consideration to the manuscript. In a previous answer, we were discussing that the runoff onset for forested areas might be less reliable. However, the canopy presence may also lead to overlapping effects as for example canopy interception, differential heat transfers due to the presence of branches in the snow or impact on the surface albedo. This might lead to very different behavior of the snowpack w.r.t. open areas. These overlapping effects are difficult to estimate, and this could be the reason why ASO is also showing an unexpected behavior as we were already addressing in the manuscript. In fact, SWE for the mid-elevation point is unexpectedly much larger than the SWE for the higher point. The presence of forest might affect the quality of the snow depth measurements with lidar or the snow density estimation.

L444: To provide only the very low bias of -5mm in the Abstract I find almost misleading, I must admit. Consider giving more information (eg rmse or r²).

We agree with the reviewer and the only reason for this decision was to take out from the abstract some of the details. We added also RSME and correlation. Furthermore, we also added the metrics that we obtained when considering the WUS-SR dataset. "It obtained good agreement both when evaluated against HR spatialized reference maps (showing an average bias of -22 mm, a root mean square error (RMSE) of 212 mm, and a correlation of 0.74), against a daily dataset at coarser resolution (showing an average bias of -44 mm, an RMSE of 127 mm, and a correlation of 0.66) and against manual measurements (showing an average bias of -5 mm, an RMSE of 191 mm, and a correlation of 0.35)."

L561: here you state you used SD and SWE data for state indentification. This contrasts the abstract (precipitation)

As explained above, we meant with precipitation data in general what is detected by pluviometers (that however makes the estimation of the new snow more challenging) as well as snow depth/SWE sensors that also return a quantification of the snowfall amount. To make it clearer, we changed this in the abstract with "snow depth/SWE and temperature data" instead of "precipitation data". We also added the temperature as a needed variable for the state determination in the conclusions. These are the variables that we are actually using in this work since they are available in our study areas. However, an interesting future development of this work can be the use of precipitation data – that however needs to be converted to solid precipitation – as discussed in section 2.1.1 also according to your previous comment.

L583: iii) is unclear to me.

We rephrased the sentence in this way: "provide consistent trends when considering the basin scale".

Technical comments
Abstract: make sure no paragraphs are doubled: eg L11-22 and 31-39 should be removed.

Thank you for noticing it. In fact, the paragraph was repeated by mistake.

L83: only true for snow regimes with distinct accumulation and ablation periods.

Thank you. We added this for clarification.

L97: not only to the peak of the accumulation, as you are presenting a method that goes beyond that (and this is still considered reconstruction).

Thank you for noticing it. We removed "up to the peak of the accumulation" to leave it more general.

L100: a general "outperforming" against snow models is a bold statement!

We softened the sentence in this way "The SWE reconstruction approaches showed good performances over large basins and even mountain ranges".

L143: most importantly, the state information is used to add or remove SWE during the reconstruction

This is already written clearly a few sentences later: "The state information is used again in this step i) to redistribute the total amount of SWE calculated for the melting in the accumulation period, and ii) to include late snowfalls that occur after the peak of accumulation to the reconstruction." Hence, we think that we can avoid this repetition.

L152: "Let us introduce": consider change of style.

We changed this with "Three states are introduced…".

L158: which variations? Be more specific.

Thank you for this suggestion. We changed "variations" with "daily increment".

Figure1: "Catchment State" ?

Thank you for noticing it. Accordingly with the revised version of the paper, we will change it with "State".

L159: "melted snow" = melting ?

Thank you, we replaced it.

L486:"..which in turn depends on the quality of air temperature recordings, the interpolation …"

Thank you, we replaced it.

I would like to acknowledge the Authors for the great work done in this revised version, which addresses all the reviewers' comments and concerns. I want to especially thank you for the level of detail and care you have included in the point-to-point answers, including those to minor comments in my previous report.

My major concerns have been addressed and the very much improved readability of methods, together with the additional information provided, have facilitated a clear view of the sequential steps and the validity (or not) of the underlying hypothesis. I especially value the so much improved methodological section. Additional information to support different decisions in the approach's building has been satisfactory to understand potential impacts, and the transferability of the results.

We also would like to thank the Reviewer for the effort and the useful comments provided during the review process. We are grateful for his positive feedbacks.

At this point, I just have some minor comments to be assessed in the new version:

1. In their answer to my comment on Section 2.1, the Authors discuss on the assumption of homogeneous occurrence of snow throughout the snow-cover area. From the dispersion graphs newly added, it is easily seen how during 2020 there is an overestimation of the model. From your knowledge of the meteorological data sets during the different years, do you think this is due to such simplification? I am curious about potential differences in the snow cover pattern, and whether that season was more torrential in terms of precipitation.

Thank you for the comment. This is a very interesting comment that is however difficult to be answered without further investigation. In the following, we will try to make some considerations about this concern.
In fact, the season 2020/21 shows an overestimation of the model when comparing the results against ASO. This season was very dry, as you can see from Figure A1. We agree with the Reviewer that possibly we had some snowfalls that happened above the actual snowline elevation. However, from our experience, we have noticed that this kind of error does not so largely affect the results, especially when considering the melting period (as most of the ASO observations). In fact, we remember that the total SWE amount on a single pixel is dependent only on the days that are in "ablation". If we assign more days as the real ones to be in "accumulation", we are simply redistributing the total SWE over more days, thus affecting the actual shape of the SWE trend but not really the peak of SWE. Also, when comparing the results with the WUS-SR dataset, it seems that we are underestimating the most relevant snowfall happening at the end of January. This is for sure a snowfall that involves all the snow-covered parts of the catchment and covers the bare soil, as it is possible to notice in Figure 13c where an important increment in terms of SCA is shown. Hence, we do not directly ascribe the differences w.r.t. ASO to the redistribution of snow throughout the snow-covered area. It is also very difficult to state if the reference is completely reliable given the inverse tendency when comparing our model with WUS-SR. However, we believe that errors can be introduced by the SCA correction in the late season (May-June). As we also discussed in the text ( Section 5.2), "an evident case where an overestimation of the SCA is introduced is in the season 2020/21 in May/June for the SFSJR (see Fig.13c). This overestimation is caused by the fact that the AWSs do not indicate an *accumulation* in correspondence with the peaks that occur in the late melting phase. Therefore, the label is corrected according to the majority rule in *ablation* in the case of old snowfall. Most of the pixels were *snow* in previous HR acquisitions, leading to the propagation of the class backward and the consequent overestimation of the SCA. " In summary, we believe that an accurate SCA estimation is very relevant for this work, especially when introducing errors in the date of snow disappearance as shown in our simplified sensitivity analysis. Future work will focus also in this direction.

2. The discussion of the choices made in Section 2.3 is very fine. I still think, however, that the use of DD constitutes a hard limitation for a successful modelling when heterogeneity is large, which, on the other hand, is one compelling fact to obtain new methods that provide SWE on large areas. Figures showing the method performance as related to altitude, slope, and aspect show a relevant shift towards overestimation of SWE in the higher area especially when less snow is present. Do you have some idea on what Is influencing this mostly, error in snowfall threshold or the errors induced by DD methods?

Thank you for this comment. This point can be discussed by looking at Fig. C3. In our opinion, without further analysis, we cannot appreciate a real trend. In fact, we do not really observe a constant overestimation of SWE for higher elevation. From Figure C3, we can appreciate an overestimation, especially for season 2020/21, whose reasons are explained in the previous answer. For the other seasons, we observe a general underestimation of SWE, happening in some cases also when considering high elevations (see for example Fig. C3g). The biggest differences are encountered when considering the slope analysis. As we discussed in Appendix C in L633 (old version): "The slope analysis shows larger differences, especially for some dates (i.e., 9 June 2019 and the three images acquired for year 2020) and when considering steep slopes. The proposed method underestimates SWE w.r.t. ASO. However, we generally expect lower SWE for these steeper slopes that promote gravitational transport. The aspect analysis

suggests an underestimation for the north facing slope when comparing our dataset with ASO (except for year 2021)." We are not sure if these differences can be directly ascribed to the use of the DD model, since we would expect an overestimation for the North exposed areas, as the DD method does not consider radiation effects. This is discussed also in Section 4.1 L416 (old version).

Finally, we fully agree with the Reviewer about the limitation of the use of a DD model. In fact, we strongly believe that the future effort should go in the direction of fully exploiting remote sensing data, which has the advantage to be natively spatially distributed observations but which require advanced technical manipulations to be fully exploited in any snow monitoring systems e.g., current satellite missions, even at low resolution, are far to provide daily SCA time series which as discussed in the paper is one of the most important information that we can exploit to reconstruct SWE . Also, we believe that our method could be assimilated into a more reliable physically-based model that better estimates the potential melting and we leave this as future cooperative development.

3. I find the sensitivity analysis at the station site really interesting. I fully understand the limitations explained by the Authors to assess a more complete analysis, and agree that in any case, this adds valuable information to understand the results from the methodological approximation. Maybe some short assessment on the representativeness of this site in both precipitation conditions and the main drivers of snow dynamics could further facilitate the interpretation of the whole set of results.

Thank you for your comment. The Volcanic Knob station is located at an elevation of around 3050 m. Being the average elevation of the catchment around 3070 m, the station is thus well representing the mean behavior of the catchment. We will add this relevant information to the manuscript L321 (new version). Given the presence of relatively long-term records for the VLC station, we have presented in Appendix A – Fig. A1 – the climatology of SWE for the analyzed years. This gives an overview of the precipitation conditions at the site. We also would like to highlight that the San Joaquin basin is a snow-dominated catchment where high-elevation snowmelt strongly contributes to runoff during spring and summer months. The same is valid also for the Italian basin presented in the manuscript. However, to fully confirm the results of the simplified sensitivity analysis, it would make sense to extend it to more stations where snow pillows are available.

4. With the new additions and graphs, I'm not really sure that Figure 11 adds relevant information, and I would consider eliminating this as the hydrological balance is not the goal of the manuscript, nor is it treated in detail and completeness as the other results included in this final version. But this is just a minor reflection from my side, up to you to decide.

Thank you for your suggestion. We agree with the Reviewer that Fig. 11 is not completely representative and also that the hydrological balance is not the purpose of the paper. However, we would prefer to keep it since we believe it represents an interesting qualitative result for this second catchment where we unfortunately do not have a proper spatialized SWE reference. Furthermore, we believe that it can stimulate further research.

Thank you again for the work done and your consideration addressing all the answers. A very good work, indeed.

Thank you for your contribution and the many constructive feedbacks.